# On the longevity and inherent hermeticity of silicon-ICs: evaluation of bare-die and PDMS-coated ICs after accelerated aging and implantation studies

Kambiz Nanbakhsh[1], Ahmad Shah Idil [2,3,4,5], Callum Lamont[2], Csaba Dücső [6], Ömer Can Akgun[1,7], Domonkos Horváth [8,9], Kinga Tóth[8,9], Domokos Meszéna [8,9], István Ulbert [8,9], Federico Mazza [3], Timothy G. Constandinou [3,4,5], Wouter Serdijn[1,10], Anne Vanhoestenberghe[2,11], Nick Donaldson [2] & Vasiliki Giagka [1,12] ✉

Silicon integrated circuits (ICs) are central to the next-generation miniature active neural implants, whether packaged in soft polymers for flexible bioelectronics or implanted as bare die for neural probes. These emerging applications bring the IC closer to the corrosive body environment, raising reliability concerns, particularly for chronic use. Here, we evaluate the inherent hermeticity of bare die ICs, and examine the potential of polydimethylsiloxane (PDMS), a moisture-permeable elastomer, as a standalone encapsulation material. For this aim, the electrical and material performance of ICs sourced from two foundries was evaluated through one-year accelerated in vitro and in vivo studies. ICs featured custom-designed test structures and were partially PDMS coated, creating two regions on each chip, uncoated "bare die" and "PDMS-coated". During the accelerated in vitro study, ICs were electrically biased and periodically monitored. Results revealed stable electrical performance, indicating the unaffected operation of ICs even when directly exposed to physiological fluids. Despite this, material analysis revealed IC degradation in the bare regions. PDMS-coated regions, however, revealed limited degradation, making PDMS a suitable IC encapsulant for years-long implantation. Based on the new insights, guidelines are proposed that may enhance the longevity of implantable ICs, broadening their applications in the biomedical field.

Since the insertion of the first silicon integrated circuit (IC) into the brain for sensing neural activity[1,2], tremendous efforts have been made to exploit the high performance and power efficiency offered by the semiconductor industry in opening applications in the field of healthcare: brain-machine interfaces[3–5], flexible bioelectronics[4,6–9], biosensors[10–13], active silicon probes[6,14,15], and optogenetics[16,17].

Driven by the need for miniaturization, these emerging applications are moving away from the hermetic metal enclosures traditionally utilized for IC protection in the body and are opting for engineered

thin organic and inorganic coatings[6,18–23]. This shift, however, introduces reliability risks by bringing the chip closer to the corrosive body environment. Silicon-ICs are liable to failure if penetrated by body fluids: field effect transistors are susceptible to mobile ions (e.g., $Na^+$, $K^+$) that might reach the gate oxide[24,25], and water anywhere in the intricate sub-micron structures of the IC will facilitate corrosion and leakage currents[26,27]. Additionally, the body itself should be protected from the electrical bias voltages on the chip, which could pose a hazard if the IC's insulation is breached. These reliability risks and the uncertainty regarding the longevity of the IC are one of the main obstacles limiting the widespread clinical adoption of these emerging applications, particularly for chronic use[28–34].

The long-term reliability of implantable ICs relies on the stability and structural integrity of their constituent material stacks. Thus far, studies have been evaluating the stability of 'IC-related' thin-film materials such as silicon nitride ($SiN_X$) and silicon dioxide ($SiO_X$), mainly as standalone material films fabricated in research cleanrooms[21,35–37]. Limited data, however, exists on the long-term stability and inherent hermeticity of foundry-fabricated ICs (i.e. the ability of the IC structure itself to prevent moisture/ion ingress) upon their direct exposure to aggressive physiological environments.

Here, we evaluated the hermeticity of IC structures sourced from two complementary metal-oxide-semiconductor (CMOS) foundries by monitoring their electrical and material performance over long-term in vitro and in vivo studies. Our primary objectives were: 1) to determine the longevity of bare ICs when directly exposed to physiological media by identifying the degradation pathways that can lead to failure and 2) to evaluate the possibility of utilizing polydimethylsiloxane

(PDMS, silicone rubber) as a soft but moisture-permeable coating to extend the longevity of implantable ICs.

We demonstrate that foundry-fabricated IC structures can be inherently hermetic and waterproof. Despite this, direct exposure to physiological environments results in material degradation, limiting their longevity. In contrast, PDMS-coated regions of the ICs showed limited degradation, demonstrating the ability of the material to extend the lifespan of implantable ICs. In the context of bioelectronics packaging, these findings suggest that for the hermeticity, one can rely on the IC itself and use PDMS as a standalone easily accessible packaging material.

## Reliability concerns for implantable ICs
### Silicon IC structures
Silicon ICs are intricate multilayer structures composed of conducting metallization layers and insulating thin-film ceramic layers that function as intermetallic dielectrics (IMDs) (Fig. 1). The inherent hermeticity of an IC is determined by its structural integrity and the barrier properties of the materials constituting the die structure. From the top, ICs are protected by a dual layer of $SiN_X$ and $SiO_X$, usually created using plasma-enhanced chemical vapor deposition (PECVD) techniques. From the sides, IC structures are protected by a die seal ring (a stack of metallization encircling the die) which acts as a metal barrier, preventing impurity ingress from the sidewalls. From the bottom, the sensitive MOS structures are protected by a ~200–300 μm thick silicon substrate with high material density[38]. Given the protections from the bottom and the sidewalls, the hermeticity and longevity of bare ICs largely depend on the stability and barrier properties of the top

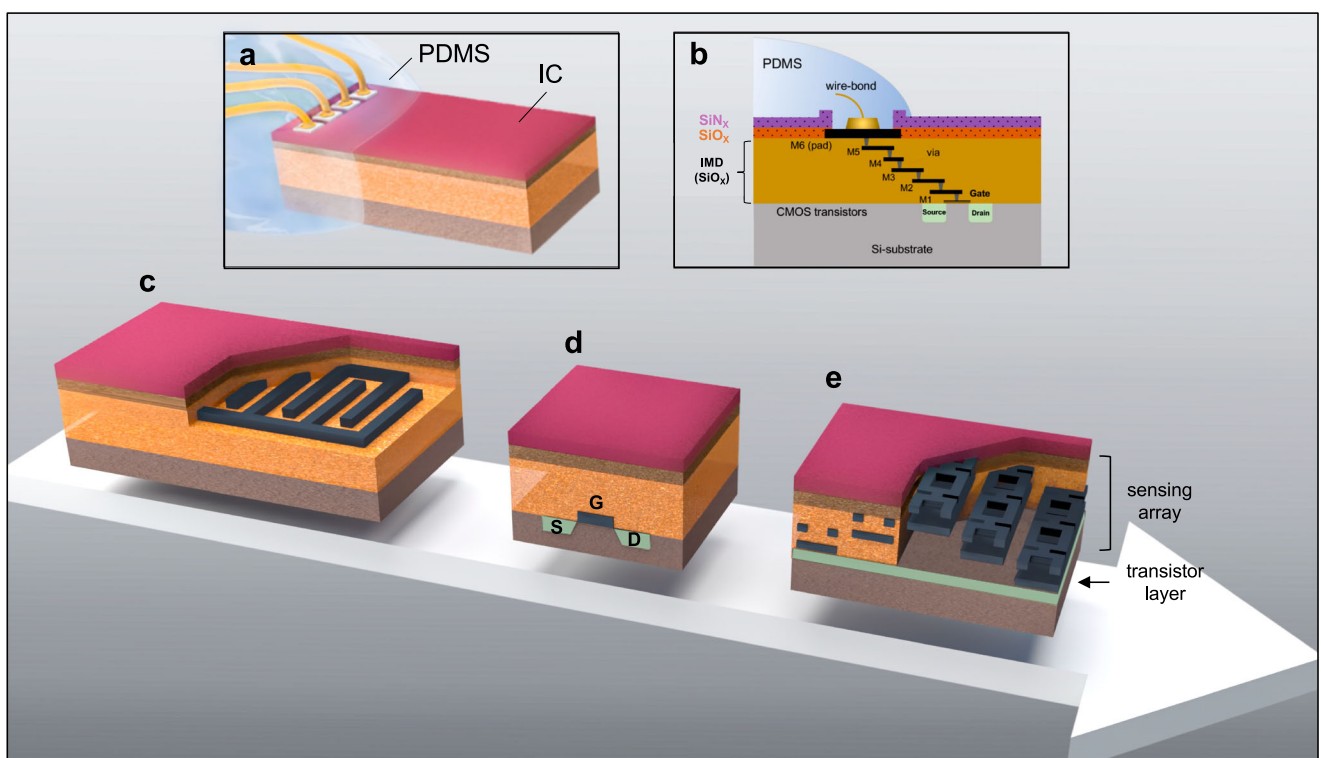

**Fig. 1 | Schematic illustrations of silicon-IC test structures (dimensions not to scale). a** A wire-bonded IC partially coated with PDMS, covering the wire-bonds and regions of the outer IC surface, leaving most of the IC structure and sidewalls exposed. **b** A cross-sectional schematic demonstrating the multilayer stack of a representative 6-metal CMOS process, from bottom to top: the source and drain diffusion regions implanted in the Si-substrate, metal layers 1 to 6 (M1 to M6) with M6 being the top-most metal, and where each metal layer is insulated with a $SiO_X$ intermetallic dielectric (IMD), and the final passivation layers generally made of $SiO_X$ and $SiN_X$. **c–e** Schematic of implemented test structures in silicon-IC, from simple to more advanced with, **c** an interdigitated capacitor (IDC) structure implemented using the top metal layers where the structure is closer to the surface, **d** A planar metal oxide semiconductor (MOS) transistor with the drain, source, and gate implemented in the Si-substrate boundary, making the MOS structures the most buried structure. **e** A dielectric sensor array with on-chip sensing circuitry for in situ monitoring of insulation and dielectric changes. Dielectric changes between different metal layers (M6-M5, M5-M4, and M4-M3) are sensed using on-chip MOS transistor circuitry.

passivation layers. While all these dielectric passivation materials are commonly identified as $SiN_X$ and $SiO_X$, their biostability and barrier properties can greatly vary based on the raw materials and unique processes used during manufacturing[21,39–41]. In addition, residual stresses within the IC's multilayer stack are equally important, as any interlayer delamination or cracks can jeopardize hermeticity[41,42]. All these key material properties determining the structural integrity and hermeticity of the IC structure are set by the unique manufacturing processes employed by each foundry.

### PDMS as a soft but moisture-permeable packaging material

Polymer packaging of ICs has been used to realize mm-scale devices[43–45]. Previous investigations, however, have shown that brittle IC passivation layers can be susceptible to cracking due to the shear stresses between the polymer (usually an epoxy compound with high Young's modulus) and the passivation[40]. Here, we propose the use of PDMS as a soft encapsulant for ICs. PDMS offers many advantages, including long-term proven biocompatibility, biostability, and a low Young's modulus[46] that can create a compliant interface for both the body and the IC. Despite these properties, the main perceived drawback of PDMS, which has discouraged its use as a standalone packaging material, is moisture permeability. PDMS is freely permeable to moisture (water vapor), with permeation rates that cause even cm-thick PDMS layers to be saturated with water vapor within a day of exposure to biofluids[47,48]. This property makes PDMS packaging non-hermetic in comparison to conventional techniques using metal or ceramic enclosures. Also, in contrast to thin-film moisture barrier packages, PDMS does not act as a water barrier. Instead, it ensures that the IC operates at 100% humidity, rather than being directly exposed to the ionic liquids and organic species in the body[48,49]. Under this condition, for successful PDMS packaging, one should not rely on the moisture barrier properties of PDMS but on the inherent hermeticity of the IC die structure itself. In addition to die hermeticity, in the wire-bond region with exposed metals, strong interfacial adhesion between the PDMS and IC material is required to prevent moisture condensation and shunt leakage paths[26,50] (Fig. 1a, b and Supplementary Note 1). Adhesion reliability itself is influenced by the PDMS type[26] (see Supplementary Table S4) and the surface chemistry of the IC's passivation layer, which again is greatly dependent on the manufacturing processes.

## Results

### Overview of IC test structures and aging study

Figure 1c–e and Fig. 2 show the custom-designed IC test structures. As mentioned earlier, the processes and materials used in the manufacturing of each IC can be different, impacting their hermicity and longevity. For this reason, two different CMOS foundries were selected, hereafter referred to as Chip-A (fabricated using a 0.35 μm 4-metal process) and Chip-B (fabricated using a 0.18 μm 6-metal process). Structures included 1) interdigitated capacitors (IDCs) implemented using different metal layers (Supplementary Note 2), 2) NMOS transistors and 3) a custom-designed dielectric sensor array. The dielectric sensor is capable of measuring IMD resistance values in the $10^{14}$ Ω range with a sub-MΩ sensitivity resolution[51]. Figure 2d, e gives a photomicrograph of the die structures and cross-sectional scanning electron microscope (SEM) images of the top material stack showing the passivation thicknesses and metallization used for the IDC structures. All design details are presented in the Methods and in Supplementary Tables S1 and S2.

All IC structures were partially coated with PDMS, creating two distinct regions on the same chip: 1) an uncoated "bare-die" region with most of the passivation and sidewalls exposed, and 2) a "PDMS-coated" region (Fig. 2a, b). On all prepared samples, the test structure under investigation was mainly left exposed (as bare die). This approach enabled us to evaluate how effectively the IC's own passivation and

dielectric layers can protect the sensitive test structures within. In the PDMS-coated regions, the PDMS thickness ranged from sub-micron (Fig. 4b) near the PDMS-edge, to a maximum of ~ 800 μm close to the wire-bonds (Supplementary Note 3). IC structures (n = 74) were either implanted in rats for a year (Fig. 2f, details in Methods) or subjected to a one-year accelerated in vitro aging study, which included electrical biasing up to 15 V in phosphate-buffered saline (PBS) solution at 67 °C (Fig. 2a, details in Methods and Supplementary Table S3). During the accelerated in vitro study, the electrical performance of all ICs was periodically monitored. Details of the preparation methods and set-ups can be found in the Methods and Supplementary Figs. S5 and S6.

Material analysis techniques were used to evaluate and compare the bio-degradation mechanisms in the bare-die and PDMS-coated regions of the aged chips, as well as the impact of the two aging environments (PBS solution at 67 °C and rat). Advanced analytical techniques, i.e., time-of-flight secondary ion mass spectrometry (ToF-SIMS) analysis, enabled us to look at these mechanisms for the first time at the nanometer level.

### Chemical composition of foundry IC passivation and IMD layers

On a group of reference samples (as supplied by the foundry), the chemical composition of the IC passivation layers was characterized using ToF-SIMS and X-ray photoelectron spectroscopy (XPS). The silicon nitride on both Chip-A and B ICs was measured to be non-stoichiometric with Si: 50% and N: 50% (N/Si: ~ 1), typical for PECVD deposited $SiN_X$ layers[42]. The hydrogen content in the nitride layers was measured using ToF-SIMS and showed to be different between the foundries with Chip-A being ~20% and Chip-B ~15% (See Supplementary Table S5 and Fig. S7).

The $SiO_X$ passivation layer for Chip-A showed a stoichiometric layer (Si: ~33% and O: ~67%, O/Si of ~2). For Chip-B, the silicon dioxide is comprised of two oxide layers. XPS results showed both layers having an O/Si of ~2 (Si: ~33% and O: ~67%). However, ToF-SIMS results showed the top ~100 nm oxide layer having a slightly higher [OH⁻] intensity (x2.5) than the bottom layer. The bottom $SiO_X$ layer had similar [OH⁻] intensities to the Chip-A $SiO_X$ passivation layer. The passivation layers contained very low carbon levels with no other detectable impurities (See ToF-SIMS depth profile results in Supplementary Fig. S7).

Note that for both ICs, the chemistry of the passivation layers is uniform across the entire IC. However, the thicknesses and conformality of the passivation layers are affected when using the topmost metal layers: M4 for Chip-A and M6 for Chip-B (See Supplementary Figs. S2 and S4).

The IMD layers for both chips were analyzed using energy-dispersive X-ray (EDX) elemental mapping on the IC cross-sections, which were exposed by focused ion beam (FIB). Silicon dioxide is used as the IMD material for both chips, with Chip-B IMDs being slightly fluorinated (SiOF) (Supplementary Fig. S3). Fluorination is done for more modern processes to reduce the dielectric constant of the IMD layers (from $k$~4.0 for $SiO_X$ to ~3.5 in the fluorinated case)[41,52].

### Electrical performance (PBS solution at 67 °C)

Figure 3 gives a representative set of electrical results for ICs aged in the accelerated in vitro study in PBS solution at 67 °C. Figure 3a presents the electrical performance results at three different time points for IDC test structures with and without top metal shield (Chip-B). During the 12-month aging, the structures were continuously biased while in soak (5 V DC). EIS results (shown as Bode plots) demonstrate stable capacitive characteristics (phase ~ −90°) and high impedance values (|Z|> $10^{11}$ Ω at 0.01 Hz) throughout the 12-month accelerated aging for n = 4 samples from each test structure. Results for other tested IDC structures, from both Chip-A and Chip-B and biased at 15 V DC, also showed similar stable capacitive behavior over the 12-month accelerated study (Supplementary Fig. S8).

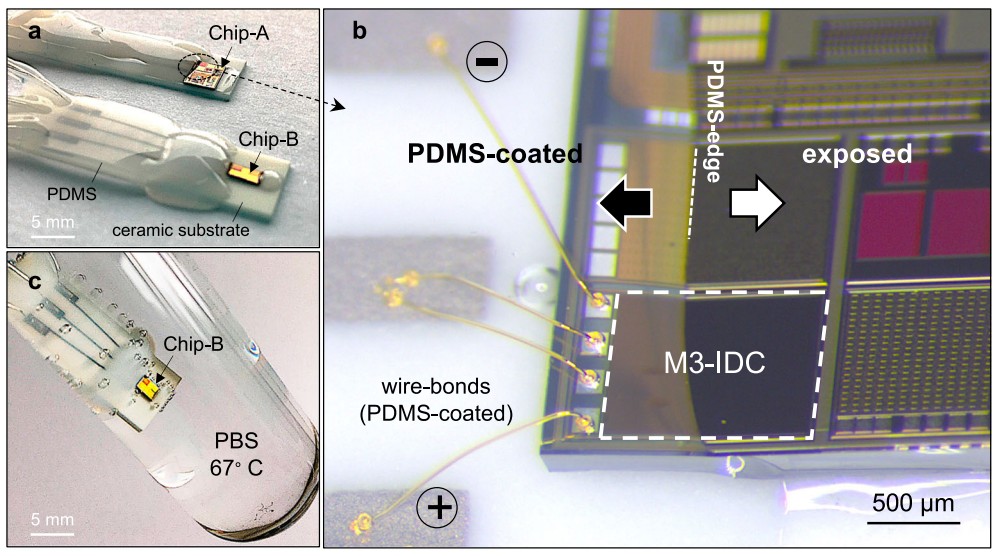

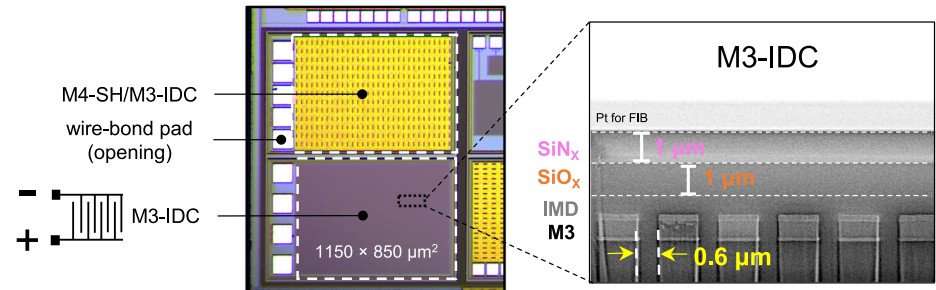

**Chip-A**: 4-metal, 0.35 μm CMOS process (4 × 5 mm²)

**Chip-B**: 6-metal, 0.18 μm CMOS process (1.7 × 3.5 mm²)

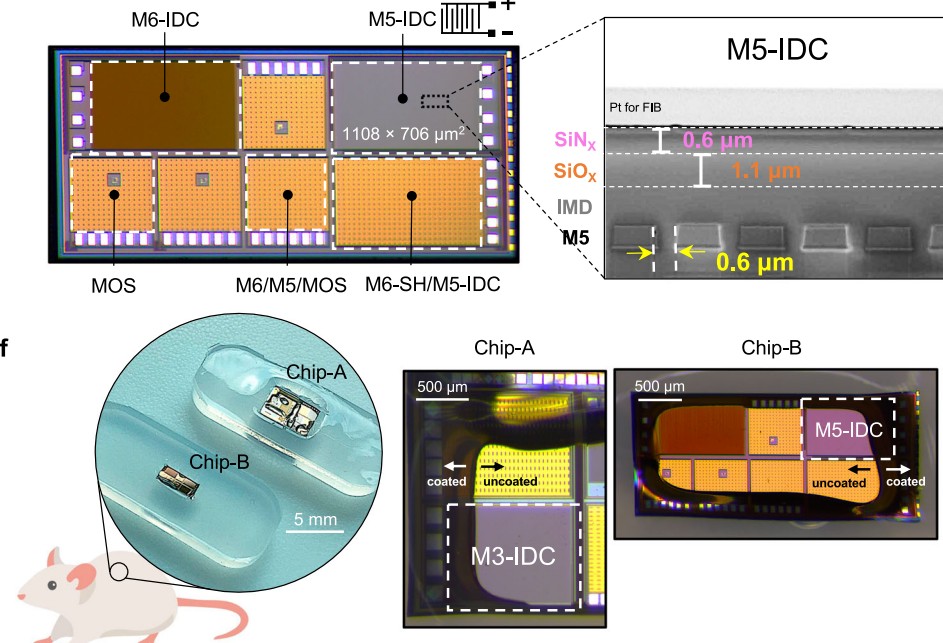

These results indicate three important characteristics of the IC materials and the applied interfaces. Firstly, the stable capacitive behavior (phase ~ −90°) indicates that the passivation and IMD ceramic layers contain no nano-defects or pinholes. Secondly, they show the electrical stability of the IC passivation and IMD layers surrounding the IDC metals when fully submerged in PBS solution while being continuously subjected to a maximum electric field of 0.25 MV/cm (at 15 V DC). In total, $n = 34$ ICs were tested for up to 12 months with $n = 17$ being continuously biased with 5 V and $n = 17$ with 15 V DC. Thirdly, regarding the PDMS coating, these results demonstrate that PDMS effectively prevents water leakage paths by forming long-lasting interfacial adhesion to the IC's nitride passivation (see

**Fig. 2 | Silicon-IC test structures prepared for long-term accelerated in vitro and in vivo aging. a–c** ICs wire-bonded and prepared for in vitro accelerated aging (voltage and temperature) in phosphate-buffered saline (PBS, pH - 7.4) at 67 °C, **a** Optical image of a Chip-A and B IC placed on ceramic substrates, wire-bonded and partially PDMS-coated, **b** Tilted optical micrograph of a representative Chip-A sample with the M3-IDC test structure wire-bonded and locally PDMS-coated, leaving most of the chip surface and sidewalls exposed, **c** Optical image of a partially PDMS-coated Chip-B fully immersed in PBS solution at 67 °C. **d, e** Top view optical micrographs and cross-sectional SEM images of Chip-A and B ICs with **d** Optical micrograph of a Chip-A IC fabricated in a 0.35 μm 4-metal CMOS process

with two IDC test structures (left). Cross-sectional SEM image of the M3-IDC test structure showing the interdigitated metal structures and the top $SiN_x$/$SiO_x$ passivation layers (right). **e** Optical micrograph of Chip-B showing various IDC and MOS transistor test structures fabricated in a 0.18 μm 6-metal CMOS process (left). Cross-sectional SEM image of M5-IDC test structure showing Metal-5 and the top $SiN_x$/$SiO_x$ passivation layers (right). **f** ICs subcutaneously implanted in rats. Chip-A and Chip-B ICs are positioned on soft PDMS substrates and locally coated with PDMS, covering the aluminum pads and regions of the top passivation while leaving most of the IC surface exposed to the body. Note that ICs used for the animal study are not wire-bonded.

Supplementary Note 1 and Supplementary Table S4). In addition, they attest to the stable electrical insulation properties of PDMS while being electrically biased and submerged in PBS solution.

Some of our IDC samples (13 out of 56) showed EIS irregularities (See Supplementary Table S6), which were attributed to either, 1) wire-bond corrosion ($n = 5$) (See Supplementary Figs. S9 and S10), 2) stress-induced cracking of the passivation layers, or 3) poor conformality of the passivation layers ($n = 9$) (Supplementary Note 4). Notably, the latter two failure scenarios on both IC foundries were found to be influenced by the use of the top metallization[40,53] (Supplementary Figs. S11b and S12). The poor conformality or cracks expose the IDC metals to the PBS solution and when electrically biased, result in severe degradation and short-term failures. All other IDC structures implemented with lower metallization showed no structural damage or EIS irregularities (M3 in Chip-A and M5 in Chip-B, see Supplementary Figs. S11a and S13).

NMOS transistors on Chip-B, both with ($n = 1$) and without ($n = 4$) the double shielded metal barrier, showed stable $V_{GS}$-$I_{DS}$ characteristics over the 12-month accelerated aging (Fig. 3b). For the NMOS structures without the shield metal barriers, the passivation and IMD stack serve as the protective barriers between the ionic solution and the MOS structure.

Figure 3c presents the results for the exposed dielectric sensor (Chip-B) after 5-months of aging in PBS solution at 67 °C. Comparing the results at 0-month and 5-month shows minimal change in the average and standard deviation results for the entire array. Testing on this chip stopped due to an intermittent wire-bond connection failure. The results, however, demonstrate the stability of the dielectric materials on Chip-B (having fluorinated silicon dioxide IMD). More significantly, these results demonstrate the stable electrical functionality of a complex CMOS IC designed in a 0.18 μm CMOS process when submerged as a bare chip in a 67 °C PBS solution.

**Material performance: exposed vs. PDMS-coated**
In the previous section, the electrical stability of various IC test structures was demonstrated over a 12-month accelerated in vitro aging study in PBS solution at 67 °C. Despite the electrical stability, exposure to ionic fluids may still degrade the IC's surface materials without causing discernible changes in the electrical characteristics. Given that identification of the degradation pathways can aid in the longevity estimation of the ICs, as a next step, we analyzed the materials on both the exposed (bare die) and PDMS-coated regions of the aged ICs. Our investigation began by using cross-sectional SEM imaging to evaluate the stability of the IC's entire multilayer stack. We were particularly evaluating the stack for any instances of interlayer delamination or intralayer degradation. Next, we examined the chemical stability and barrier properties of the IC's $SiN_x$/$SiO_x$ passivation layers using ToF-SIMS and XPS depth profiling. During the material evaluations, a comparative analysis was also conducted to understand the impact of the two aging environments on degradation: PBS solution at 67 °C and the in vivo animal environment at 37 °C. Finally, we investigated the effect of long-term exposure to electrical fields by focusing on a subset of IDC structures continuously subjected to electrical biasing (5 V and 15 V) in PBS solution at 67 °C for 12 months.

**Stability of the IC structures (unbiased).** The stability of the IC's multilayer stack was first evaluated using optical microscopy and later by SEM, examining the surface and various cross-sections across the chip (corner, center, and PDMS-edge). Figure 4a–c shows representative cross-sectional SEM images taken at the PDMS-edge for two M3-IDC test structures (from Chip-A) at two different time points, 3-month and 10-month in PBS solution at 67 °C. SEM results show intact metallization and no sign of interlayer delamination in the IC stack. Higher magnification imaging, however, reveals a thinning of the top $SiN_x$ passivation in the exposed region. Note that despite the loss of the $SiN_x$ layer, EIS results for the M3-IDC test structures on Chip-A remained stable for at least 12 months of aging in PBS solution. The PDMS-coated regions show no sign of nitride loss. Similar results were found for the explanted ICs demonstrating that even thin layers of PDMS (<1 μm), despite its moisture-permeability, can prevent nitride dissolution by maintaining a strong interfacial adhesion and effectively blocking tissue contact with the film (see Supplementary Figs. S14–S16 and Supplementary Note 5).

Figure 4d compares the dissolution rates of $SiN_x$ passivation after exposure to various aging environments, de-ionized (DI) water (done as Supplementary experiments as given in Supplementary Table S3) and PBS both at 67 °C, and rat at 37 °C. The higher dissolution rate in vivo at 37 °C illustrates the body's more aggressive environment and suggests that the dissolution is not entirely hydrolytic but that the ions, proteins and other organic species in the body can impact the process. Dissolution of thin-film ceramic materials has been reported in previous literature[36,54]. Our nanometer surface analysis using ToF-SIMS, however, revealed subtle differences in the 0–5 nm of the surface suggesting that the different dissolution rates could be due to the type of oxide layer created on top of the $SiN_x$ upon its exposure to various media (Supplementary Note 6).

Comparing the $SiN_x$ dissolution rates on Chip-A and B shows a higher dissolution rate for Chip-A material despite the similar N/Si for both chips (Chip-A and B both having a N: 50% and Si: 50% as given in Supplementary Fig. S7). This data highlights how the details in the manufacturing processes could impact the biostability of these layers. The higher rate for Chip-A could be due to the slightly higher hydrogen (H) content in the layer (~20% for Chip-A and ~15% for Chip-B)[37,55]. Other factors, such as the morphology (atomic arrangements) could also play a role in the chemical stability when exposed to wet environments[39].

Despite the gradual dissolution of the $SiN_x$ passivation, no adverse inflammation or tissue reaction was observed for all the animal models throughout the 12-month implantation, suggesting the biocompatibility of the ICs (See Supplementary Figs. S19, S20).

A group of Chip-A ($n = 10$) and Chip-B ($n = 8$) ICs were aged for longer than 12 months in PBS at 67 °C. After 12 months, optical microscopy on all these Chip-A ICs showed gradual signs of delamination near the sidewalls of the chip. SEM investigations revealed the delamination to be between the silicon substrate and the entire IC stack (See Supplementary Fig. S21). The delamination, however, was only observed on the side walls directly exposed to the PBS solution with the PDMS-protected regions remaining intact. Chip-B samples

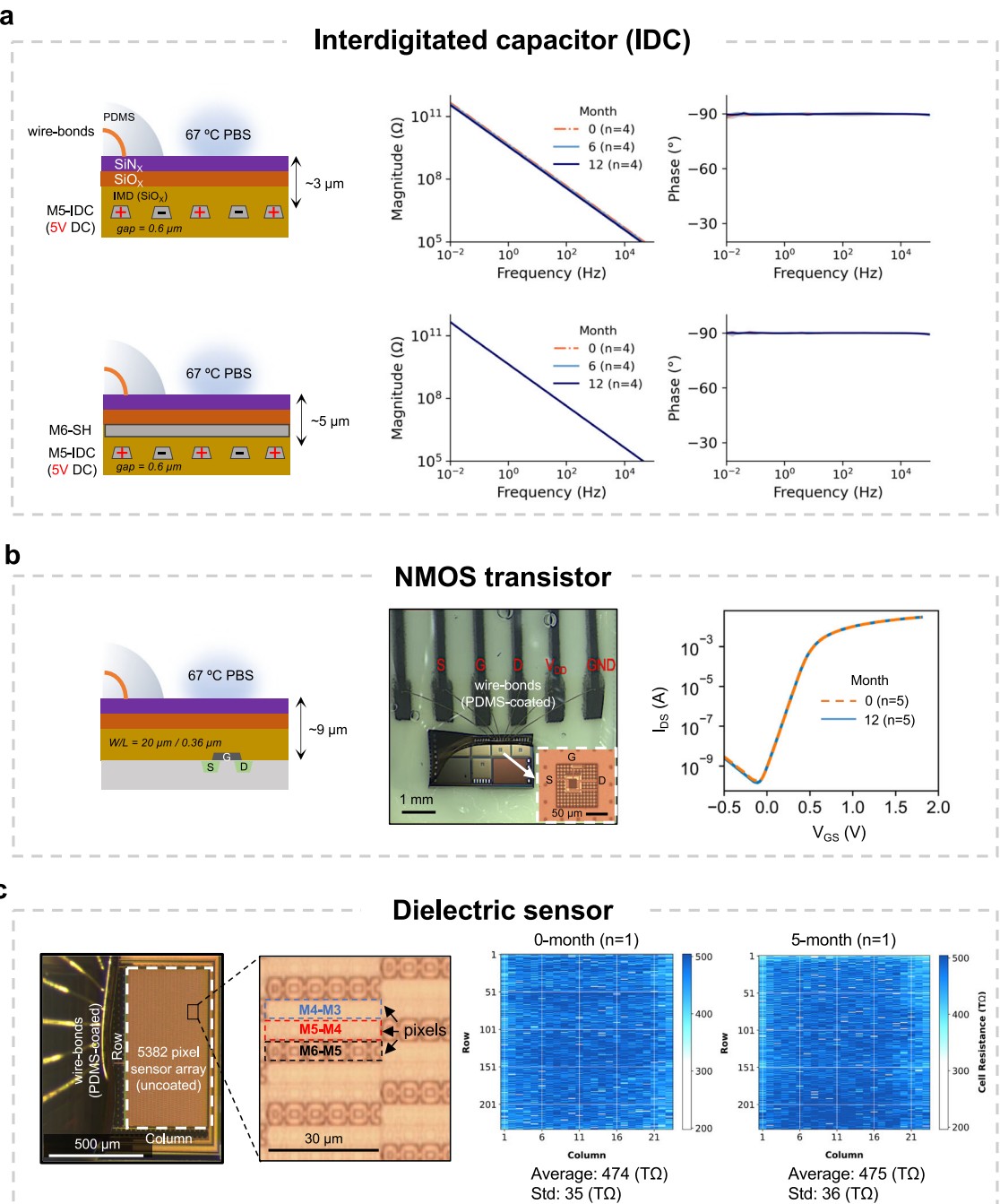

**Fig. 3 | Long-term electrical performance of test structures on Chip-B during accelerated aging in PBS solution at 67 °C. a** Electrochemical impedance spectroscopy (EIS) results of representative M5-IDC test structures without (top, $n = 4$) and with a top-metal (M6) shield (bottom, $n = 4$), showing stable capacitive characteristics over the duration of aging. Results are given as average and standard deviation ($n = 4$). **b** Top view optical micrograph and average $V_{GS}$-$I_{DS}$ transfer characteristics of NMOS transistors at 0-month and 12-month ($n = 4$ unshielded and $n = 1$ shielded MOS structures). Schematics show the distance between the test structures and the surface of the IC which is exposed to PBS solution (dimensions not to scale). **c** Top view optical micrograph of the dielectric sensor with a high magnification image of the pixels (left). Measurement results giving the value of each pixel within the array (total 5382 pixels) at 0-month and 5-month ($n = 1$). Comparing the average and standard deviation of the entire array at 0-month and 5-month show minimal change (right). All electrical measurements are performed while the ICs and their PDMS-coated wire bonds are fully submerged in PBS solution.

aged for longer than 12 months showed no signs of delamination (Supplementary Fig. S22).

**Ionic barrier performance of IC passivation layers (unbiased).** Next, we investigated ionic barrier properties of the $SiN_X/SiO_X$ passivation after their long-term exposure to physiological media.

The ingress of various cations ($Na^+$, $K^+$, $Ca^{+}$ and $Mg^+$) and anions ($Cl^-$, $PO_4^-$, $S^-$) in the exposed regions of the ICs was measured by positive and negative mode ToF-SIMS depth profiling, respectively. Figure 5 presents the positive mode depth profiles of Chip-A and B ICs, explanted after 7 and 12 months in vivo. For the 7-month explanted ICs, ~100 nm and ~200 nm of the $SiN_X$ passivation remained on Chip-A and Chip-B, respectively, supporting the dissolution rates given in Fig. 4. Despite the dissolution and long-term exposure, no ingress of alkali metals is present in the remaining layers, indicating that the $SiN_X$ passivation from both IC foundries

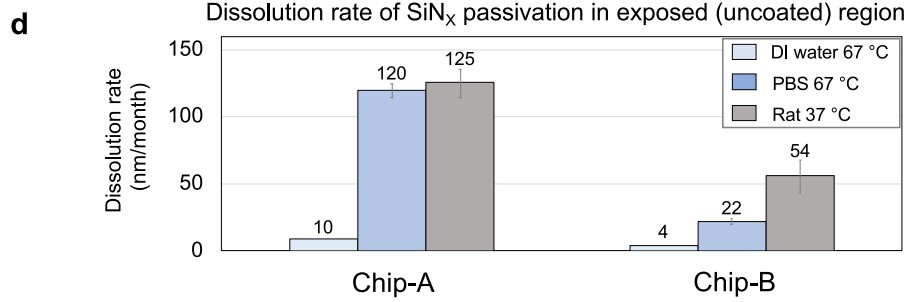

**Fig. 4 | Stability of IC multilayer stack and top passivation layers after accelerated in vitro and in vivo aging. a–c** Representative cross-sectional SEM images of Chip-A samples after accelerated in vitro aging in PBS solution at 67 °C, **a** Cross-sectional SEM image on the PDMS edge of an M3-IDC (Chip-A) structure after 3-months of aging in 67 °C PBS showing the IMD material stack from surface to silicon substrate. Inset: optical micrograph of the area used for cross-section analysis. Right, EDX elemental analysis shows intact aluminum IDC metallization. **b** Magnified cross-sectional SEM image at the PDMS-edge showing a thinning of exposed $SiN_X$ passivation ( ~ 360 nm) after 3-months of aging in 67 °C PBS. **c** Cross-sectional SEM image of a Chip-A sample after 10 months with complete loss of $SiN_X$ passivation in the uncoated region, leaving the $SiO_X$ passivation exposed to the PBS solution. **d** Dissolution rates of exposed $SiN_X$ passivation from Chip-A and B in different aging media (data is presented from $n = 2$ samples, as mean ± std, with each sample measured on two different locations). Individual data points can be found in Supplementary Fig. S18.

maintained its ionic barrier properties over the long-term exposure in vivo.

On the 12-month explanted ICs, no $SiN_X$ was detected on either chip, leaving the $SiO_X$ passivation directly exposed to the body environment. Based on the estimated $SiN_X$ dissolution rates in Fig. 4, after the complete $SiN_X$ dissolution, the $SiO_X$ passivation is exposed to tissue for approximately 4 and 2 months for Chip-A and Chip-B samples, respectively. Interestingly, no ionic ingress was detected for the exposed $SiO_X$ layers from either IC foundry. Negative mode depth

profiles also did not reveal any ingress of anions and do not contain additional information; thus, they are not presented.

These results indicate that both $SiN_X$ and $SiO_X$ passivation layers from the two selected silicon-IC foundries are effective long-term barriers against ionic ingress from the body. Note that in some profiles, a slight increase in the aluminum (Al) intensity was found in the deeper depths of the $SiO_X$ as the profile was getting closer to the IDC metallization. Closer to the surface or in the $SiN_X$, however, no aluminum was detected, indicating that the passivation layers act as a two-sided

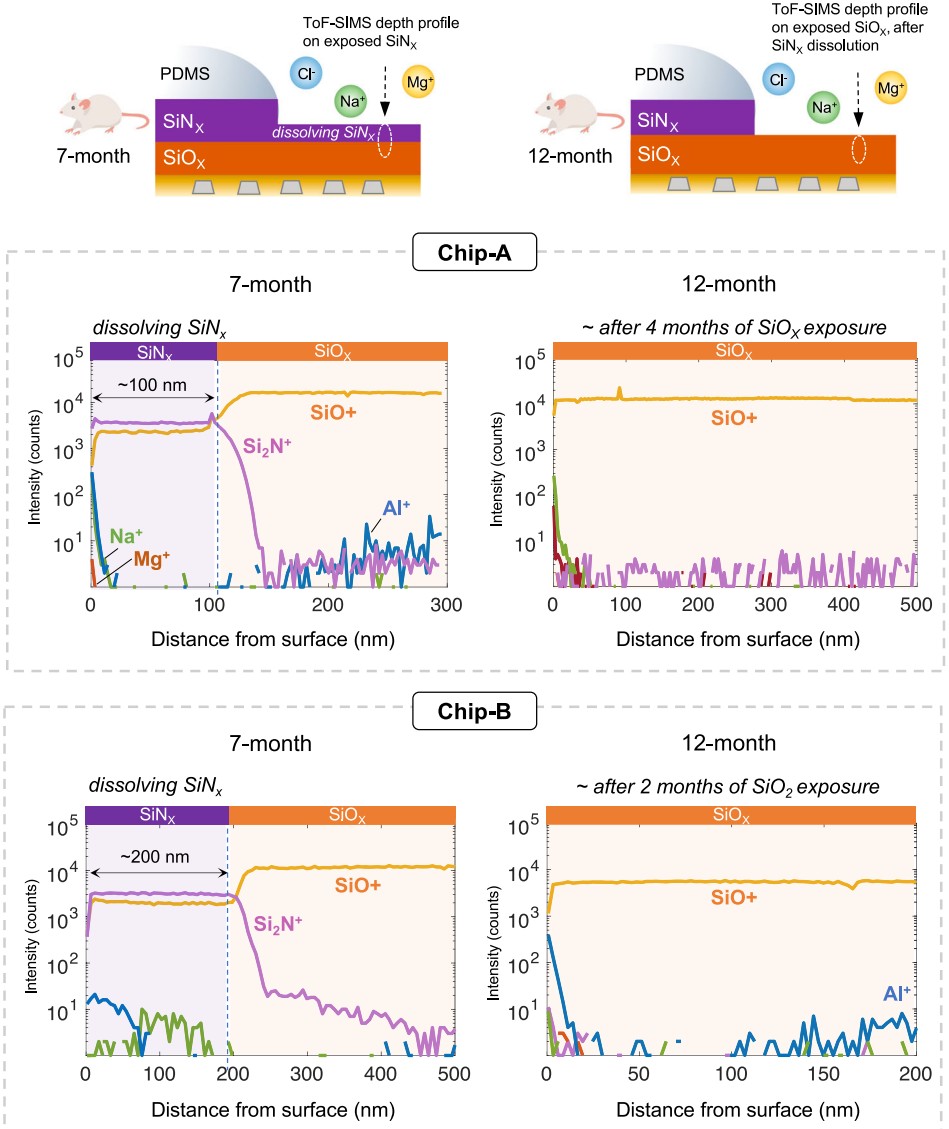

**Fig. 5 | Positive mode ToF-SIMS depth profiles analyzing the ionic barrier performance of the exposed passivation layers after 7 and 12 months implantation in rat.** After 7 months, approximately 100 nm and 200 nm of the dissolving $SiN_x$ remained on Chip-A and B samples, respectively. Despite the dissolution, no detectable ionic penetration is found within the remaining $SiN_x$ layers. After 12 months, with the complete dissolution of $SiN_x$, the $SiO_x$ passivation is directly exposed for ~4 and 2 months for Chip-A and B, respectively. Nevertheless, no signs of ionic penetration are detected within the $SiO_x$ passivation layers.

barrier, preventing ionic ingress into the chip, but also inhibiting the out-diffusion of IC metals into the body.

**Chemical stability and moisture barrier performance of IC passivation (unbiased).** Next, we investigated the chemical stability and moisture barrier properties of the $SiN_x$ and $SiO_x$ passivation layers on the directly exposed and PDMS-coated regions of the aged ICs.

For this purpose, we used negative mode ToF-SIMS depth profiling to evaluate the chemical stability of the $SiN_x$ and $SiO_x$ passivation layers. Profiling was done at three different depths: $SiN_x$-surface (analyzed from 0–15 nm), $SiN_x$-bulk (analyzed from 50–100 nm within the $SiN_x$ layer), and $SiO_x$-bulk (analyzed from 100 to 200 nm within the $SiO_x$ passivation layer). Surface analysis (0–15 nm) was conducted with sub-nanometer (0.139 nm) step sizes (See Methods). All bulk profiles were collected using ~1 nm step sizes.

The schematic in Fig. 6 summarizes the findings from the depth profile analysis. In the exposed regions, a thickness increase in the surface oxidation of the $SiN_x$ passivation is observed. Within the oxidized layer, a slight increase in ionic impurities is also seen. In the

PDMS-coated region, on the other hand, only an increase in oxidation thickness was found with no increase in the ionic impurities.

Figure 6a, b presents the negative mode depth profiles of a representative Chip-A sample after 7 months of implantation in rats, comparing the first 15 nm of the $SiN_x$ passivation in the PDMS-coated and exposed regions. In both regions, the first 5 nm demonstrated low intensities of $[SiN^-]$ with high intensities of $[OH^-]$ and $[SiO_2^-]$. This pattern indicates a surface oxidation of the $SiN_x$, revealing an additional degradation mechanism affecting the SiNx passivation on the ICs. In the exposed region, a slightly higher chlorine $[Cl^-]$ and sulfur $[S^-]$ intensity was also detected in the oxidized layer. These ions could be a result of biofluids and amino acids touching the oxidized layer and penetrating within the layer[56]. In the PDMS-coated region, on the other hand, the oxidized $SiN_x$ layer showed lower $[Cl^-]$ and $[S^-]$ intensities, similar to reference level ICs. Similar surface ToF-SIMS depth profile results for a 7-month explanted Chip-B can be found in Supplementary Fig. S23.

Figure 6c compares the thicknesses of the oxidized $SiN_x$ layer in the exposed and PDMS-coated regions for Chip-A and B after 7 months

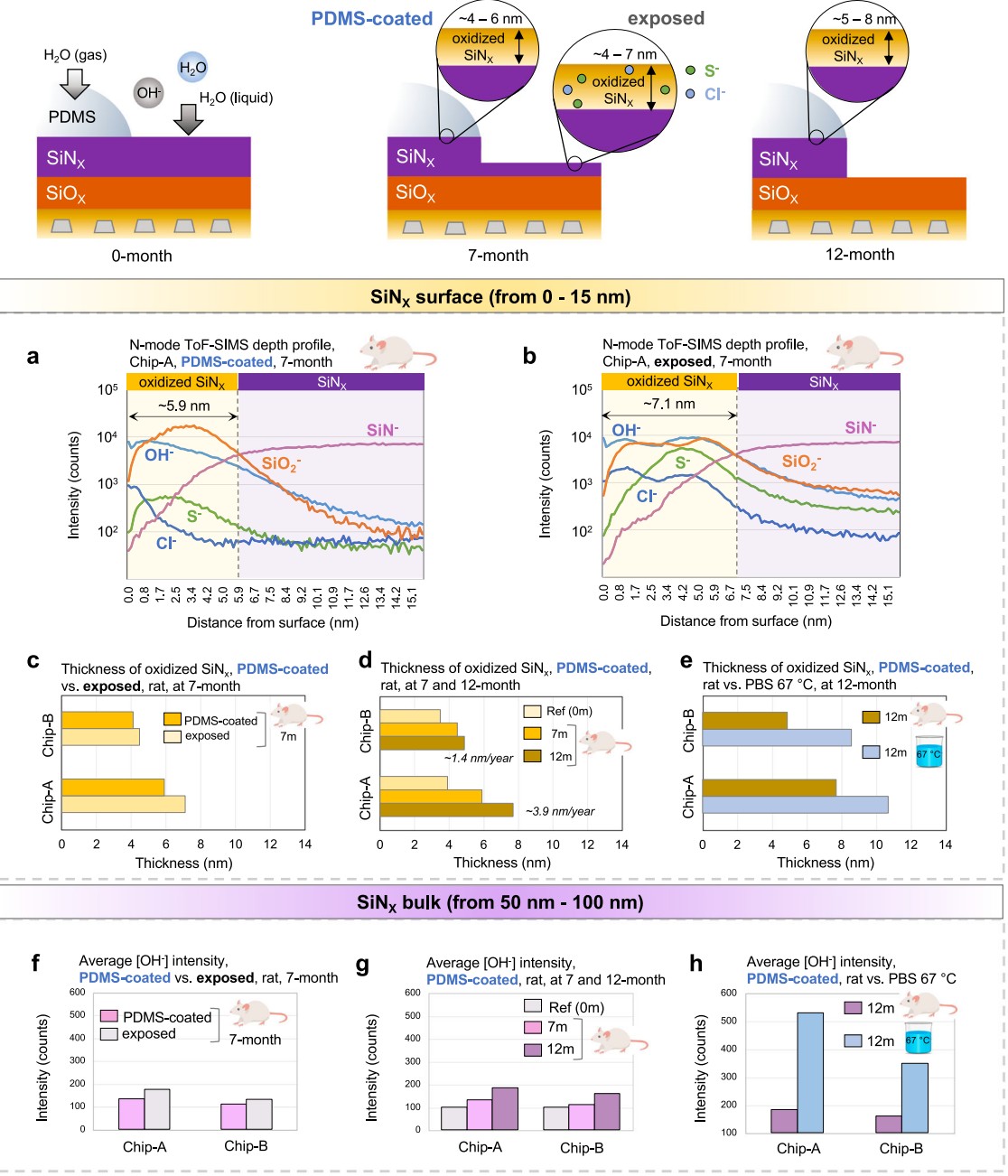

**Fig. 6 | Chemical stability and moisture barrier performance of PDMS-coated and exposed passivation layers after accelerated in vitro (unbiased) and in vivo aging. a, b** Representative negative (N) mode ToF-SIMS surface depth profiles from 0–15 nm of the PDMS-coated and exposed regions of a 7-month explanted Chip-A sample (acquired with 0.1 nm step-size). Surface depth profiles show oxidation of the SiN_X passivation with higher chlorine (Cl) and sulfur (S) impurity ions in the oxidized layer for the exposed region. **c–e** Comparing the thicknesses of the oxidized SiN_X at different time points and aging media (in 7 m or 12 m, 'm' denotes month). **c** Comparing the thicknesses in the PDMS-coated and exposed regions after 7 months in vivo, **d** Comparing the thicknesses in the PDMS-coated regions after 7 and 12 months in vivo with that of a dry reference sample, and **e** Comparing the thicknesses in the PDMS-coated regions after 12 months exposure to accelerated in vitro and in vivo environment. **f–h** Average [OH⁻] intensities within the SiN_X passivation bulk (averaged ToF-SIMS depth profile data from 50–100 nm). Data is presented as average of 4 measurement results, performed on $n = 2$ samples (per chip foundry) with 2 measurements per sample.

in vivo. The oxidized layer on Chip-B, for both regions, shows a thinner layer compared to Chip-A, suggesting the higher density and stability of the nitride passivation from this foundry. Figure 6d shows the thickness of the oxidized SiN_X in the PDMS-coated regions over time. After 12 months of implantation in rats, ~7.8 nm (0.78% of total SiN_X thickness) and ~4.9 nm (0.75% of total SiN_X thickness) of the nitride passivation has been oxidized for the Chip-A and B ICs, respectively. Note that for reference ICs (as supplied by the foundry, 0-month), a ~3.5–4 nm oxide layer is already present on the surface of the SiN_X

passivation. Assuming a linear oxidation rate, a ~3.9 nm/year and ~1.4 nm/year oxidation rate can be anticipated for PDMS-protected Chip-A and B ICs when implanted in the body.

Figure 6e compares the effect of the two aging environments (PBS solution at 67 °C and rat at 37 °C) on the oxidation of the nitride passivation in the PDMS-coated region. After 12 months, ICs soaked in 67 °C PBS solution showed a thicker oxidized layer on the SiN_X. These results indicate that the 67 °C accelerates the moisture diffusion and oxidation process within the SiN_X passivation layer.

Within the $SiN_X$-bulk, the $[OH^-]$ and $[SiN^-]$ signals were evaluated by averaging the intensities within the 50 – 100 nm depth of the $SiN_X$ layer. Figure 6f presents the $[OH^-]$ intensities within the $SiN_X$-bulk for the exposed and PDMS-coated regions after 7 months of implantation in rat, showing slightly higher $[OH^-]$ in the exposed regions. In the PDMS-coated regions, the $[OH^-]$ intensity was also evaluated over time (Fig. 6g), indicating a gradual increase in the intensity with time. When comparing the two aging environments at 12-month, a marked increase in the $[OH^-]$ intensity is observed for the ICs soaked in PBS solution at 67 °C (Fig. 6h), possibly resulting from the higher diffusion rate of moisture with temperature.

The $[SiN^-]$ intensity in the bulk region showed stable intensities for all samples except those soaked in PBS solution at 67 °C, where a slight decrease was observed (Supplementary Note 7). This decrease suggests that elevated temperatures accelerate moisture diffusion and promote the breakage of Si-N bonds within the $SiN_X$ passivation layer. The $SiO_X$ passivation was similarly investigated, showing the film's high stability and moisture barrier properties (Supplementary Note 7).

**Chemical stability, moisture, and ionic barrier performance of IC passivation under continuous electrical bias in PBS solution at 67 °C.** Given that implantable ICs will be subjected to electrical bias voltages during their lifetime operation, as our final investigation, we evaluated the long-term effect of electric biasing on the stability of the IC materials and the PDMS-interfaces.

Figure 7a gives a schematic representation of a partially PDMS-coated IC applied to an electrical DC bias (between the IDC metallization and solution). During biasing, both combs of the IDC were connected to the negative (−) potential and +5 V or +15 V DC was connected to a stainless-steel rod in the PBS solution. The polarity was chosen to drive the alkaline ions ($Na^+$ and $K^+$) towards the IC passivation. The right schematic in Fig. 7a gives an overview of the findings demonstrating the effect of the continuous biasing on the exposed and PDMS-coated regions of the chip.

Negative and positive mode ToF-SIMS depth profiling was used to evaluate the chemical stability and barrier properties of the exposed and PDMS-coated regions of the biased ICs. For analysis, M3-IDC ($n = 2$, Chip-A) and M5-IDC ($n = 2$, Chip-B) test structures were used which were continuously biased for 12 months with 15 V DC in PBS solution at 67 °C. Figure 7b presents the depth profiling results for the exposed region on a representative M3-IDC structure (Chip-A). Negative mode depth profile results revealed a ~120 nm oxidized layer on top of a 100 nm $SiN_X$ layer. Within the oxide layer, high intensities of $[SiO_2^-]$, $[OH^-]$, and carbon $[C^-]$ were found. Positive mode depth profiles also revealed high intensities of $[Na^+]$ and $[K^+]$ ions in the top ~120 nm oxidized layer. Beyond the 120 nm depth, once the profile enters the remaining ~100 nm $SiN_X$ layer, a marked drop in the impurity ions was observed, demonstrating that the remaining $SiN_X$ still acts as an ion barrier.

XPS depth profiling was used to quantify the chemical composition and ionic impurity levels within the oxidized layers (See Fig. S24, Supporting Information). The chemical composition of the oxidized $SiN_X$ layer indicated a slightly off-stoichiometric oxide film for both Chip-A (Si: 28%, O: 58%) and Chip-B (Si: 28%, O: 62%) samples. For the ionic impurities, despite the high intensities detected by ToF-SIMS, XPS found a 2–3% sodium (Na) incorporation in the oxidized layer for both Chip-A and B samples. Note that the sensitivity levels of the ToF-SIMS measurement to $Na^+$ and $K^+$ ions can be 3 to 5 orders of magnitude higher than XPS (See Methods).

In comparison to the exposed region, depth profiling on the PDMS-coated region of the biased IDC structures showed a stable $[SiN^-]$ profile within the entire ~1 μm $SiN_X$ passivation layer (Fig. 7c). A thin (~12 nm) oxide layer was observed on top of the $SiN_X$ layer with no ionic penetration. The absence of ionic penetration in the oxidized

layer is due to the ionic barrier properties of the PDMS coating[57], in this case, even in the presence of an electrical field.

In the PDMS-coated regions, to examine if the applied electrical fields affected the oxidation rate of $SiN_X$, the thicknesses of the top oxidized layer were measured for a group of samples biased for 12 months with a 5 V ($n = 2$) or 15 V ($n = 2$) DC voltage (between IDC and PBS at 67 °C) (Fig. 7d). At 5 V DC ($n = 2$ from each chip), no significant change in oxide thickness was measured compared to unbiased IDC structures. However, at 15 V DC ($n = 2$ from each chip), both Chip-A and Chip-B samples showed a slightly thicker oxidized layer, suggesting that higher electrical fields may enhance the oxidation rate of $SiN_X$ in the PDMS-coated region.

Figure 7e, f gives the average $[OH^-]$ intensity in the bulk (averaging the intensities between 100−200 nm) of the $SiN_X$ and $SiO_X$ passivation layers. In the $SiN_X$ layer, higher $[OH^-]$ intensities were observed on samples subjected to continuous 15 V DC voltages as compared to 5 V and unbiased chips (Fig. 7e). The $[OH^-]$ incorporation may result from the higher applied voltage, as higher electrical fields may introduce strain in the molecular structure of the dielectric[58], making the structure more susceptible to moisture attack. It should be noted that the increased $[OH^-]$ incorporation in the $SiN_X$ layer was exclusively detected using ToF-SIMS due to its enhanced sensitivity. XPS depth profiles revealed no change or oxygen incorporation in the $SiN_X$ passivation for both Chip-A and Chip-B samples (Supplementary Fig. S24). The $SiO_X$ passivation was similarly investigated (Fig. 7f), where a slightly higher $[OH^-]$ intensity was observed for devices under 15 V continuous DC biasing.

## Discussion and guidelines for longevity

Miniaturizing implantable devices necessitates novel packaging materials and processes. PDMS, despite its many favorable properties, has rarely been used as a long-term standalone microelectronic packaging material due to its high moisture permeability. To address the moisture issue, in recent years various advanced thin-film barrier materials and coatings have been proposed[18–21,59]. However, packaging that depends on highly specific materials and processes available only to a few academic labs will not easily find its path to translation. For that, standardized materials and processes accessible to all must be used. To this end, in this work we have consciously chosen to rely on the barrier performance of commercially fabricated ICs and use PDMS as an easily accessible standalone packaging material, despite its non-hermetic properties. For a successful PDMS packaging, all components and interfaces underneath must be moisture-resistant with good adhesion to the polymer[48]. Our long-term electrical and material evaluations suggest that IC structures can be inherently hermetic themselves, preventing the outer moisture and ions reaching the sensitive structures within. This is likely due to the tightly controlled manufacturing processes and high-quality raw materials utilized by the semiconductor foundries. Additionally, the large sample size tested here ($n = 34$ electrically biased and tested for up to 12 months) and the reproducibility of the results underscore the consistency and reliability of the IC manufacturing process which has always been the hallmark of the semiconductor industry.

By comparing the exposed bare-die and PDMS-coated regions of the IC structures, we demonstrated how PDMS can effectively eliminate or retard the observed degradations seen on the exposed regions of the die. Considering the oxidation of the $SiN_X$ to be one of the main degradation paths and taking a conservative estimate, we believe PDMS-coated ICs have the potential to reach decades of functional lifetime in the body (based on the observed oxidation rates of ~3.9 and 1.4 nm/year for Chip-A and B, respectively).

Besides die hermeticity, as explained in Section "PDMS as a soft but moisture-permeable packaging material" and Supplementary Note 1, a successful PDMS packaging also requires strong and long-lasting PDMS adhesion to the underlying materials. During

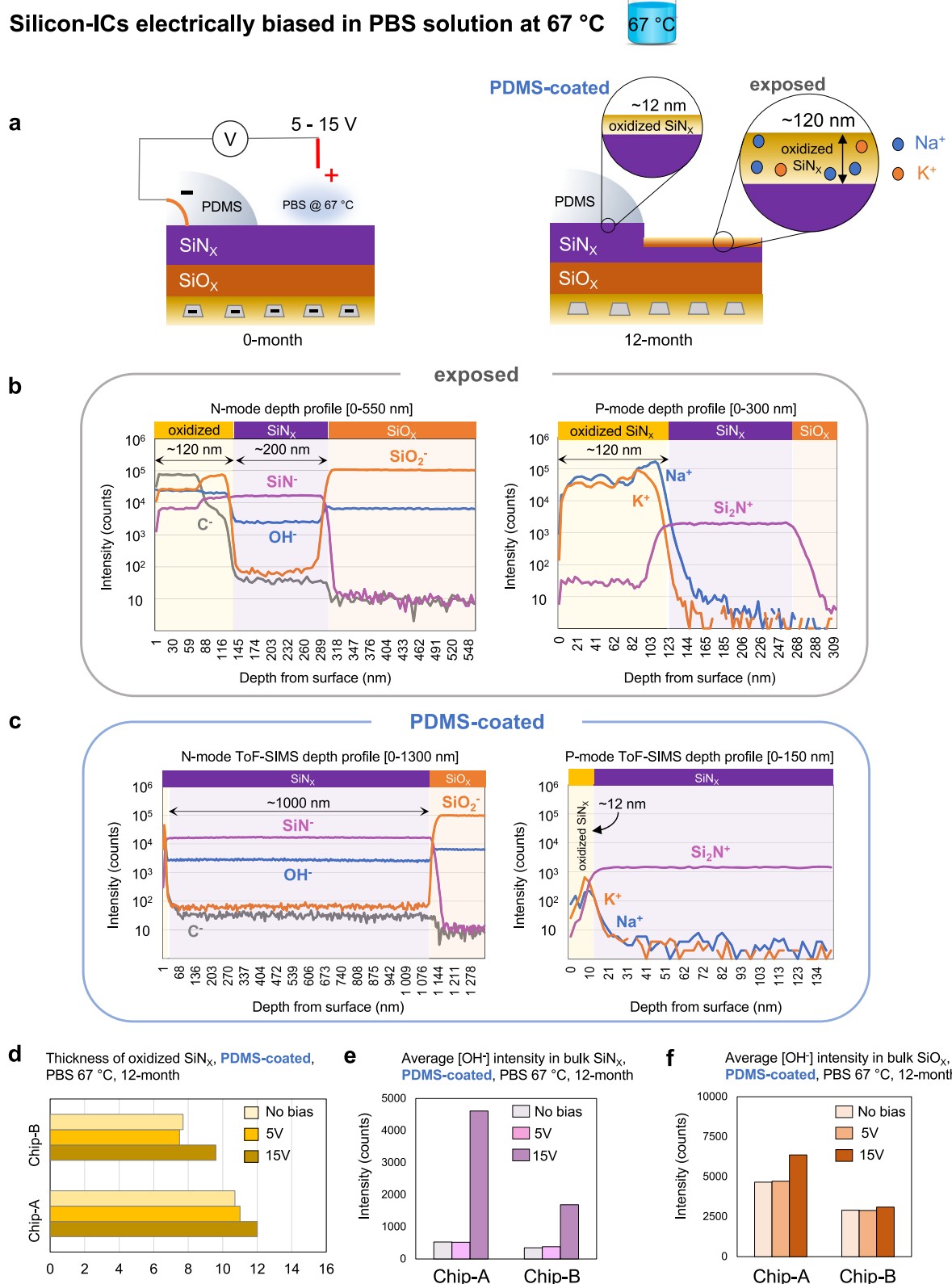

**d** Thickness of oxidized SiNₓ, **PDMS-coated**, PBS 67 °C, 12-month

**e** Average [OH⁻] intensity in bulk SiNₓ, **PDMS-coated**, PBS 67 °C, 12-month

**f** Average [OH⁻] intensity in bulk SiOₓ, **PDMS-coated**, PBS 67 °C, 12-month

implantation, various environmental factors, such as body fluids and mechanical stress, can degrade and weaken these interfacial bonds[60]. In this study, qualitative evaluations of PDMS adhesion on both accelerated in vitro and in vivo samples showed no separation between the PDMS and IC surface after aging (see Supplementary Note 8). It should be noted though that the mechanical stresses applied to each

implant may vary depending on its shape, size, and location of implantation[61,62]. Therefore, for a more accurate prediction of implant longevity, the mechanical stresses expected in real-use conditions should be modeled and replicated in the accelerated in vitro study.

For applications using bare ICs such as neural probes[15], we showed the dissolution of the SiNₓ passivation to be one of the primary

**Fig. 7 | Chemical stability and moisture and ion barrier performance of PDMS-coated and exposed passivation layers after continuous DC biasing in PBS solution at 67 °C. a** Left schematic: biasing configuration with DC voltage applied between the IDC metallization (combs) and an electrode in PBS solution. Right schematic: the effect of the continuous biasing on passivation. **b**, **c** Positive(P) and negative(N) mode ToF-SIMS depth profiles of the exposed and PDMS-coated regions showing thinning and oxidation of the $SiN_X$ passivation. In the exposed region, ion ingress in the top ~120 nm oxidized layer is detected. PDMS coated region shows a 12 nm oxide layer with no ion penetration. **d** Thickness of the oxidized $SiN_X$ surface for Chip-A and Chip-B ICs in the PDMS-coated region after continuous biasing at five and 15 V between IDC and PBS. **e**, **f** Average [OH⁻] intensities in the bulk of the $SiN_X$ and $SiO_X$ passivation layers (averaged from 100–200 nm within the layer) in the PDMS-coated region after continuous electrical biasing with 5 or 15 V DC. Data is presented as average of 4 measurement results, performed on $n = 2$ samples (per chip foundry) with 2 measurements per sample.

degradation pathways in vivo (dissolution rate of ~125 and ~54 nm/month for Chip-A and B, respectively). Despite the nitride dissolution, the $SiO_X$ passivation underneath also exhibited excellent ion and moisture barrier properties, suggesting that uncoated/unprotected ICs can maintain their functionality in the body for at least 12 months. Note that this is only when the top metallization is not used. When the top metal is used, due to the poor conformality of the PECVD passivation layers, in vivo (unbiased) results demonstrate that the top metal could be exposed to tissue within 6–7 months of implantation, causing metal corrosion (see cross-sectional SEM images in Supplementary Fig. S25). Besides device malfunction, this would pose toxicity and safety risks, as it would expose the body to electrical DC voltages. In fact, our DC biased in vitro studies showed the detrimental effects of DC signals when applied to top metal IDC structures (see Supplementary Figs. S11 and S12). Therefore, for applications requiring the use of the top metal layer of the IC (nominal or thick top metal), applying a PDMS coating or a thin conformal coating is advised[6,18].

In the accelerated in vitro study, corrosion of interconnect wire bonds was found to cause failure for some of the devices (5 out of in total 62 electrically tested ICs). Corrosion is believed to be driven by moisture and galvanic reactions between the gold wire bond and aluminum pads, causing the corrosion of the pad[63,64]. Non-bonded aluminum pads showed no signs of corrosion (Supplementary Fig. S9), mainly due to a native oxide layer on the aluminum, which has been reported to create strong interfacial adhesion bonds with PDMS[65]. As mentioned earlier, given PDMS's moisture permeability, all components and interfaces underneath should be moisture-resistant. In the case of wire-bond interconnects, dissimilar metals should be avoided. A potential mitigation strategy involves using aluminum wire bonds instead of gold[64], which could prevent galvanic reactions and concurrently result in stronger adhesion with PDMS.

For emerging implantable applications embedded with an IC, selecting a CMOS process suitable for chronic implantation requires extensive and long-term testing, often not within the scope of academic research groups. Therefore, we encourage researchers to use the technology nodes evaluated in this work when designing their ICs. It should be highlighted though that not all foundry-fabricated ICs will have the same level of hermeticity. In fact, in smaller CMOS technology nodes (90 nm and smaller), significant changes are made in the IC materials[41]. As an example for the IMD layers, porous dielectric materials such as silicon oxycarbide (SiOC) are replacing the conventional $SiO_X$ and have been reported to have stability issues in the presence of moisture[53,66,67]. Therefore, if smaller technology nodes are to be used, the top shield metal and extra wall-of-vias (WoV) around the IC may increase the chip's hermeticity and longevity by creating a metal cage around the entire IC structure. In addition, a simplified version of the dielectric sensor proposed in this study could be employed to detect early moisture ingress.

Stress-induced cracking during packaging or post-processing could undermine the hermeticity of the IC die[40]. Die thinning, for example, has been explored as a technique to enable die-embedded flexible bioelectronics[9,68,69]. This technique, however, may introduce stress and cracks in the passivation and dielectric layers. Moreover, as the silicon substrate becomes thinner, moisture penetration through this route may need further research. Therefore, as a future

investigation, and by incorporating the dielectric sensor proposed here, one can investigate how thin the ICs can be made without compromising the hermeticity of the die.

The effect of continuous electric fields on the stability of IC materials, both in the exposed and PDMS-coated regions, was investigated by analyzing a group of samples applied to DC bias voltages between the IDC metals and the PBS solution. In the exposed regions, the formation of a relatively thicker oxidized $SiN_X$ layer was found for 15 V biased ICs compared to unbiased samples. This thicker oxidized layer also appeared to slow the dissolution of $SiN_X$, whereas total layer loss occurred in unbiased devices after 12 months (Fig. 4c). In the PDMS-coated regions, enhanced [OH⁻] incorporation in the $SiN_X$ and $SiO_X$ passivation layers was recorded for the ICs subjected to 15 V DC in comparison to unbiased ICs. At 5 V DC, the typical range used for implantable ICs, the applied bias was found to have no long-term effect on the stability of the passivation layers. Based on these insights, for implantable ICs operating at higher voltages (>15 V DC), further investigations may be required to determine the long-term stability of the materials and interfaces. In our complementary investigation[70], the usefulness of these guidelines is validated by demonstrating ultra-long lifetimes for PDMS-coated ICs.

## Methods

### Fabrication of silicon-IC test structures

A key factor in the design of the test structures is the presence or absence of a metal "shield (SH)". The shield is a metal layer implemented using the top-metal of the process and designed to further protect the test structures underneath from any moisture/ion ingress. Additionally, around each individual test structure, a wall-of-via (WoV) is implemented which is a side metal barrier similar to the CMOS die seal ring.

Chip-A test structures were fabricated in a 0.35 μm CMOS 4-metal process. Two IDC test structures were implemented on Chip-A: (1) M3-IDC and (2) M4-SH/M3-IDC (Fig. 2d and Supplementary Table S1). M3-IDC (area: 1150 μm x 850 μm) contained 606 interdigitated fingers implemented in M3 (one layer below the top metal). It is protected by the IMD and the top $SiN_X/SiO_X$ passivation layers, each being ~1 μm thick, as presented in the cross-sectional scanning electron microscope (SEM) image in Fig. 2b. The adjacent M4-SH/M3-IDC test structure used a similar M3-IDC interdigitated structure but included a thick top metal shield (~2.8 μm thick) implemented in M4 (See cross-sectional SEM image in Supplementary Fig. S2).

Chip-B test structures were fabricated using a 0.18 μm 6-metal process and contained three types of structures: (1) IDCs, (2) MOS transistors, and (3) a dielectric sensor array (Supplementary Fig. 2e and Table S2). Supplementary Fig. S3 shows a representative WoV around the test structures on Chip-B and the material used by the Chip-B foundry. A similar WoV was also used around the two structures on Chip-A.

The IDC structures on Chip-B were implemented in Metal 5 (M5-IDC) or Metal 6 (M6-IDC). One IDC structure was also implemented with a shield on top (M6-SH/M5-IDC). In the M5-IDC structure, the comb metallization is in M5 and is protected by the IMD and the top $SiN_X/SiO_X$ passivation layers, each having a thickness of ~ 0.6 μm and ~ 1.1 μm, respectively (see cross sectional SEM image in Fig. 2e). M6-IDC

is implemented in the top metal layer of the process and has the comb metallization protected only by the $SiN_X/SiO_X$ passivation layers (Supplementary Fig. S4). M6-SH/M5-IDC has the comb metallization on M5 and a shield layer on M6 (-1 μm thick).

There are two MOS transistor structures with or without a shield barrier. All transistors have a W/L of 20 μm/0.36 μm. The shield barrier for the MOS structure is a double-layer metal implemented in M5 and M6.

The dielectric sensor was diced and separated from the other test structures on Chip-B to ease the wire bonding and sample preparation required for the long-term accelerated in vitro aging.

## Sample preparation and PDMS coating for the long-term accelerated in vitro aging study

For the accelerated in vitro study, bare dice were glued on ceramic adaptors with pre-printed Pt/Au tracks, gold wire-bonded, and then partially coated with PDMS (Fig. 2a).

The adaptors were soldered to Teflon-coated stainless-steel wires. Subsequently, the ICs were mounted and wire-bonded to the adaptor using 30 μm diameter gold wires. Before wire-bonding, samples were placed in a plasma chamber (RF power 400 W, 80% Argon, 20% Oxygen) for 180 seconds to surface activate the aluminum pads on the IC for optimal wire-bonding. To electrically insulate the metal tracks on the ceramic substrates and the connecting wire bonds, PDMS (DOW CORNING 3140, see Supplementary Table S4) was locally applied using a dispenser. For optimal PDMS adhesion, samples were surface-activated using UV-ozone for 15 minutes before PDMS coating. PDMS curing was done according to the manufacturer's guidelines. The ceramic adaptors were subsequently connected to bungs. Full details of the sample preparation can be found elsewhere[71].

## Accelerated in vitro aging study with continuous electrical biasing

For the accelerated in vitro aging study, a dedicated apparatus[72] was used (See Supplementary Fig. S5) where the IDC structures ($n = 23$ for Chip-A and $n = 35$ for Chip-B) were fully immersed in PBS solution at 67 °C while being continuously biased. This was done either by applying a DC electrical voltage between the combs of the IDC or between the combs and the PBS solution. For biasing, three voltages were used: unbiased ($n = 13$), 5 V DC ($n = 24$), and 15 V DC ($n = 21$) (Supplementary Fig. S5a-b). These voltages were selected based on their relevance for neurostimulator ICs[30]. Supplementary Table S3 provides an overview of all the samples, with their respective aging conditions, durations in test, and sample sizes (n) used for each analysis. The electrical performance of the IDCs was monitored monthly using electrochemical impedance spectroscopy (EIS) for a duration of at least 12 months. The NMOS test structures ($n = 5$) and dielectric sensor ($n = 1$) were immersed in PBS at 67 °C and monitored monthly while in soak for a 12-month duration (Fig. S5c, d, Supplementary Information). Every three months, samples were taken out, microscopically inspected, and placed back in fresh PBS solution.

## Sample preparation and PDMS coating for the long-term in vivo animal study

For animal studies, the chips were placed on 3 mm thick, soft PDMS substrates fabricated from medical-grade silicone rubber (MED2-4213, NuSil, see Supplementary Table S4). For manufacturing the PDMS substrates, a custom-made polytetrafluoroethylene (PTFE) mold was filled with uncured PDMS, which was then semi-cured in the oven at 100 °C for 30 minutes. At the same time, for optimal PDMS adhesion, the ICs were surface activated and cleaned using UV-ozone for 15 minutes with their top passivation facing the ozone lamp. The ICs were then placed in the center of the semi-cured PDMS substrates. After positioning, all aluminum pads and parts of the IC passivation were coated using the same medical grade PDMS (MED2-4213, NuSil).

Finally, the samples were fully cured in the oven at 100 °C for two hours.

## In vivo animal study

For the long-term in vivo study, $n = 12$ ICs ($n = 6$ from each IC foundry) were placed on PDMS substrates and implanted in 6 rats for a maximum duration of 12 months (Supplementary Fig. 2f and Fig. S6). While implanted, the ICs were neither powered nor electrically monitored. Explantations were phased at 3 months, 7 months, and 12 months. At every phase, two chips from each IC foundry were explanted.

## Animals, implantation, and explanation procedures

All experiments were performed per the EC Council Directive of September 22, 2010 (2010/63/EU). All procedures were reviewed and approved by the Animal Care Committee of the Research Center for Natural Sciences and the National Food Chain Safety Office of Hungary (license number: PE/EA/1253-8/2019). The implantation procedures were carried out as follows. Wistar rats ($n = 6$; weight, 315.5 ± 59.6 g at the initiation of the treatment; 3 females, 3 males) were anaesthetized by an intraperitoneal injection of a mixture of ketamine (75 mg/kg of body weight; CP-Ketamin, Produlab Pharma B. V., Raamsdonksveer, The Netherlands) and xylazine (10 mg/kg of body weight; CP-Xylazin, Produlab Pharma B. V., Raamsdonksveer, The Netherlands). A body temperature of 37 °C was sustained with an electric heating pad connected to a temperature controller (Supertech, Pécs, Hungary). Before implantation, samples were sterilized by immersing them in isopropyl alcohol for at least 20 min and then washing them with a continuous stream of distilled water for 2 min. To reduce the number of animals used, each animal was subcutaneously implanted with two ICs: Chip-A on the right and Chip-B on the left side of the back close to the neck. The incision was closed using interrupted sutures followed by a standard combined postoperative analgesic regimen. Samples were explanted at three time points: months 3, 7, or 12. Before explantations, rats were initially anaesthetized in the same way as described above. After anesthesia induction, the animals were overdosed with isoflurane (5% in 100% oxygen) until breathing stopped. Before explantations, the skin around the implants was examined for any inflammation. After long-term implantation, a tissue pocket was formed around each sample. The samples were removed from the pocket to analyze the tissue pocket. The pocket was immersed in 4% paraformaldehyde solution for 24 hours, then washed in 0.1 M PBS and stored in PBS. The tissue samples were then sectioned and stained with hematoxylin and eosin stain, and photomicrographs were taken for tissue analysis. Explanted microchips were carefully rinsed with DI water, blow-dried, and stored in a dried condition for a week before analysis.

## Electrical characterization of IDC structures: Electrochemical impedance spectrometry (EIS)

EIS was conducted with a Solartron Analytical Modulab XM equipped with a Potentiostat, Frequency Response Analyzer, and a Femtoammeter (Supplementary Fig. S5). For low-current measurements, the samples were placed in a dedicated Faraday cage with the shield of the Solartron coaxial cable connected to the cage. Unless stated otherwise, all EIS measurements were performed between the interdigitated combs in a frequency range of 0.01 Hz to 100 KHz using a 100 mV (RMS) sinusoidal voltage drive. Prior to initiating the actual measurements, a reference 1 TΩ ($10^{12}$ Ω) resistor load was tested to validate the accuracy and reliability of the measurement setup. Once the setup was confirmed to function correctly, the measurements on the IC test structures were performed.

## Electrical characterization of transistor structures

The transistor structures were characterized using an HP4145AB parameter analyzer, which monitored the transistor drain-source

current ($I_{DS}$) over a range of gate-source voltages (-0.5 V < $V_{GS}$ < 1.8 V) while having the $V_{DS}$ at 1.8 V DC.

## PDMS decapsulation

PDMS decapsulation was performed post-aging to analyze the IC materials protected by the PDMS coating. Decapsulation was done in two steps: 1) gently removing the excess PDMS material using a scalpel without damaging the ICs, and 2) dissolving the remaining PDMS material using a PDMS solvent (DOWSIL™ DS-2025). The first step of removing the excess PDMS on the samples would reduce the required exposure time to the solvent. The samples were, therefore, placed in the solvent for only 30 minutes. Decapsulated ICs were then thoroughly rinsed in acetone, IPA, and DI water, and finally, blown dry. All samples used for surface analysis were prepared similarly.

## Light microcopy

light microscopy was done using the Leica MZ6 and DM2500 stereo microscopy systems.

## IC multilayer stack characterization: Scanning electron microscopy (SEM), focused ion beam (FIB) milling, and energy-dispersive X-ray (EDX) analysis

For SEM, samples were coated with a thin (10–20 nm) evaporated carbon layer. The carbon-coated samples were investigated in a Thermo Scientific SCIOS2 system with a Pt deposition gun filled with Me3PtCpMe and a Sidewinder gallium liquid metal ion source (LMIS) for FIB. Before cross-sectioning, Pt was deposited in situ to develop the cross-sectional images' structure properly. EDX was done with an Oxford Xmax 20 SDD (silicon drift detector). The applied Ga milling parameters were 30 keV and 15–30 nA, depending on the trench dimensions for coarse milling. Later, the cross-section surface was polished with 30 keV, 5 nA.

## Surface characterization: Atomic force microscopy (AFM)

The Bruker Icon was used in tapping and PeakForce Quantitative Nanomechanics (QNM) mode. After PDMS decapsulation, a 20 μm x 20 μm area on the PDMS-edge (boundary between the PDMS-coated and exposed region) was used for AFM scanning. On the explanted ICs, AFM was used to determine the loss of the $SiN_X$ and $SiO_X$ passivation layers.

## Chemical composition of IC passivation layers pre and post-aging: X-ray photoelectron spectroscopy (XPS) depth profiling

XPS analysis was carried out in a Quantera Hybrid SXMtm from ULVAC-PHI at Eurofins EAG Laboratories, The Netherlands. The measurements were performed using monochromatic AlKα-radiation (25 Watt) and a take-off angle 45°. The information depth during surface measurement is approximately 7 nm at this angle. A spot size of 100 μm in diameter was used for the analyses. First, at the surface, a survey scan was recorded. Subsequently, accurate narrow scans of Si, N, O, C, Al, Na, Cl, and F have been measured for quantification in the depth profiles. Standard sensitivity factors were used to convert peak areas to atomic concentrations. For Chip-A and Chip-B ICs, XPS analysis was done on the M3-IDC and the M5-IDC test structures, respectively, as they contained flat surfaces with no microtopography.

## Chemical composition and barrier property of IC passivation layers pre- and post-aging: Time-of-flight mass spectrometry (ToF-SIMS) depth profiling

ToF-SIMS analyses were performed using an ION TOF IV instrument at Eurofins EAG Laboratories, The Netherlands. The instrument was operated in positive and negative mode using 2 keV $O^{2+}$ and 2 keV $Cs^+$ sputtering ions, respectively. At negative mode, sputtering was carried out using the 2 keV $Cs^+$ primary ion beam to enhance the detection level of the electronegative species [$O^-$], [$OH^-$], and [$H^-$]. The sputtered

area was 250 x 250 $μm^2$ and the analyzed area was 50 x 50 $μm^2$ centered in the sputtered area. The analysis was done with a beam of 25 keV $Bi^+$ ions. To increase the sensitivity for lighter elements like hydrogen [$H^-$], all samples were stored in the instrument under ultra-high vacuum (<$10^{-9}$ bar) for 64 hours before analysis. Like XPS, all ToF-SIMS measurements were performed on the M3-IDC and the M5-IDC test structures for Chip-A and Chip-B, respectively, where the surface of the chip is flat. The depth scale has been calibrated using known $SiN_X$ reference samples and has a measurement error of ~ 10% (including a possible systematic error due to differences in stoichiometry). Shallow depth profiles (down to ~20 nm) have been measured with 500 eV $Cs^+$ ions for better depth resolution. For analyzing the $SiN_X$ and $SiO_X$ passivation layers the [$SiN^-$] and [$SiO_2^-$] cluster ions were used to evaluate the layers[73]. It has been previously demonstrated that moisture can dissociate the Si-N, Si-O and Si-Si bonds in silicon-based ceramics and create silanol groups (Si-OH) in the atomic structure[37,53,74]. For this reason, the moisture barrier properties of the passivation layers were analyzed by monitoring the intensity of the [$OH^-$] cluster ion within the depth profiles[75-77]. In all cases, results on aged ICs were compared to depth profile results from reference ICs (as received from the foundries) after normalizing the data with respect to the reference [$^{30}Si^-$] signal. To compare the thickness of the oxidized layers between the exposed and PDMS-coated regions, the depth in which the [$SiO_2^-$] intensity is higher than the [$SiN^-$] intensity is used to identify the oxidized layer on the $SiN_X$ passivation.

Given the high sensitivity of ToF-SIMS compared to XPS (ppm to ppb for ToF-SIMS vs. 0.1–1.0 at. % for XPS), certain elements could be identified but not quantified. For this reason, all data on aged ICs have been compared to the levels measured on reference samples (as received from the foundries).

## Statistic and reproducibility

For SEM cross-sectional analysis and AFM, each sample was measured at two separate locations with measured values being <5% different from each other. For ToF-SIMS and XPS analysis, each sample was measured on two different locations for both the PDMS-coated and exposed regions of the IC surface. Data reproducibility was <10%. For EIS electrical characterization, at each time point, the IDC structures were measured twice with reproducibility of the results being <1%. Measurements were taken from different samples at different time points (see Table S3, Supplementary Information). For the dielectric sensor, at each time point, measurements across the entire sensor array were taken three consecutive times with measurements being <5% different from each other.

## Reporting summary

Further information on research design is available in the Nature Portfolio Reporting Summary linked to this article.

# Data availability

All data supporting the findings of this study are available within the article and its supplementary files. Any additional requests for information can be directed to, and will be fulfilled by, the corresponding authors. Source data are provided with this paper.

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

## Acknowledgements

The authors thank Jurgen van Berkum, Jeannette Smulders and Peter Graat from Eurofins EAG Laboratories, Eindhoven, The Netherlands, for their support in the ToF-SIMS and XPS and AFM analysis. For SEM analysis, we would like to thank Mr. Levente Illés from the Center for Energy Research, Budapest, Hungary. This research was funded by the following projects: Project CANDO (Controlling Network Dynamics with Optogenetics), funded by UK EPSRC (grant ref: NS/A000026/1) and the Wellcome Trust (contract ref: 102037/Z/13/Z), T.G.C. and N.D. Project POSITION-II, funded by the Electronic Components and Systems for European Leadership Joint Undertaking (ECSEL JU) in collaboration with the European Union's H2020 Framework Program (H2020/2014-2020) and National Authorities (grant agreement Ecsel-783132-Position-II-2017-IA), D.C, I.U. and V.G. NerveRepack, co-funded by the European Union and the CHIPS Joint Undertaking, Grant Agreement number 101112347, V.G. The Hungarian Brain Research Program 3.0, grant number NAP2022-I-8/2022 (to IU), by the European Union and the Hungarian Government: PharmaLab, grant number RRF-2.3.1-21-2022-00015 (to IU), I.U.

## Author contributions

K.N. planned and designed the experiments, prepared the samples, performed the study, collected and analyzed the data, and wrote and revised the manuscript. A.S.I. analyzed the data, and wrote and revised the manuscript. C.L. designed and performed the study, and collected and analyzed the data. C.D., Ö.C.A., D.H., K.T., D.M., and I.U. collected and analyzed the data. F.M., T.G.C., and N.D. designed the Chip-A test structures. K.N. and Ö.C.A. designed the Chip-B test structures. W.S.,

A.V., N.D., and V.G. analyzed the data, and wrote and revised the manuscript. C.D., I.U., T.G.C., W.S., A.V., N.D., and V.G. supervised the study, acquired the funding and provided resources. All authors contributed to reviewing and editing the manuscript.

## Funding

## Competing interests
The authors declare no competing interests.

## Additional information

[1]Department of Microelectronics, Faculty of Electrical Engineering, Mathematics and Computer Science, Delft University of Technology, Delft, The Netherlands. [2]Department of Medical Physics and Biomedical Engineering, University College London, London, UK. [3]Department of Electrical & Electronic Engineering, Imperial College London, London, UK. [4]UK Dementia Research Institute, Care Research and Technology Centre, London, UK. [5]Mint Neurotechnologies Ltd, London, UK. [6]Centre for Energy Research, HUN-REN, Budapest, Hungary. [7]Nikhef - Dutch National Institute for Subatomic Physics, Amsterdam, the Netherlands. [8]Research Centre for Natural Sciences, Institute of Cognitive Neuroscience and Psychology, HUN-REN, Budapest, Hungary. [9]Pazmany Peter Catholic University, Faculty of Information Technology and Bionics, Budapest, Hungary. [10]Department of Neuroscience, Erasmus Medical Center, Rotterdam, The Netherlands. [11]School of Biomedical Engineering & Imaging Sciences, King's College London, London, UK. [12]Department of System Integration and Interconnection Technologies, Fraunhofer Institute for Reliability and Microintegration IZM, Berlin, Germany. ✉e-mail: v.giagka@tudelft.nl

