## [Transparent Peer Review file · Nature Communications]

On the Longevity and Inherent Hermeticity of Silicon-ICs: Evaluation of Bare-Die and PDMS-Coated ICs After Accelerated Aging and Implantation Studies

Corresponding Author: Dr Vasiliki Giagka

Version 0:

Reviewer comments:

Reviewer #1

(Remarks to the Author)

This manuscript provides detailed, in depth material characterizations of the passivation layer of silicon IC and PDMS as biofluid barrier during long-term implantation and chronic in vitro aging tests. There is no doubt that the manuscript contains a lot of useful information, such as the speed of SiNx and SiO₂ etching and leakage of metal to tissue in the setting of long-term implantation/aging test. SiNx and SiO₂ are common passivation layers in IC and there have been numerous studies of these materials for biofluid barriers for microelectronic implants. This work, while being more detailed and in depth, has a rather incremental contribution built on many previous studies. The assumed novelty of the work largely lies on PDMS as a bio-permeable packaging material. However, I do not find sufficient evidence in the manuscript that can establish a thin layer of PDMS as a long-lasting barrier for microelectronic implants.

The major issues centered around this severe caveat are 1) while the authors motivate the study by the need of high-density packaging and PDMS as a soft biocompatible coating, they use extremely thick, bulky PDMS, more than 10 times the volume of the IC chip (see Fig. 2f, the PDMS is several mm in thickness and tens of mm in width). This thick encapsulation is not useful or provide no benefit over other approaches for most applications, and I would argue if one can use unrestrictedly large packaging volume, many materials will work as a barrier for a year. The authors did point out that the thin edge where PDMS encapsulation ends protected the IC. I argue that a small area of thin edge is likely to be exempted from the leading failure mode of thin-layer packaging: the mechanical stress during implantation and subsequent cracking and delamination.

2) there is very little electrical measurements. The authors admitted electrical measurements had to terminate pre-mature because the corrosion of the wire bonding, which was encapsulated by thick blob of PDMS. This actually provides strong contradicting evidence against PDMS for sealing and packaging, because even a huge amount of PDMS as used in this study is not effective in protecting metals from bio-fluid. I want to emphasize, and I hope the authors agree that, the success of packaging for implantable bioelectronics lies on being able to make and maintain electrical connections during long-term implantation. However, the manuscript provides no evidence supporting that PDMS packaging can achieve this.

In summary, because packaging is a critical issue for bioelectronics implants, I carefully read the manuscript over and over again but I still feel that nothing strikes as a breakthrough. I would be supportive if the authors could demonstrate thin coating of PDMS as effective protective packaging by providing 1) thin coating of PDMS with quantified thickness; 2) long-term electrical measurements in vivo and in vitro using a thin layer of PDMS as the encapsulation. Additionally, the paper writes in a style more like a Ph. D. thesis rather than a punchy journal paper. There are a lot of methods mixed in the results, making it difficult to read and understand the key outcome of the study. Significant modification on writing will be necessary.

Reviewer #2

(Remarks to the Author)

Reviewer #3

(Remarks to the Author)

The manuscript is highly detailed and clearly demonstrates the effort and care with which the authors conducted the work. However, it does suffer from a few flaws that prevent an immediate recommendation for publication, and is perhaps in need of a re-focusing. In the reviewer's opinion, the manuscript is unnecessarily lengthy, and spends too much time describing findings which merely support or provide additional evidence for already-known/described phenomena, such as the dissolution of silicon compounds in various media. The manuscript acknowledges these observations via the citations on page 4, lines 140-141. The reviewer acknowledges that new and useful insights can always be obtained, and that this was likely done in the current manuscript, but the work does not do an excellent job informing the reader on how the new findings complement or update existing knowledge on the subject. This should be clarified and made more concise. Secondly, the aspects of the work which may be more novel, such as the "custom-designed" dielectric sensor, receive relatively little attention. The effects of constant biasing are also of interest in the larger field, and could perhaps receive a greater share of focus here, though they were addressed. Overall, the work is good, but could be slimmed down and the focus honed to better represent its original contributions. It could also be made somewhat clearer throughout that the work does not represent any attempt to address or improve the known issues with these material systems, strictly to observe and quantify them. A few other specific points are introduced below.

1. Lines 101-102 Page 3: The authors claim that limited in-depth studies have been performed evaluating the long-term stability of IC materials in the implant environment. This may be true, but depending on the materials intended and definition of "in-depth", studies certainly do exist on the aging and degradation behavior of IC-related materials and should be acknowledged. For instance, references 39-42 of the manuscript may check some of these boxes already. DOI: 10.1109/NEMS50311.2020.9265554, 10.1109/JSEN.2022.3178640, 10.1109/ICSJ.2017.8240098, 10.1109/JSEN.2021.3068077, 10.3390/mi11090810, 10.1109/EMBC44109.2020.9176443, 10.1016/j.bios.2021.113683, 10.1109/IFETC46817.2019.9073712, and "Quantifying the Biodegradation of Packaging Thin Films Intended for Medical Micro Devices" by Herrera Morales et al., 2015 are some examples of other potentially useful papers.

2. Lines 155-156 Page 4: It may be worthwhile to specify that the ICs used in this study used the same traditional IC passivation materials as described more generally earlier on.

3. Line 173 Page 4: For those interested, it might be nice to specify which "advanced material analysis techniques" were used, since this is apparently part of the novel aspect of the work.

4. Line 185-186 Page 6: It may be useful to provide some context as to why chips from two different foundries were compared. Differences in manufacturing practices, materials, etc.? The chips were not the same layouts, so why couldn't one foundry have made both? And so on.

5. Line 221-226 Page 6: Was it assumed that no side-permeation would occur thanks to the WoV? Or was this assessed/verified in any way? I believe later on some amount delamination is described.

6. Line 257-264 Page 7: The custom-designed dielectric sensor is interesting, and more could be said about its function and possible future utility for other researchers.

7. Line 279-280 Page 7: Because the implanted ICs were not powered, if a comparison is made between the implanted devices and the in vitro devices, it is important to only consider the un-biased subset of in vitro devices. If this was done, it should be made clear in following sections where relevant.

8. Line 280 Page 7: Simply noting that the text says "Explanations" where "Explantations" was likely intended.

9. Line 337 Page 10: 14 out of 56 samples is described as "a few of our samples". I don't necessarily agree with the characterization of 1/4 of the samples as "a few".

10. Line 389-396 Page 12: It should be clarified whether the observations being made here apply to PBS solution at 67C or rat at 37C, as the section header includes both.

11. Line 618-619 Page 19: The authors note that "In the PDMS-coated regions, however, the SiO₂ is always protected by both PDMS and the top SiN_x passivation layer". While this is true, is there any concern that after the SiN_x is dissolved from the exposed areas, dissolution could continue on now the exposed SiN_x underneath the PDMS layer? If so, this could undercut the PDMS layer over time, which appears to be assumed not to happen here. Also, any side permeation at this point?

12. Line 795-798 Page 25: Would an amenable solution to the issue of SiN_x surface dissolution be to simply include a top SiO₂ layer? This would appear to solve most of the issues described in the manuscript.

13. Line 809-810 Page 25: A reference would be useful regarding the corrosion between gold wire bond and aluminum pads.

14. Figures 6 and 7 are generally quite busy. Could any parts be placed in the supplement? For instance, Figure 7b doesn't seem all that useful when the focus is more on the oxidation information, etc.

Version 1:

Reviewer comments:

Reviewer #1

(Remarks to the Author)

I would like to thank the authors for their effort in revising the manuscript, which has significantly improved its clarity and readability. However, I still have some major concerns and comments that I would like the authors to address. One significant caveat of the study is that in all the in vitro and in vivo tests, the ICs were largely free from mechanical stress. This does not accurately reflect the continuous mechanical stress the implants would experience during real-world applications, particularly noting that the ICs were not connected in the in vivo tests. While the authors emphasize the critical nature of interfacial adhesion between the PDMS and the passivation layer, they do not provide bonding strength tests conducted during aging. This is a critical piece of data that should be included in the manuscript. Fundamentally, if the bonding of PDMS to the IC surface has significantly weakened, and the PDMS merely sits on top of the passivation layer after some time in the aging test or in vivo implantation, it may still act as a barrier to slow down the etching of SiNx under the mechanical stress-free conditions presented in this manuscript. However, it is unlikely to withstand real-world testing where mechanical stress between the device and PDMS is present. Providing convincing evidence that the PDMS-device bonding remains intact during long-term aging or implantation is essential for establishing PDMS as an effective barrier for implanted devices and for preventing any potential misinterpretation in the field. I appreciate Supplementary Note 1, where the author highlighted the interface between PDMS and IC surface. I appreciate Supplementary Note 1, where the author highlights the interface between the PDMS and the IC surface. However, since no mechanical stress was applied and no evaluation of the bonding strength was conducted, this significant concern of degradation of bonding strength cannot be dismissed. A particularly misleading term used in the manuscript is 'Hermeticity/Hermetic' in the title and throughout the text. Hermetic packaging typically refers to air- and water-tight barriers, while the authors acknowledge that PDMS is a moisture-permeable material, which, in my opinion, renders it non-hermetic in nature. I recommend removing 'hermetic' from the title and abstract, and clearly distinguishing PDMS packaging from hermetic packaging throughout the manuscript. Another limitation of the PDMS packaging approach is its reliance on defect-free SiNx as the passivation layer, which not only provides an excellent moisture and ionic barrier for the active component but also facilitates strong interfacial bonding to PDMS when properly surface treated. However, this strategy may not be widely applicable, as many biological implants feature a metal top layer or vias without the protection of a passivation layer. ICs with these exposed metal components were not tested, and the strong bonding between metal and PDMS can be challenging, raising concerns about the adaptability of this approach in diverse applications. This limitation should be articulated in the discussion session. The most valuable result from the manuscript, in my opinion, is that the ICs remained functional and structurally intact under continuous DC bias for 12 months in vitro aging test at 67 degree. While it is well known that SiNx constructs a stable passivation layer, having these quantitative measures of longevity is valuable. The authors reported failures, too. It is important to clearly document the success rate in these tests. Along this vein, the manuscript provides comprehensive characterizations of the longevity of IC implants under various in vitro and in vivo conditions, which will be valuable information for the field. However, I would like to request that the authors provide more specific details about the test conditions and results than what is currently included in Table S6. Currently, the test durations are grouped together. Please include the sample number, time to failure, and failure mode for each test duration and condition. Furthermore, it would be helpful to label the samples used in each figure panel and to clearly indicate whether the same samples were used across multiple experiments. If the same samples were indeed reused, please justify this approach, as multiple samples should ideally be tested under identical conditions. Some claims made should be more specific and not overstated. A few examples of potential overstated or misleading claims include:
The use of 'Hermeticity/Hermetic' in the title and throughout the manuscript, which I articulated above.
"Secondly, they show the electrical stability of the IC passivation and IMD layers surrounding the IDC metals when fully submerged in PBS solution for at least a year while being continuously subjected to a maximum electric field of 0.25 MV/cm (at 15V DC)"
This section appears overly optimistic. Can the authors provide a more precise summary here, detailing the numbers of tested at various bias voltages and how many successfully lasted for 12 months? Additionally, please refer to Table S6 and modify it according to my earlier suggestions.
"Thirdly, regarding the PDMS coating, these results demonstrate that PDMS effectively prevents water leakage paths by forming long-lasting interfacial adhesion to the IC's nitride passivation. In addition, they attest to the stable electrical insulation properties of PDMS while electrically biased and submerged in PBS solution."
I don't believe these results adequately address the role of PDMS. The authors have demonstrated that the IC passivation layer is stable without the presence of PDMS, but they did not test the interfacial adhesion between PDMS and the IC passivation layer. Even if the PDMS does not function as a barrier, the IC may still operate effectively. I will not enumerate all the instances along this line, but I suggest that the authors thoroughly review the manuscript to correct any overstated claims.

Reviewer #2

(Remarks to the Author)

Reviewer #3

(Remarks to the Author)

Nanbakhsh et al. addressed all the reviewer's comments and corrected them accordingly. I am happy to see the revised manuscript structure; it is ready to be accepted.

Reviewer response key:

- Reviewer comments in blue
- Response to reviewer comments in black
- Changes to the manuscript in red

Reviewer 1 (comments to the Authors):**Comment 1.1:**

This manuscript provides detailed, in depth material characterizations of the passivation layer of silicon IC and PDMS as biofluid barrier during long-term implantation and chronic in vitro aging tests. There is no doubt that the manuscript contains a lot of useful information, such as the speed of SiNx and SiOx etching and leakage of metal to tissue in the setting of long-term implantation/aging test. SiNx and SiOx are common passivation layers in IC and there have been numerous studies of these materials for biofluid barriers for microelectronic implants. This work, while being more detailed and in depth, has a rather incremental contribution built on many previous studies.

Response 1.1:

We thank the reviewer for their positive feedback on the depth and detail of our study. We also thank the reviewer for the chance to modify our novelty statement to be more specific and highlight the advances of the present work over the state-of-the-art.

As the reviewer mentioned, the bio-etching of IC-related materials such as SiNx and SiOx have been reported before, but mainly as stand-alone material films. Here, we aimed at evaluating the electrical and material stability of IC structures as a whole which include the silicon bulk, transistors, multilayer intermetal dielectric and metallization layers and the top SiNx and SiOx passivation layers. As such, our study, investigates not only known phenomena, such as the dissolution of SiNx and SiOx, but also the integrity of such material stack.

We would like to highlight two key aspects of this study which we believe are breakthroughs in the field of polymer packaging for implantable devices.

1. In this work, we directly exposed bare chips to physiological environments and for the first time demonstrated that IC structures fabricated by commercial CMOS foundries, can maintain their electrical performance for at least 12 months even in the presence of continuous DC bias voltages. Our complementary material evaluation demonstrated that the stable electrical performance is due to the excellent moisture and ionic barrier properties of the IC structure, making the IC an inherently 'hermetic' and waterproof structure itself.
2. The second novelty aspect of this work lies in using PDMS as a standalone moisture-permeable packaging material. PDMS is known to be freely permeable to moisture (water vapour). As it will be explained in detail in Response #1.2, even when using centimetres thick PDMS, the entire polymer will be saturated

with moisture within hours [R1.1, R1.2]. In this scenario, the underlying IC will be **exposed to 100% humidity**. Despite this, PDMS can be considered a biological barrier, i.e. preventing ions and organic species from touching the device [R1.2, R1.3]. The high moisture permeability of PDMS, however, has discouraged many from using it as a standalone packaging material. Here, we have demonstrated that ICs themselves can be inherently hermetic, allowing a soft elastomer such as PDMS to be used for long-term packaging.

Regarding the bio-etching of SiO_x and SiN_x films, we acknowledge that extensive studies have examined the aging behaviour of these thin-films in biological environments. However, we would like to emphasize that, although all these films are commonly referred to by their generic names SiO_x and SiN_x, they are not created equally. Their behaviour and biostability are significantly influenced by their manufacturing processes and chemistry. In fact, the vast majority of studies have demonstrated that the barrier properties, defect/pinhole densities, and bio-etching rates can vary greatly depending on the specific deposition techniques and post-processes used by the manufacturer. Therefore, to investigate the hermeticity of IC structures, we evaluated the long-term barrier properties of their SiO_x and SiN_x passivation layers, in addition to other materials in the stack. Our findings suggest that ICs can be inherently hermetic, likely due to the advanced processes and high-quality materials which are used by the commercial foundries. In the context of micropackaging, this means that for the hermeticity, one can rely on the IC itself and use PDMS as an easily accessible standalone packaging material.

We would like to highlight that if the success of packaging is dependent on specific processes and barrier materials available only in a few academic labs, it will limit its path to translation. For that, processes and materials that are standardized and available to all must be used. Therefore, we have consciously chosen to evaluate the hermeticity and barrier properties of commercially available IC structures and we believe our results are a breakthrough in the field of bioelectronic packaging.

Finally, it should be noted that not all CMOS ICs will have the same hermiticity as the ones evaluated here (350 nm and 180 nm nodes). By moving to smaller technology nodes (<90 nm) different materials are being used for the barrier layers and intermetallic dielectrics (IMD) [R1-4]. For example, instead of SiN_x, SiCN (carbon-doped SiN_x) is used which has reported to have a lower density. And for the IMD layers, porous dielectrics are replacing the conventional high density SiO_x to lower the k-value. These modifications can undermine the hermiticity of the structure. Therefore, our findings are greatly valuable in designing the next generation of devices when it comes to foundry selection, IC design and final packaging.

References:

[R1.1] Traeger, Richard. "Nonhermeticity of polymeric lid sealants." *IEEE Transactions on parts, Hybrids, and packaging* 13, no. 2 (1977): 147-152. **Added as reference [47] to the manuscript.**

[R1-2] Donaldson, P. E. K. "The essential role played by adhesion in the technology of neurological prostheses." *International journal of adhesion and adhesives* 16, no. 2 (1996): 105-107. **Reference [48] in the manuscript.**

[R1-3] Kim, Sung Hwan, Jin-Hee Moon, Jeong Hun Kim, Sung Min Jeong, and Sang-Hoon Lee. "Flexible, stretchable and implantable PDMS encapsulated cable for implantable medical device." *Biomedical Engineering Letters* 1 (2011): 199-203. **Added as Reference [49] to the manuscript.**

[R1-4] Cartailier, Vivien. "Study of temperature and moisture impact on water diffusion in materials and on electronic circuits failure mechanisms." PhD diss., Université de Bordeaux, 2021. **Reference [38] in the manuscript.**

Our modifications to the manuscript: Based on the reviewer's feedback and recommendations, we have refocused and modified the manuscript which can be found at the end of this response letter. A summary of the changes is given below:

Title: modified to better reflect the scope **(Lines 5-7)**

Abstract: minor modification (Lines 67-70)

1. Introduction: modified and shortened

- **Removed lines 123-126, 132-134, 141-143 and 145-212**
- **Added lines 136-138, 141-143 and 214-220**
- **Added new section: 2. Reliability concerns for implantable ICs (lines 221-264)**

3.1: Overview of IC test structures and aging study: modified and shortened

- **Moved major parts of lines 277-375 to Experimental**
- **Added new measurement showing PDMS thickness: see lines 427-429 and Supplementary Note 3**
- **Moved lines 377-404 to new section (Section 3.2)**
- **Added new section: 3.2. Chemical composition of the foundry IC passivation and IMD layers (lines 407-435)**

3.3. Electrical performance in PBS solution at 67 °C: Lines 504-506.

3.4.1.1. Stability of the IC structures in PBS (unbiased) and rat: modified and shortened

- **Moved lines 593-604 to Supplementary Note 5**
- **Removed lines 553-566 and 575-646**
- **Removed Figure 4d-e to shorten manuscript**
- **Added lines 646-663**

3.4.1.2. Ionic barrier properties of IC passivation layers: modified lines 679-684

3.4.1.3. Chemical stability and moisture barrier properties of IC passivation layers: modified and shortened

- **Moved lines 725-729, 735-736 and 761-764 to Experimental Section**
- **Moved lines 794-812 to Supplementary Note 7**
- **Moved Figures 6i-m to Supplementary Note 7**
- **Added lines 814-819**

3.4.2. Effect of electrical field: DC electrical bias in PBS solution at 67 °C: modified to simplify Figure 7

- Removed lines 825-830, 840-856 and 886-887
- Removed Figures 7b-c to shorten manuscript

3. Discussion and guidelines for longevity: modified and shortened

- Removed lines 913-957, 1003, 1005-1006, 1017-1023 and 1057-1075
- Added lines 959-995, 1007-1009, 1015, 1023-1029 and 1077-1084

Conclusion: removed (lines 1099-1110) to shorten manuscript

5. Experimental section:

- Added lines from main text: 1114-1152, 1167-1180, 1194-1198 and 1295-1304

Comment 1.2:

The assumed novelty of the work largely lies on PDMS as a bio-permeable packaging material. However, I do not find sufficient evidence in the manuscript that can establish a thin layer of PDMS as a long-lasting barrier for microelectronic implants.

The major issues centered around this severe caveat are 1) while the authors motivate the study by the need of high-density packaging and PDMS as a soft biocompatible coating, they use extremely thick, bulky PDMS, more than 10 times the volume of the IC chip (see Fig. 2f, the PDMS is several mm in thickness and tens of mm in width). This thick encapsulation is not useful or provide no benefit over other approaches for most applications, and I would argue if one can use unrestrictedly large packaging volume, many materials will work as a barrier for a year.

Response 1.2:

As the reviewer mentioned, using PDMS as a moisture-permeable material for IC packaging is indeed one of the new aspects of our study. As explained in Response 1.1, it is well established that PDMS is freely permeable to moisture (gaseous water). To address the concern of the reviewer regarding the thickness of the PDMS, we would like to raise two points.

First, we would like to clarify a misconception. In Figure 2f, the PDMS the reviewer is referring to is not used as a coating but only as a handling substrate during implantation. To help clarify this, below we have added a schematic showing how most of the top IC surface was left exposed to tissue. In fact, the critical top 10 μm stack of the chip structure with the microelectronics (i.e., the passivation, buried IC metallization and active areas) is only partially PDMS coated. The remaining surface of the die is left directly exposed to tissue.

For the samples in the accelerated in vitro setup, this PDMS handling substrate is fully absent, and the IC's sidewalls, in addition to the top surface, are fully exposed to the electrolyte. Again, the top passivation near the wire-bond region is only thinly PDMS coated.

Secondly, regarding the overarching comment of the reviewer concerning the PDMS thickness, we argue that the PDMS volumes used here for coating ICs, are, in fact, irrelevant for moisture permeation for the long durations investigated. As mentioned earlier, it is well known that PDMS is freely permeable to moisture, and the permeation rates are such that even a cm-thick PDMS layer will be saturated with water vapour within a day upon exposure to biofluids [R1.1].

Based on the above, we argue that even in the thickest areas, PDMS offers no additional advantage compared to a thin layer in terms of moisture protection. **In fact, in terms of moisture protection, we have not relied on PDMS, but on the hermeticity of the IC structure itself.** Based on PDMS's moisture permeability, it can only act as a successful packaging material if all the structures underneath are hermetic and/or moisture-robust with good adhesion to PDMS [R1.2]. For this reason, we focused on the stability and hermeticity of foundry IC die structures. The stability of the electrical connections, as the reviewer has also asked in Comment 1.4, is equally important and mitigations are proposed in Response 1.5 for realizing a moisture-robust wire-bond connection.

Despite PDMS's moisture-permeability, it has been shown in literature that it can be used as a protective coating, blocking the biological species and ions from touching the underlying device [R1.3]. Our results also support previous findings but additionally show two important aspects: i) As shown in Figure 4 and Supplementary Figures S13-S14, we show that even a thin PDMS layers can prevent the SiNx from dissolution by preventing the body's ionic fluids and enzymes from touching the underlying nitride layer. ii) In Figure 7, by comparing the exposed and PDMS-coated regions of biased ICs, we show that PDMS can prevent ion ingress within the top oxidized layer of the SiNx, even in the presence of electrical fields.

Our modifications to the manuscript: Following the reviewer's suggestion, we have also conducted **new measurements to quantify the PDMS thickness across the passivation area for the in vivo and in vitro samples data can be found in Supplementary Note 3. Also lines 254-255 and 427-428 have been added to the manuscript (see at the end of response letter).**

Supplementary Note 3: Measurements were done to quantify the thickness of the PDMS coating on the **thickest region** for the *in vitro* and *in vivo* samples. Measurements done using DCM8 3D surface metrology microscope.

Sample	Thickness (μm)
In vitro sample #1 (with wire-bonds)	793
In vitro sample #2 (with wire-bonds)	631
In vitro sample #3 (with wire-bonds)	750
In vivo sample #1 (without wire-bonds)	284
In vivo sample #2 (without wire-bonds)	310

Comment 1.3:

The authors did point out that the thin edge where PDMS encapsulation ends protected the IC. I argue that a small area of thin edge is likely to be exempted from the leading failure mode of thin-layer packaging: the mechanical stress during implantation and subsequent cracking and delamination.

Response 1.3:

We thank the reviewer for this comment. We would like to first address a possible misunderstanding: our goal in showing the thin layer near the PDMS edge was mainly to show that even thin layers of PDMS, despite its high moisture-permeability, can prevent SiNx dissolution by blocking the tissue and biological species from direct contact with the IC material.

We would like to clarify that, unlike thin-layer (sub- μm) moisture barrier packaging, our packaging approach uses a standalone sub-mm soft elastomer, and rather relying on the moisture barrier properties of the packaging, we depend on the hermeticity of the embedded IC structure itself.

In fact, when packaging ICs for real-use applications, the use of a sub-mm soft top layer is unavoidable if the sharp edges and wire-bond loops are to be protected, even when using thin-layer packaging solutions. As an example, in [R1.5] a thicker PDMS layer is applied on an Parylene coated IC to create a soft transition to the body.

We examined the thin PDMS layer near the edge as we believe this area represents the worst-case scenario for stress-induced cracking and delamination compared to a fully

coated IC. In the edge, the interface between the PDMS and IC passivation is a point where lateral tissue ingress could occur, further degrading the SiN_x beyond the edge. In a fully coated IC, that interface would be completely sealed from the biological environment, enhancing our confidence in the protection offered by PDMS.

As the reviewer has pointed out, delamination of the polymer could lead to failure. The adhesion of PDMS to the IC structure was, in fact, carefully monitored and evaluated (See Supplementary Note 1). Our impedance measurements and material analysis showed that delamination of the PDMS from the chip passivation never occurred. In terms of stress-induced delamination, a thicker PDMS, as opposed to the thin edge, will act as a mechanical buffer absorbing the stresses rather than transferring them to the PDMS-IC interface.

Furthermore, cracking of thin-film packaging materials is indeed a known phenomenon for brittle materials with high Young's modulus. However, cracking of the thin PDMS was not observed as softness, compliance, and high biostability are some of the unique properties of PDMS.

References:

[R1.5] Stieglitz, Thomas. "Development of a micromachined epiretinal vision prosthesis." *Journal of neural engineering* 6, no. 6 (2009): 065005.

Our modifications to the manuscript: Following the reviewer's comment, we have added lines 246-264 for clarification. Also, we have added reference [R1.5].

Comment 1.4:

There is very little electrical measurements. The authors admitted electrical measurements had to terminate pre-mature because the corrosion of the wire bonding, which was encapsulated by thick blob of PDMS. This actually provides strong contradicting evidence against PDMS for sealing and packaging, because even a huge amount of PDMS as used in this study is not effective in protecting metals from bio-fluid.

Response 1.4:

The reviewer raises a concern that we present too few electrical measurements, and that measurements were terminated pre-maturely (at 5-month) due to wire-bond failure. We would like to further clarify that only the electrical measurements for the dielectric sensor chip were terminated pre-maturely. For all other test structures (IDC and NMOS structures), we have presented a full set of electrical measurements. For example, Figures 3, and Supplementary Figure S8 show in total n=28 sample which were tested for at least 12 months.

Regarding the comment on the wire bonding, we have addressed this in Response 1.5.

Our modifications to the manuscript: Following the reviewer's comment, we have added lines 969-974 for clarification.

Comment 1.5:

I want to emphasize, and I hope the authors agree that, the success of packaging for implantable bioelectronics lies on being able to make and maintain electrical connections during long-term implantation. However, the manuscript provides no evidence supporting that PDMS packaging can achieve this.

Response 1.5:

We fully agree with the reviewer that "the success of packaging for implantable bioelectronics lies on being able to make and maintain electrical connections during long-term implantation". However, we would add that this success equally relies on the integrity and stability of the IC die structures with the microelectronics within.

Here, we disagree with the comment that the observed corrosion and wire bond failures are a sign that PDMS is not a suitable encapsulant. Rather, the Au/Al wire bonds, as made on the samples in this study, are not a suitable interconnect method for PDMS packaging. As mentioned in Response 1.2, given that PDMS is freely permeable to water vapour, it can only act as a packaging material under the condition that all the structures underneath can withstand 100% humidity. In this case, our wire-bond failures stem from the fact that two dissimilar metals are used, leading to galvanic corrosion at their interface. This is a well-documented phenomenon as shown in [R1.7, R1.8], and expected when these dissimilar metals are exposed to 100% humidity. The obvious way to circumvent this is to avoid using dissimilar metals: use Al wire-bonds instead of Au [1-7], or gold plate aluminium pads as in [R1.9]. As reported in [R1.7] and also demonstrated in our work (Supplementary Figure S9), PDMS creates a long-lasting bond to the native oxide layer on the aluminium pads and prevents it from corrosion. In our study, however, the unavailability of Al wire-bonding equipment in our facility and limitations which were imposed during COVID lockdown prevented us from optimising this aspect.

Another aspect of maintaining a stable electrical connection is the ability of the packaging to prevent shunt leakage paths between metal interconnects. In this regard, PDMS prevented any leakage paths in between the interconnect metals even when applied to 15 V DC voltages for at least 12 months. This is due to the stable underwater insulation properties of PDMS and the strong interface adhesion that it makes to the IC passivation. Our long-term impedance measurements showed values > 100 GOhms, which to the best of our knowledge, is a breakthrough in terms of long-term insulation stability when fully exposed to wet ionic environments.

References:

[1-7] Iannuzzi, Melanie. "Bias Humidity Performance and Failure Mechanisms of Nonhermetic Aluminum SIC's in an Environment Contaminated with Cl₂." *IEEE Transactions on Components, Hybrids, and Manufacturing Technology* 6, no. 2 (1983): 191-201. **Added as reference [62] to the manuscript.**

[1-8] Baylakoğlu, İlknur, Aleksandra Fortier, San Kyeong, Rajan Ambat, Helene Conseil-Gudla, Michael H. Azarian, and Michael G. Pecht. "The detrimental effects of water on electronic devices." *e-Prime-Advances in Electrical Engineering, Electronics and Energy* 1 (2021): 100016. **Added as reference [61] to the manuscript.**

[R1.9] Frederick, Rebecca A., Ellen Shih, Vernon L. Towle, Alexandra Joshi-Imre, Philip R. Troyk, and Stuart F. Cogan. "Chronic stability of activated iridium oxide film voltage transients from wireless floating microelectrode arrays." *Frontiers in Neuroscience* 16 (2022): 876032.

Our modifications to the manuscript: We have made the corresponding **modifications in lines 1002-1009.**

We have also added references [R1-7] and [R1-8] to the main manuscript.

Comment 1.6:

In summary, because packaging is a critical issue for bioelectronics implants, I carefully read the manuscript over and over again but I still feel that nothing strikes as a breakthrough. I would be supportive if the authors could demonstrate thin coating of PDMS as effective protective packaging by providing 1) thin coating of PDMS with quantified thickness; 2) long-term electrical measurements in vivo and in vitro using a thin layer of PDMS as the encapsulation.

Response 1.6:

We thank the reviewer for acknowledging that packaging is a critical issue for bioelectronic implants. In Response 1.1, we have reformulated our novelty statement and breakthrough. In Response 1.2, we have conducted additional measurements to quantify the thickness of the PDMS used in our study. To address the misconception of PDMS packaging, we have clarified that given the high moisture-permeability of PDMS, the entire PDMS will be saturated with 100% moisture within hours/days regardless of the thicknesses used on the samples. In fact, we have highlighted in the paper, that for a successful PDMS protection, one should not rely on the water barrier properties of the coating, but on the inherent hermeticity and moisture-robustness of the structures underneath. In this regard, we have evaluated and demonstrated that foundry IC structures can be waterproof and hermetic themselves, only requiring a soft coating like PDMS to protect them from direct contact with tissue. In Response 1.4, we have addressed the concerns on the limited electrical measurements.

Our modifications to the manuscript: Following the reviewer's comment, **we have major modifications to re-focus and clarify the manuscript. These are presented in Response 1.1 and 1.2 and 1.3.**

Comment 1.7:

Additionally, the paper writes in a style more like a Ph. D. thesis rather than a punchy journal paper. There are a lot of methods mixed in the results, making it difficult to read and understand the key outcome of the study. Significant modification on writing will be necessary.

Response 1.7:

We thank the reviewer for these points. Based on the reviewer's feedback, we have made major modifications that are highlighted in Response 1.1, and shown at the end of this response letter.

Reviewer #2 (Remarks to the Authors):

Comment #2.1

Response #2:

We would like to thank the reviewer for their contribution and detailed review.

Reviewer #3 (Remarks to the Authors):

Comment #3.1

The manuscript is highly detailed and clearly demonstrates the effort and care with which the authors conducted the work. However, it does suffer from a few flaws that prevent an immediate recommendation for publication, and is perhaps in need of a re-focusing.

Response #3.1

We thank the reviewer for their positive feedback on the effort and detail in the work. We also appreciate the detailed comments from the reviewer that provide a foundation for us to re-focus and revise our manuscript.

Our modifications to the manuscript: Based on the reviewer's feedback and recommendations, we have refocused and modified the manuscript which can be found at the end of this response letter. A summary of the changes is given below:

Title: **modified** to better reflect the scope **(Lines 5-7)**

Abstract: **minor modification (Lines 67-70)**

1. Introduction: **modified** and **shortened**

- **Removed lines 123-126, 132-134, 141-143 and 145-212**
- **Added lines 136-138, 141-143 and 214-220**

- Added new section: 2. Reliability concerns for implantable ICs (lines 221-264)

3.1: Overview of IC test structures and aging study: modified and shortened

- Moved major parts of lines 277-375 to Experimental
- Added new measurement showing PDMS thickness: see lines 427-429 and Supplementary Note 3
- Moved lines 377-404 to new section (Section 3.2)
- Added new section: 3.2. Chemical composition of the foundry IC passivation and IMD layers (lines 407-435)

3.3. Electrical performance in PBS solution at 67 °C: Lines 504-506.

3.4.1.1. Stability of the IC structures in PBS (unbiased) and rat: modified and shortened

- Moved lines 593-604 to Supplementary Note 5
- Removed lines 553-566 and 575-646
- Removed Figure 4d-e to shorten manuscript
- Added lines 646-663

3.4.1.2. Ionic barrier properties of IC passivation layers: modified lines 679-684

3.4.1.3. Chemical stability and moisture barrier properties of IC passivation layers: modified and shortened

- Moved lines 725-729, 735-736 and 761-764 to Experimental Section
- Moved lines 794-812 to Supplementary Note 7
- Moved Figures 6i-m to Supplementary Note 7
- Added lines 814-819

3.4.2. Effect of electrical field: DC electrical bias in PBS solution at 67 °C: modified to simplify Figure 7

- Removed lines 825-830, 840-856 and 886-887
- Removed Figures 7b-c to shorten manuscript

3. Discussion and guidelines for longevity: modified and shortened

- Removed lines 913-957, 1003, 1005-1006, 1017-1023 and 1057-1075
- Added lines 959-995, 1007-1009, 1015, 1023-1029 and 1077-1084

Conclusion: removed (lines 1099-1110) to shorten manuscript

5. Experimental section:

- Added lines from main text: 1114-1152, 1167-1180, 1194-1198 and 1295-1304

Comment #3.2

In the reviewer's opinion, the manuscript is unnecessarily lengthy, and spends too much time describing findings which merely support or provide additional evidence for already-known/described phenomena, such as the dissolution of silicon compounds in various media. The manuscript acknowledges these observations via the citations on page 4, lines 140-141. The reviewer acknowledges that new and useful insights can always be obtained, and that this was likely done in the current manuscript, but the work does not do an excellent job informing the reader on how the new findings complement or update existing knowledge on the subject. This should be clarified and made more concise.

Response #3.2

We thank the reviewer for the comments and based on their suggestion we have re-focused and shortened the manuscript.

Our modifications to the manuscript: modifications are given in Response #3.1 and are presented at the end of this response letter.

Comment #3.3

Secondarily, the aspects of the work which may be more novel, such as the “custom-designed” dielectric sensor, receive relatively little attention. The effects of constant biasing are also of interest in the larger field, and could perhaps receive a greater share of focus here, though they were addressed. Overall, the work is good, but could be slimmed down and the focus honed to better represent its original contributions. It could also be made somewhat clearer throughout that the work does not represent any attempt to address or improve the known issues with these material systems, strictly to observe and quantify them.

Response #3.3

We thank the reviewer for their recognition of the value of this work and for their feedback to improve the manuscript. Based on the reviewers feedback we have re-focused and slimmed down the paper. Regarding the comment that the work does not represent any attempt to address the known issues with these material systems, we would like to highlight that the materials and fabrication processes used by commercial CMOS foundries are strictly controlled and cannot be modified. Here, our aim was not only to quantify the degradation mechanisms of foundry IC structures, but to demonstrate that despite the degradation, these IC structures can be inherently hermetic and waterproof. More importantly, our intention was to show that PDMS, regardless of its moisture permeability, can prevent the degradations as observed in the exposed regions, making it a suitable IC packaging material for long-term applications.

Our modifications to the manuscript: Following the reviewer’s comment, we have added clarifying lines to the manuscript:

- Added lines: 136-143
- Added lines: 214-220
- Added lines: 246-264

A few other specific points are introduced below.

Comment #3.4

Lines 101-102 Page 3: The authors claim that limited in-depth studies have been performed evaluating the long-term stability of IC materials in the implant environment. This may be true, but depending on the materials intended and definition of “in-depth”, studies certainly do exist on the aging and degradation behavior of IC-related materials and should be acknowledged. For instance, references 39-42 of the manuscript may check some of these boxes already. DOI: 10.1109/NEMS50311.2020.9265554, 10.1109/JSEN.2022.3178640, 10.1109/ICSIJ.2017.8240098, 10.1109/JSEN.2021.3068077, 10.3390/mi11090810,

10.1109/EMBC44109.2020.9176443, 10.1016/j.bios.2021.113683, 10.1109/IFETC46817.2019.9073712, and “Quantifying the Biodegradation of Packaging Thin Films Intended for Medical Micro Devices” by Herrera Morales et al., 2015 are some examples of other potentially useful papers.

Response #3.4

We thank the reviewer for their comment and suggested references. We acknowledge that there have been extensive studies examining the aging and degradation behaviour of ‘IC-related’ materials. Nevertheless, we would like to emphasise that although all these thin-films materials may be commonly identified as SiO_x or SiN_x, they are not created equally, and their long-term stability and barrier properties can greatly vary based on the manufacturing processes. In fact, the vast majority of these studies, including the ones suggested by the reviewer, attest to the fact that the barrier properties, defect/pinhole densities and bio-etching properties can greatly vary based on the raw materials and unique deposition processes of the manufacturer. Here, we evaluated the electrical and material stability of foundry IC structures as a whole, with the foundry SiN_x and SiO_x layers being integral components of the structure. Our results, in fact, showed that the SiN_x from these two foundries, despite having very similar chemistries (N/S), showed an x2.3 difference in the *in vivo* dissolution rate. We believe these degradation rates are important as these CMOS chips are also being used bare die structures for neural probes.

Our modifications to the manuscript: For clarification we have made the corresponding modifications in the main text (see modified manuscript at the end of this response letter):

- Removed lines 123-126 and replaced with lines 127-130
- Added lines 225-242 and 262-264

Comment #3.2

Lines 155-156 Page 4: It may be worthwhile to specify that the ICs used in this study used the same traditional IC passivation materials as described more generally earlier on.

Response #3.2

We thank the reviewer for raising this point.

Our modifications to the manuscript: We have included section 3.2 (lines 444-470).

Comment #3.3

Line 173 Page 4: For those interested, it might be nice to specify which “advanced material analysis techniques” were used, since this is apparently part of the novel aspect of the work.

Response #3.3

We thank the reviewer for the comment.

Our modifications to the manuscript: We have added lines 436-440.

Comment #3.4

Line 185-186 Page 6: It may be useful to provide some context as to why chips from two different foundries were compared. Differences in manufacturing practices, materials, etc.? The chips were not the same layouts, so why couldn't one foundry have made both? And so on.

Response #3.4

As the reviewer has also mentioned, the processes and materials that are used in the manufacturing of the metallization and dielectric layers can be different for each foundry, and this will greatly impact their stability in aggressive physiological environments. Therefore, our goal was to evaluate the stability of ICs from two commercial foundries and also to investigate the effect of different metal layers on the hermiticity and lifetime.

Our modifications to the manuscript: We have added the lines 409-413.

Comment #3.5

Line 221-226 Page 6: Was it assumed that no side-permeation would occur thanks to the WoV? Or was this assessed/verified in any way? I believe later on some amount delamination is described.

Response #3.5

With regards to the hermiticity of bare IC structures, side-permeation was, in fact, carefully monitored using electrical measurements (using the IDC structures and the dielectric sensor) and SEM cross-sectional analysis. As we did not have test structures without the WoV, we cannot verify that side-permeation was solely prevented due to the use of the WoVs or the high barrier properties of the intermetallic dielectrics (IMD). Nevertheless, it can be expected that the WoV would further reinforce the side integrity of the IC structure.

Regarding the delamination: the observed delamination happened for Chip-A ICs between the whole IC stack and the Si substrate, where the WoV would not be effective. Chip-B ICs, fabricated by foundry-B, never showed this failure mode, highlighting again the difference which can exist in the manufacturing processes of CMOS chips. Nevertheless, for all Chip-A devices which showed delamination in the bare die regions, PDMS-coated regions showed an intact structure, suggesting that PDMS is an effective coating in preventing this failure mode.

Comment #3.6

Line 257-264 Page 7: The custom-designed dielectric sensor is interesting, and more could be said about its function and possible future utility for other researchers.

Response #3.6

We thank the reviewer for this comment.

Our modifications to the manuscript: We have added the following lines 1028-1029 and 1082-1084 in the manuscript (see manuscript at the end of this response letter).

Comment #3.7

Line 279-280 Page 7: Because the implanted ICs were not powered, if a comparison is made between the implanted devices and the *in vitro* devices, it is important to only consider the un-biased subset of *in vitro* devices. If this was done, it should be made clear in following sections where relevant.

Response #3.7

We thank the reviewer for raising this point.

Our modifications to the manuscript:

We have modified the section 3.4.1.1 header as ‘Stability of the IC structures in PBS (**unbiased**) and rat’, line (542-432)

We have modified the section 3.4.1.3 header as: ‘Chemical stability and moisture barrier properties of IC passivation layers (**unbiased**)’, line (720-721)

We have also changed the legend for Figure 6: Chemical stability and moisture barrier properties of PDMS-coated and exposed passivation layers after accelerated *in vitro* (**unbiased**) and *in vivo* aging analyzed using negative mode ToF-SIMS depth profiling.

Comment #3.8

Line 280 Page 7: Simply noting that the text says “Explanations” where “Explantations” was likely intended.

Response #3.8

We thank the reviewer for their correction.

Our modifications to the manuscript: We have made the correction in line 1197.

Comment #3.9

Line 337 Page 10: 14 out of 56 samples is described as “a few of our samples”. I don’t necessarily agree with the characterization of ¼ of the samples as “a few”.

Response #3.9

We agree with the reviewer and have made the following modifications in the main manuscript.

Our modifications to the manuscript: We have made the corresponding modification in the line 496 (**Some** of our samples (14 out of 56) showed EIS irregularities).

Comment #3.10

Line 389-396 Page 12: It should be clarified whether the observations being made here apply to PBS solution at 67C or rat at 37C, as the section header includes both.

Response #3.10

We thank the reviewer for the comment which we believe will help bring more clarity.

Our modifications to the manuscript: Line 550-551, we have added **PBS solution at 67 °C** to the sentence.

Comment #3.11

Line 618-619 Page 19: The authors note that “In the PDMS-coated regions, however, the SiO_x is always protected by both PDMS and the top SiN_x passivation layer”. While this is true, is there any concern that after the SiN_x is dissolved from the exposed areas, dissolution could continue on now the exposed SiN_x underneath the PDMS layer? If so, this could undercut the PDMS layer over time, which appears to be assumed not to happen here. Also, any side permeation at this point?

Response #3.11

The reviewer is correct that dissolution of the SiN_x will eventually continue laterally underneath the PDMS layer. Our objective by leaving the ICs partially exposed was to evaluate the moisture/ion barrier properties of the foundry SiN_x and SiO_x layers. In real-use applications when packaging IC structures, the entire IC structure will be fully encapsulated with a sub-mm PDMS layer, eliminating this failure mode.

Comment #3.12

Line 795-798 Page 25: Would an amenable solution to the issue of SiN_x surface dissolution be to simply include a top SiO_x layer? This would appear to solve most of the issues described in the manuscript.

Response #3.12

We thank the reviewer for raising this interesting question. In the exposed bare-die regions we indeed observed better stability of the foundry SiO_x passivation layer compared to the SiN_x layer. If the reviewer was referring to using a foundry deposited SiO_x on top of the SiN_x layer, we would like to note that CMOS foundries do not alter their manufacturing processes. If the reviewer was referring to including a non-foundry SiO_x layer, we would like to emphasise that such SiO_x layers deposited for example at academic research facilities, can contain pinholes and defects and be hydrolytically unstable even when no dissolution is observed [Reference 69 in manuscript].

Even if an ideal SiO_x layer would exist, besides complicating the packaging process, it would limit its application to a few facilities. Our goal here was to show that ICs, despite the observed degradations, can be structurally hermetic, likely due to the advanced processes and high-quality materials used by these foundries. As these ICs are commercially available for all, our second goal was to show how PDMS, despite its moisture permeability, can be used as a cheap and easily accessible material to solve the degradation issues.

Comment #3.13

Line 809-810 Page 25: A reference would be useful regarding the corrosion between gold wire bond and aluminum pads.

Response #3.13

We thank the reviewer for this comment. We have added the references below to the main paper.

Our modifications to the manuscript:

We have added the following two reference to the manuscript:

[R3.1] Iannuzzi, Melanie. "Bias Humidity Performance and Failure Mechanisms of Nonhermetic Aluminum SIC's in an Environment Contaminated with Cl₂." *IEEE Transactions on Components, Hybrids, and Manufacturing Technology* 6, no. 2 (1983): 191-201.

[R3.2] Baylakoğlu, İlknur, Aleksandra Fortier, San Kyeong, Rajan Ambat, Helene Conseil-Gudla, Michael H. Azarian, and Michael G. Pecht. "The detrimental effects of water on electronic devices." *e-Prime-Advances in Electrical Engineering, Electronics and Energy* 1 (2021): 100016.

Comment #3.14

Figures 6 and 7 are generally quite busy. Could any parts be placed in the supplement? For instance, Figure 7b doesn't seem all that useful when the focus is more on the oxidation information, etc.

Response #3.14

We thank the reviewer for this point. Based on the feedback, Figures 6 and 7 have been modified with some sections moved to the Supplementary Information.

Our modifications to the manuscript:

- Moved Figures 6i-m to Supplementary Note 7 (see Figure 6 in manuscript below)
- Moved lines 794-812 to Supplementary Note 7 (see manuscript below)
- Removed Figures 7b-c to shorten manuscript (see Figure 7 in manuscript below)

**Longevity of Implantable Silicon-ICs for Emerging Neural**
**Applications: PDMS and Silicon-IC Passivation Layers as**
**Long-Term Body-Fluid Barriers**

**On the Hermeticity and Longevity of silicon-ICs: Evaluation**
**of Bare-Die and PDMS-Coated ICs After Accelerated Aging**
**and Implantation Studies**

*Kambiz Nanbakhsh¹, Ahmad Shah Idil², Callum Lamont², Csaba Dücső³, Ömer Can Akgun¹,*
*Domonkos Horváth⁴, Kinga Tóth⁴, István Ulbert⁴, Federico Mazza⁵, Timothy G.*
*Constandinou^{5,6,7}, Wouter Serdijn^{1,10}, Anne Vanhoestenbergh^{8,9}, Nick Donaldson², Vasiliki*
*Giagka^{1,11}*

*¹Department of Microelectronics, Faculty of Electrical Engineering, Mathematics and*
*Computer Science, Delft University of Technology, Delft, The Netherlands.*

*²Department of Medical Physics and Biomedical Engineering, University College London,*
*London, United Kingdom.*

*³Centre for Energy Research, Budapest, Hungary.*

*⁴Research Centre for Natural Sciences, Institute of Cognitive Neuroscience and Psychology,*
*HUN-REN, Budapest, Hungary.*

*⁵Department of Electrical & Electronic Engineering, Imperial College London, United*
*Kingdom.*

*⁶UK Dementia Research Institute, Care Research and Technology Centre, London, United*
*Kingdom.*

*⁷Mint Neurotechnologies Ltd, London, United Kingdom.*

*⁸School of Biomedical Engineering & Imaging Sciences King's College London.*

*⁹London Institute of Healthcare Engineering – LIHE, London, United Kingdom.*

*¹⁰Department of Neuroscience Erasmus Medical Center, Rotterdam, The Netherlands.*

*¹¹Department of System Integration and Interconnection Technologies, Fraunhofer Institute*
*for Reliability and Microintragration IZM, Berlin, Germany.*

Keywords: Implantable bioelectronics, active neural interface, integrated circuits, biofluid barriers, microelectronics packaging, PDMS.

Abstract

Silicon integrated circuits (ICs) are central to the next-generation miniature active neural implants, whether packaged in soft polymers for flexible bioelectronics or implanted as bare die for neural probes. These emerging applications bring the IC closer to the corrosive body environment, raising reliability concerns, particularly for chronic use. Here, we evaluate if the advanced materials utilized by the IC foundries can function as effective long-term barriers, protecting the IC in the body. For this aim, the electrical and material performance of ICs sourced from two foundries were evaluated through one-year accelerated *in vitro* and *in vivo* studies. ICs featured custom-designed test structures and were partially PDMS coated, creating two regions on each chip, uncoated "bare die" and "PDMS-coated". During the accelerated *in vitro* study, ICs were electrically biased and periodically monitored, demonstrating Results demonstrated stable electrical performance due to the superior barrier properties of the IC's passivation layers suggesting the inherent hermeticity of bare IC structures. Despite this, material analysis, however, revealed degradation of IC materials in the bare die regions. In contrast, PDMS-coated regions revealed no degradation, PDMS-coated regions, however, revealed limited degradation making PDMS a highly suitable IC encapsulant for years-long implantation. Based on the new insights, guidelines are proposed that may enhance the longevity of implantable ICs, significantly broadening their applications in the biomedical field.

1. Introduction

Since the insertion of the first silicon integrated circuit (IC) into the brain for sensing neural activity^{1,2}, tremendous efforts have been made to exploit the high performance and power efficiency offered by the semiconductor industry in opening new applications in the field of healthcare: brain-machine interfaces³⁻⁵, flexible bioelectronics^{4,6-9}, biosensors¹⁰⁻¹³, active silicon probes^{6,14,15}, and optogenetics^{16,17}.

Driven by the need for miniaturization, these emerging applications are moving away from the hermetic metal enclosures traditionally utilized for IC protection in the body and are opting for newly engineered thin organic and inorganic coatings^{6,18-23}. This shift, however, introduces reliability risks by bringing the chip closer to the corrosive body environment. Silicon-ICs are liable to failure if penetrated by body fluids: field effect transistors are susceptible to mobile ions (e.g., Na⁺, K⁺) that might reach the gate oxide^{24,25}, and water anywhere in the intricate sub-micron structures of the IC will facilitate corrosion and leakage currents^{26,27}. Additionally, the body itself should be protected from the electrical bias voltages on the chip, which could pose a hazard if the IC's insulation is breached. With the growing interest in miniaturizing active neural implants, these reliability risks, and the uncertainty regarding the longevity of the IC have been identified as one of the main obstacles limiting the widespread clinical adoption of these emerging applications, particularly for chronic use²⁸⁻³⁴.

The long-term reliability of implantable ICs relies on the stability of their constituent materials. ~~Thus far, however, there have been limited in-depth studies evaluating the long-term stability of IC materials in the implant environment. Gaining such insight will be instrumental in guiding future IC design and material engineering for IC packaging, ultimately contributing to the development of next-generation, long-lasting active neural implants.~~ Thus far, studies have been evaluating the stability of 'IC-related' thin-film materials such as silicon nitride (SiN_x) and silicon dioxide (SiO₂), mainly as standalone material films fabricated in research cleanrooms^{21,35-37}. Limited data, however, exists on the long-term stability and inherent hermeticity of actual foundry-fabricated IC structures as a whole.

~~In this paper, we evaluated the electrical and material performance of ICs sourced from two complementary metal-oxide-semiconductor (CMOS) foundries after long-term *in vitro* and *in vivo* studies.~~

~~Here, we evaluated the hermeticity of IC structures sourced from two complementary metal-oxide-semiconductor (CMOS) foundries by monitoring their electrical and material performance over long-term *in vitro* and *in vivo* studies. Our primary objectives were: 1) to determine longevity of bare ICs when directly exposed to physiological media by identifying the degradation pathways that can lead to failure and 2) to evaluate the possibility of utilizing polydimethylsiloxane silicone rubber (PDMS) as a soft *biocompatible but moisture-permeable* coating, *in-preventing the degradations and extending the longevity of implantable ICs to extend the longevity of implantable ICs.*~~

~~Silicon ICs are intricate multilayer structures comprised of conducting metallization layers and insulating ceramic layers that are used as intermetallic dielectrics (IMD). Between the different metal layers, metal vias are used for making vertical interconnections. All layers of the IC stack are deposited onto a silicon substrate (Figure 1). The chemical composition of the materials used in the IC can vary depending on the technology node and the deposition processes utilized by the IC foundry³⁵. For the metallization, aluminum (Al) or copper (Cu) are generally used; for the vias tungsten (W) and the insulating IMD layers, silicon dioxide is commonly employed³⁶. The topmost insulating layers, known as the "passivation" layers, are usually a dual layer of silicon nitride (SiN_x) and silicon dioxide (SiO₂). These layers are deposited by plasma-enhanced chemical vapor deposition (PECVD) techniques, coating the~~

155 entire surface of the IC except for the openings used as bonding pads. The passivation
layers have been included explicitly in the CMOS process to protect the IC against ambient
humidity and contamination. The IC's sidewalls, however, are not protected by the
passivation layers. For this reason, the CMOS industry has used a die seal ring (a stack of
metallization encircling the die) as a side metal barrier to protect the inner circuitry from any
impurity ingress and to provide structural support during dicing³⁷. From the bottom, the IC is
protected by the ~ 200–300 μm thick silicon substrate, which has high density and high
barrier properties³⁸. Given the protections from the bottom and the sidewalls, the longevity of
the IC largely depends on the top passivation layer's chemical stability and barrier
properties. Both these properties are determined by the material type and the deposition
processes utilized by the CMOS foundry.

When implanted as bare die, the IC passivation layers serve as the main protecting barriers.
However, pre-existing defects, pinholes, or nanopores have been reported for PECVD
ceramic layers, which could compromise their protection in wet ionic environments such as
the human body³⁹. Water molecules may also cause interlayer delamination in the IC stack
when reaching interfaces, causing mechanical damage and failure^{35,39}. PECVD silicon nitride
and dioxide layers have also been reported to dissolve when exposed to wet ionic media39–
42.

When coated in PDMS, the IC is protected from direct exposure to tissue and body fluids.
Nevertheless, due to PDMS's high moisture permeability, the IC material may still undergo
degradation when implanted in the body. In our previous work⁴³, we demonstrated that not
all silicon-based dielectrics are equally suitable for long-term implantation as diffusing
moisture through the PDMS may also penetrate the dielectric layer and gradually degrade its
insulating and dielectric properties. Thus, for PDMS-coated ICs, having long-lasting moisture
barrier properties is a key requirement for the IC materials when targeting chronic use.
Besides barrier properties, strong adhesion between the IC material and the PDMS ensures
long-term electrical functionality. Strong interfacial adhesion is required to maintain electrical
insulation between the wire-bonded pads^{26,44}. Adhesion stability itself is influenced by the
surface chemistry of the IC's passivation layer and the PDMS type²⁶.

In this investigation, ICs were partially coated with PDMS, intentionally leaving most of the
IC's passivation surface and sidewalls exposed as bare die (**Figures 1a–b**). Where bare, the
chip's intricate metallization and transistor structures are only protected by the IC's own
passivation and IMD layers. In the PDMS-coated region, PDMS serves as additional
protection but also insulates the wire bonds and the bonding pads through its adhesion to
the IC passivation. ICs were either subjected to a one-year accelerated *in vitro* aging study
in phosphate-buffered saline (PBS) solution at 67 °C or implanted in rats for a year.

Three test structures were implemented on the ICs: 1) interdigitated capacitors (IDC), 2)
metal-oxide-semiconductor (MOS) transistors, and 3) a dielectric sensor array with on-chip
measurement and processing circuitry (**Figures 1c–e**). Using these structures, the electrical
performance of the ICs is periodically monitored throughout the accelerated *in vitro* study,
contributing to the long-term evaluation of the IC materials and the applied PDMS interfaces.

Complementary to the electrical measurements, several material analysis techniques were
used to evaluate and compare the degradation mechanisms in the bare die and PDMS-
coated regions of the aged chips. For both regions, the impact of two aging environments
(PBS solution at 67 °C and rat) was also evaluated and compared. For this purpose, for the
first time, advanced material analysis techniques have been used on CMOS foundry ICs to
explore the bio-chemical degradation mechanisms after their long-term exposure to the
implant environment.

~~The study presented here advances our understanding of the electrical and material stability~~
~~of silicon ICs in the implant environment, both as bare die and when coated with PDMS. The~~
~~newfound insights from this study will inform and enable the design of state-of-the-art chip-~~
~~scale active neural implants intended for chronic implantation.~~

[revised manuscript text omitted]

3. Results

IC design, fabrication, and PDMS coating

Here, we present the custom-designed IC test structures fabricated using two commercial CMOS foundries (hereafter called Chip-A and Chip-B). All ICs were partially PDMS-coated and subjected to either a one-year accelerated *in vitro* study in PBS solution at 67 °C or a one-year *in vivo* animal study in rats.

For the accelerated *in vitro* study, bare dice were glued on ceramic adaptors with pre-printed Pt/Au tracks, gold wire-bonded, and then partially coated with PDMS (**Figure 2a**, details in Experimental Section). By partially coating the ICs, two distinct regions are created on the same chip: 1) an exposed “bare die” region and 2) a “PDMS-coated” region (**Figure 2b**). During the aging studies, both regions were simultaneously stressed and evaluated. On all prepared samples, the test structure under investigation was mainly left exposed (as bare die). This approach enabled us to evaluate how effectively the IC’s own passivation and dielectric layers can protect the sensitive test structures underneath while being subjected to the different aging media.

In the PDMS-coated region, critical interfaces are created between the PDMS and the IC material (i.e., SiN_x, aluminum pad, silicon substrate). Strong long-lasting adhesive bonds at these interfaces are key in preventing voids and shunt water leakage paths that are required to ensure the long-term electrical functionality of the IC. On our structures, the most critical PDMS interface is with the SiN_x passivation at the PDMS-edge area (**Figure 2b** and **Figure S1**, Supporting Information). In this area, the PDMS-SiN_x interface is directly subjected to the PBS solution. Interfacial debonding would allow lateral ingress of the solution, causing significant failure when the electrolyte reaches the electrically biased wire-bonds.

During the accelerated *in vitro* study, the IDC test structures were electrically biased. This was done either by applying a DC electrical voltage between the combs of the IDC or between the combs and the PBS solution. The main objectives for biasing the IDC structures were to: 1) electrically stress the dielectrics and passivation layers while being directly exposed to liquid PBS solution in the uncoated regions, or to diffusing moisture (gaseous water) in the PDMS-protected regions, 2) accelerate failure in the directly exposed region in case of microcracks or defects in the passivation layers, as previously reported⁴⁶⁻⁴⁷, and 3) in the PDMS-coated region, investigate if the combined effects of a continuous electric field and moisture would weaken the PDMS-SiN_x interface adhesion, potentially leading to debonding and lateral ingress of saline.

A key factor in the design of the test structures is the presence or absence of a metal “shield (SH)”. The shield is a metal layer implemented using the top-metal of the process and designed to further protect the test structures underneath from any moisture/ion ingress. Additionally, around each test structure, a wall-of-via (WoV) is implemented which is a side metal barrier similar to the CMOS die seal ring. The WoV is included around each individual test structure to reduce the possibility of moisture/ion penetration from the edges (sidewalls) of the test structure.

Chip-A test structures were fabricated in a 0.35 μm CMOS 4-metal process. Two IDC test structures were implemented on Chip-A: 1) M3-IDC and 2) M4-SH/M3-IDC (**Figure 2d** and **Table S1**, Supporting Information). M3-IDC (area: 1150 μm x 850 μm) contained 606 interdigitated fingers implemented in M3 (one layer below the top metal). It is protected by the IMD and the top SiN_x/SiO₂ passivation layers, each being ~1 μm thick, as presented in the cross-sectional scanning electron microscope (SEM) image in **Figure 2b**. The adjacent M4-SH/M3-IDC test structure used a similar M3-IDC interdigitated structure but included a

329 thick top metal shield ($\sim 2.8 \mu\text{m}$ thick) implemented in M4 (See cross-sectional SEM image
in **Figure S2**, Supporting Information).

Chip-B test structures were fabricated using a $0.18 \mu\text{m}$ 6-metal process and contained three
types of structures: 1) IDCs, 2) MOS transistors, and 3) a dielectric sensor array (**Figure 2e**
and **Table S2**, Supporting Information). **Figure S3**, Supporting Information, shows a
representative WoV around the test structures on Chip-B and the material used by the Chip-
B foundry. A similar WoV was also used around the two structures on Chip-A.

The IDC structures on Chip-B were implemented in Metal 5 (M5-IDC) or Metal 6 (M6-IDC).
One IDC structure was also implemented with a shield on top (M6-SH/M5-IDC). In the M5-
IDC structure, the comb metallization is in M5 and is protected by the IMD and the top
$\text{SiN}_x/\text{SiO}_2$ passivation layers, each having a thickness of $\sim 0.6 \mu\text{m}$ and $\sim 1.1 \mu\text{m}$, respectively
(see cross-sectional SEM image in **Figure 2e**). M6-IDC is implemented in the top metal layer
of the process and has the comb metallization protected only by the $\text{SiN}_x/\text{SiO}_2$ passivation
layers (**Figure S4**, Supporting Information). M6-SH/M5-IDC has the comb metallization on
M5 and a shield layer on M6 ($\sim 1 \mu\text{m}$ thick).

There are two MOS transistor structures with or without a shield barrier. All transistors have
a W/L of $20 \mu\text{m} / 0.36 \mu\text{m}$. The shield barrier for the MOS structure is a double-layer metal
implemented in M5 and M6.

The dielectric sensor was diced and separated from the other test structures on Chip-B to
ease the wire bonding and sample preparation required for the long-term accelerated *in vitro*
aging. The custom-designed sensor can measure IMD resistance values in the $10^{14} \Omega$ range
with the sensitivity to detect changes at a sub-M Ω resolution⁴⁸. This sensor was designed
and tested with two goals in mind: 1) to verify if a complex CMOS circuit could operate when
fully submerged in PBS solution at 67°C , and 2) to acquire more sensitive measurements of
the dielectric changes and capture a 3D mapping of the possible moisture/ion penetration
pathways into the chip.

For the accelerated *in vitro* aging study, the IDC structures ($n=22$ for Chip-A and $n=34$ for
Chip-B) were fully immersed in PBS solution at 67°C while continuously electrically biased.
For biasing, three voltages were used: unbiased ($n=12$), 5 V DC ($n=22$), and 15 V DC ($n=22$)
(**Figures S5a-b**, Supporting Information). These voltages were selected based on their
relevance for neurostimulator ICs³⁰. **Table S3**, Supporting Information, provides an overview
of all the samples and the respective test conditions used in the study. The electrical
performance of the IDCs was monitored monthly using electrochemical impedance
spectroscopy (EIS) for a duration of at least 12 months. The NMOS test structures ($n=5$) and
dielectric sensor ($n=1$) were immersed in PBS at 67°C and monitored monthly while in soak
for a 12-month duration (**Figures S5c-d**, Supporting Information).

For the long-term *in vivo* study, $n=12$ ICs ($n=6$ from each IC foundry) were placed on PDMS
substrates and implanted in 6 rats for a maximum duration of 12 months (**Figure 2f** and
**Figure S6**, Supporting Information). While implanted, the ICs were neither powered nor
electrically monitored. Explantations were phased at 3 months, 7 months, and 12 months. At
every phase, two chips from each IC foundry were explanted.

On a group of reference samples (as is from foundry), the chemical composition of the IC
passivation layers was characterized using time-of-flight secondary ion mass spectrometry
(ToF-SIMS) and X-ray photoelectron spectroscopy (XPS). The silicon nitride on both Chip-A
and B ICs was measured to be non-stoichiometric with a Si: 50% and N: 50% (N/Si: ~ 1),
typical for PECVD deposited SiN_x layers³⁵. The hydrogen content in the nitride layers was
measured using ToF-SIMS to be $\sim 20\%$ and $\sim 15\%$ for Chip-A and B, respectively (See
**Table S5** and **Figure S7**, Supporting Information).

[revised manuscript text omitted]

silicon dioxide used in the Chip-B IMD stack, which has been reported to be unstable in the
presence of moisture^{47,49}. Thirdly, regarding the PDMS coating, these results demonstrate

that PDMS effectively prevents water leakage paths by forming long-lasting interfacial
adhesion to the IC's nitride passivation. In addition, they attest to the stable electrical
insulation properties of PDMS while electrically biased and submerged in PBS solution.

**Some** of our samples (14 out of 56) showed EIS irregularities (See Supplementary **Table**
**S6**), which were attributed to either, 1) wire-bond corrosion (See Supplementary **Figures S9**
**and S10**, Supplementary Information), 2) stress-induced cracking of the passivation layers,
or 3) poor conformality of the passivation layers (Supplementary **Note 4**). Notably, the latter
two failure scenarios on both IC foundries were found to be influenced by the use of the top
metallization^{40,50} (Supplementary **Figures S11 and S12**). The poor conformality or cracks
expose the IDC metals to the PBS solution and when electrically biased, result in severe
degradation and short-term failures. All other IDC structures implemented with lower
metallization (~~M3 in Chip-A and M5 in Chip-B~~) showed no EIS irregularities, showed no
structural damage or EIS irregularities (M3 in Chip-A and M5 in Chip-B, see Supplementary
**Figures S11a and S13**).

NMOS transistors on Chip-B, both with (n=1) and without (n=4) the double shielded metal
barrier, showed stable V_{GS} - I_{DS} characteristics over the 12-month accelerated aging (**Figure**
**3b**). For the NMOS structures without the shield metal barriers, the passivation and IMD
stack serve as the protective barriers between the ionic solution and the MOS structure.

**Figure 3c** presents the results for the exposed dielectric sensor (Chip-B) after 5-months of
aging in PBS solution at 67 °C. Comparing the results at 0-month and 5-month shows
minimal change in the average and standard deviation results for the entire array. Testing on
this chip stopped due to an intermittent wire-bond connection failure. The results, however,
demonstrate the stability of the dielectric materials on Chip-B (fluorinated silicon dioxide).
More significantly, these results demonstrate the stable electrical functionality of a complex
CMOS IC designed in a 0.18 μm CMOS process when submerged as a bare chip in a 67 °C
PBS solution.

Figure 3. Long-term electrical performance of test structures on Chip-B during accelerated aging in PBS solution at 67 °C. **a)** Electrochemical impedance spectroscopy (EIS) results of representative M5-IDC test structures without (top, n=4) and with a top-metal (M6) shield (bottom, n=4), showing stable capacitive characteristics over the duration of aging. Results are given as average and standard deviation (n=4). **b)** Top view optical micrograph and average V_{GS} - I_{DS} transfer characteristics of NMOS transistors at 0-month and 12-month (n=4 unshielded and n=1 shielded MOS structures). Schematics show the distance between the test structures and the surface of the IC which is exposed to PBS solution (dimensions not to scale). **c)** Top view optical micrograph of the dielectric sensor with a high magnification image of the pixels (left). Measurement results giving the value of each pixel within the array (total 5382 pixels) at 0-month and 5-month (n=1). Comparing the average and standard deviation of the entire array at 0-month and 5-month show minimal change (right). All electrical measurements are performed while the ICs and their PDMS-coated wire bonds are fully submerged in PBS solution.

3.4. Material performance: exposed vs. PDMS-coated

In the previous section, the electrical stability of various IC test structures was demonstrated
over a 12-month accelerated *in vitro* aging study in PBS solution at 67 °C. Despite the
electrical stability, exposure to ionic fluids may still degrade the IC's materials without
causing discernible changes in the electrical characteristics. Given that identification of the
degradation pathways can aid in the longevity estimation of the ICs, as a next step, we
analyzed the materials on both the exposed (bare die) and PDMS-coated regions of the
aged ICs. Our investigation began by using cross-sectional SEM imaging to evaluate the
stability of the IC's entire multilayer stack. We were particularly evaluating the stack for any
instances of interlayer delamination or intralayer degradation. Next, we examined the
chemical stability and barrier properties of the IC's SiN_x/SiO₂ passivation layers using ToF-
SIMS and XPS depth profiling. During the material evaluations, a comparative analysis was
also conducted to understand the impact of the two aging environments on degradation:
PBS solution at 67 °C and the *in vivo* animal environment at 37 °C. Finally, we investigated
the effect of long-term exposure to electrical fields by focusing on a subset of IDC structures
continuously subjected to electrical biasing (5 V and 15 V) in PBS solution at 67 °C for 12
538 months.

3.4.1. Effect of environment: PBS solution (67 °C) and rat (37 °C)

3.4.1.1. ~~Stability of IC multilayer stack~~ Stability of the IC structures in PBS 543 (unbiased) and rat

The stability of the IC's multilayer stack was first evaluated using optical microscopy and
later by SEM, examining the surface and various cross-sections across the chip. For both
Chip-A and B samples, cross-sections were created using focused ion beam (FIB) at three
different locations on the IDC test structures (corner, center, and PDMS-edge). **Figure 4a-c**
shows representative cross-sectional SEM images taken at the PDMS-edge for two M3-IDC
test structures (from Chip-A) at two different time points, 3-month and 10-month **in PBS**
**solution at 67 °C**. SEM results show intact metallization and no sign of interlayer
delamination in the IC stack. Higher magnification imaging, however, reveals a thinning of
the top SiN_x passivation in the exposed region. ~~while no such degradation is observed for~~
~~the PDMS-coated region (Figure b-c). At 3 month, ~640 nm of the entire 1 μm SiN_x~~
~~passivation remains on the sample, corresponding to a SiN_x dissolution rate of ~120~~
~~nm/month for Chip-A. Cross sections on other locations of the IDC showed the dissolution to~~
~~be uniform. Based on this dissolution rate (Figure S20, Supporting Information), at ~8.5-~~
~~month, a complete loss of the entire SiN_x is expected. Figure 4c shows a cross-sectional~~
~~SEM image of a similar test structure aged for 10 months where only the SiO₂ passivation is~~
~~present in the exposed region, corroborating the previously calculated SiN_x dissolution rate.~~
~~Note that despite the complete loss of the SiN_x layer after ~8.5 months, EIS results for the~~
~~M3-IDC test structures on Chip-A remained stable for at least 12 months of aging in PBS~~
~~solution. Based on the SEM images, from month 8.5 to 12, the SiO₂ passivation is exposed,~~
~~and together with the intermetallic dielectric layer (IMD), they serve as the protective barriers~~
~~on the M3-IDC. Similar cross-sectional SEM investigations for a Chip-B sample also showed~~
~~a chemical dissolution of the exposed nitride layer (Figure S13).~~

Note that despite the loss of the SiN_x layer, EIS results for the M3-IDC test structures on
Chip-A remained stable for at least 12 months of aging in PBS solution. The PDMS-coated
regions show no sign of nitride loss. Similar results were found for the explanted ICs
demonstrating that even thin layers of PDMS (< 1 μm), despite its moisture-permeability, can
prevent nitride dissolution by maintaining a strong interfacial adhesion and effectively
blocking tissue contact with the film (see Supplementary **Figures S14-S16** and
**Supplementary Note 5**).

A group of Chip-A (n=10) and Chip-B (n=8) ICs were aged for longer than 12 months in PBS
at 67 °C. After 12 months, optical microscopy on all these Chip-A ICs showed gradual signs
of delamination near the sidewalls of the chip. SEM investigations revealed the delamination
to be between the silicon substrate and the entire IC stack (See **Figure S14**, Supporting
Information). The delamination, however, was only observed on the side walls directly
exposed to the PBS solution with the PDMS-protected regions remaining intact. Chip-B
samples aged for longer than 12 months showed no signs of delamination (**Figure S15**,
Supporting Information).

Next, the explanted ICs were investigated using SEM cross-sectional imaging and atomic
force microscopy (AFM). The 3-month explanted ICs were examined using optical
microscopy and cross-sectional SEM imaging (**Figures S16** and **S17**, Supporting
Information). Besides SiN_x dissolution on the directly exposed regions, no other degradation
in the IC material stack was observed for the 3-month explanted chips. The PDMS-coated
regions showed no observable SiN_x dissolution. More notably, it was found that even thin
PDMS layers (< 1 μm) protected the SiN_x from degradation as it prevented tissue and body
fluid contact with the IC.

The 7-month and 12-month explanted samples were analyzed using optical microscopy and
AFM. Samples were first cleaned from tissue residues, PDMS decapsulated and later
analyzed using optical microscopy and AFM on a 20 μm x 20 μm area at the PDMS edge
(**Figure 4d-e** and **Figure S18**, Supporting Information). Optical inspections revealed a non-
uniform color on the exposed regions of the chip surface, indicating possible non-uniform
dissolution of the passivation. The non-uniform dissolution could be due to the non-
homogeneous coverage of various enzymes and tissue on the IC's surface. After
decapsulation, PDMS-protected regions were similarly inspected, showing no signs of
degradation (See **Figure S18**, Supporting Information). AFM analysis on the 7-month and
12-month explanted ICs showed a loss of the SiN_x passivation (**Figure 4d-e** and **Figure**
**S19**, Supporting Information). Based on the calculated dissolution rates (**Figure S20**,
Supporting Information) on the 12-month explanted Chip-A samples, the 1 μm SiN_x
passivation layer had completely dissolved after ~ 8 months, leaving the SiO₂ passivation
directly exposed to the body for ~4 months. Despite the 4-month direct exposure, AFM
results show the depth to be no less than 1 μm, indicating that the SiO₂ passivation on Chip-
A did not dissolve after 4 months of direct exposure to the body environment. Similarly, for
the 12-month explanted Chip-B samples, based on the dissolution rate of the SiN_x (**Figure**
**S20**, Supporting Information), after the complete dissolution of the SiN_x at month ~10, the
SiO₂ passivation was directly exposed for 2 months to the body environment. The AFM
profile at 12-month showed a loss of ~700 nm, indicating that in addition to the ~ 600 nm
SiN_x dissolution, the top SiO₂ passivation on Chip-B has also experienced a ~100 nm loss
within the two months of exposure to body. As revealed in section 2.1, the SiO₂ passivation
on Chip-B contains two oxide layers, with the topmost layer (~100 nm) having an x2.5 higher
[OH] intensity (**Figure S7**, Supporting Information). Higher hydroxyl groups in SiO₂-PECVD
ceramics have been reported to reduce their density and chemical stability, making the layer
more vulnerable to dissolution in wet environments⁵⁰.

Regarding the biocompatibility of the implanted ICs, no adverse inflammation or tissue
reaction was observed for all the animal models throughout the 12-month implantation,
despite the gradual dissolution of the SiN_x passivation on both IC groups (See **Figures S21**
**and S22**, Supporting Information).

In **Figure 4f**, we compare the dissolution rates of SiN_x passivation after exposure to various
aging environments. A comparison of the dissolution rates in PBS solution at 67 °C (120
627 nm/month for Chip-A and 22 nm/month for Chip-B) and *in vivo* at 37 °C (~125 nm/month for
Chip-A and ~54 nm/month for Chip-B) clearly illustrates the body's more aggressive
environment for silicon nitride dissolution compared to PBS at 67 °C.

Supporting experiments were conducted by aging some ICs in de-ionized (DI) water at
67 °C (see **Table S3**, Supporting Information). The lower dissolution rate of SiN_x in DI water
compared to PBS solution at the same temperature indicates that the dissolution process for
SiN_x is not entirely hydrolytic (water driven) but that the ions present in the solution (Na⁺, Cl⁻;
PO₄⁻) significantly affect the dissolution kinetics. Dissolution of thin-film ceramic materials
has been reported in previous literature^{40,52}, showing the chemical instability of these
PECVD-deposited ceramics when exposed to wet ionic environments.

When comparing the SiN_x dissolution rates on Chip-A and B, a higher dissolution rate for
Chip-A material is observed in all aging environments. This is despite the similar N/Si for
both chips (Chip-A and B both having a N: 50% and Si: 50%) as given in **Figure S7**,
Supporting Information. The higher SiN_x dissolution rate for Chip-A, could be due to the
slightly higher hydrogen (H) content in the layer (~20% for Chip-A and ~15% for Chip-B)^{53,54}.
Other factors such as the morphology (atomic arrangements) could also play a role in the
chemical stability when exposed to wet environments³⁹.

**Figure 4d** compares the dissolution rates of SiN_x passivation after exposure to various aging
environments, de-ionized (DI) water (done as Supplementary experiments as given in **Table**
**S3**, Supplementary Information) and PBS both at 67 °C, and rat at 37 °C. The higher
dissolution rate *in vivo* at 37 °C illustrates the body's more aggressive environment and
suggests that the dissolution is not entirely hydrolytic but that the ions, proteins and other
organic species in the body can impact the process. Dissolution of thin-film ceramic
materials has been reported in previous literature^{36,51}. Our nanometer surface analysis using
ToF-SIMS, however, revealed subtle differences in the 0 – 5 nm of the surface suggesting
that the different dissolution rates could be due to the type of oxide layer created on top of
the SiN_x upon its exposure to various media (Supplementary **Note 6**).

Comparing the SiN_x dissolution rates on Chip-A and B shows a higher dissolution rate for
Chip-A material despite the similar N/Si for both chips (Chip-A and B both having a N: 50%
and Si: 50% as given in Supplementary **Figure S7**). This data highlights how the details in
the manufacturing processes could impact the biostability of these layers. The higher rate for
Chip-A could be due to the slightly higher hydrogen (H) content in the layer (~20% for Chip-
A and ~15% for Chip-B)^{52,53}. Other factors, such as the morphology (atomic arrangements)
could also play a role in the chemical stability when exposed to wet environments⁵⁴.

Despite the gradual dissolution of the SiN_x passivation no adverse inflammation or tissue
reaction was observed for all the animal models throughout the 12-month implantation,
suggesting the biocompatibility of the ICs (See Supplementary **Figures S19-S20**).

Silicon-ICs in PBS solution at 67 °C

Silicon-ICs in rat

Figure 4. Stability of IC multilayer stack and top passivation layers after accelerated *in vitro* and *in vivo* aging. a-c) Representative cross-sectional SEM images of Chip-A samples after accelerated *in vitro* aging in PBS solution at 67 °C, a) Cross-sectional SEM image on the PDMS edge of an M3-IDC (Chip-A) structure after 3-months of aging in 67 °C PBS showing the IMD

material stack from surface to silicon substrate. Inset: optical micrograph of the area used for cross-section analysis. Right, EDX elemental analysis shows intact aluminum IDC metallization. **b)** Magnified cross-sectional SEM image at the PDMS-edge showing a thinning of exposed SiN_x passivation (~360 nm) after 3-months of aging in 67 °C PBS. **c)** Cross-sectional SEM image of a Chip-A sample after 10 months with complete loss of SiN_x passivation in the uncoated region, leaving the SiO₂ passivation exposed to the PBS solution. ~~**d-e) Representative optical micrograph and 3D AFM surface profiles of explanted Chip-A and B samples after PDMS decapsulation. All AFM profiles were taken within a 20 μm × 20 μm window around the former PDMS edge (red square). AFM profiles were taken after 7 and 12 months of implantation in rats.**~~ **fd)** Dissolution rates of exposed SiN_x passivation from Chip-A and B in different aging media (n≥2, with each sample measured on two different locations).

3.4.1.2. Ionic barrier properties of IC passivation layers

Next, we investigated ionic barrier properties of the SiN_x/SiO₂ passivation after their long-term exposure to physiological media.

The ingress of various cations (Na⁺, K⁺, Ca²⁺ and Mg²⁺) and anions (Cl⁻, PO₄³⁻, S²⁻) in the exposed regions of the ICs was measured by positive and negative mode ToF-SIMS depth profiling, respectively. Depth profiles were taken from the surface to ~200 nm within the SiO₂ passivation layer. **Figure 5** presents the positive mode depth profiles of Chip-A and B ICs, explanted after 7 and 12 months *in vivo*. ~~In the positive mode, [Si₂N⁺] and [SiO⁺] cluster ions were used to identify the SiN_x and SiO₂ passivation layers⁵⁶.~~ For the 7-month explanted ICs, ~ 100 nm and ~ 200 nm of the SiN_x passivation remained on Chip-A and Chip-B, respectively. ~~These results support the AFM measurements showing the dissolution and thinning of the SiN_x passivation when exposed to the body (Figure 4d-e) supporting the dissolution rates given in Figure 4.~~ Despite the dissolution and long-term exposure, no ingress of alkali metals is present in the remaining SiN_x layers, indicating that the SiN_x passivation from both IC foundries maintained its ionic barrier properties over the long-term exposure *in vivo*.

On the 12-month explanted ICs, no SiN_x was detected on either chip, leaving the SiO₂ passivation directly exposed to the body environment. Based on the estimated SiN_x dissolution rates in **Figure 4f**, after the complete SiN_x dissolution, the SiO₂ passivation is exposed to tissue for approximately 4 and 2 months for Chip-A and Chip-B samples, respectively. Interestingly, no ionic ingress was detected for the exposed SiO₂ layers from either IC foundry. Negative mode depth profiles also did not reveal any ingress of anions and do not contain additional information; thus, they are not presented.

These results indicate that both SiN_x and SiO₂ passivation layers from the two selected silicon-IC foundries are effective long-term barriers against ionic ingress from the body. Note that in some profiles, a slight increase in the aluminum (Al) intensity was found in the deeper depths of the SiO₂ as the profile was getting closer the IDC metallization. Closer to the surface or in the SiN_x, however, no aluminum was detected, indicating that the passivation layers act as a two-sided barrier, preventing ionic ingress into the chip, but also inhibiting the out-diffusion of IC metals into the body.

3.4.1.3. Chemical stability and moisture barrier properties of IC passivation layers (unbiased)

Next, we investigated the chemical stability and moisture barrier properties of the SiN_x and SiO₂ passivation layers on the directly exposed and PDMS-coated regions of the aged ICs. ~~It has been reported that moisture can dissociate the Si-N, Si-O and Si-Si bonds in silicon-based ceramics and create silanol groups (Si-OH) in the atomic structure^{50,54,56}. For IC's, these silanol groups can have detrimental effects as they could increase the dielectric constant of the insulating layers and lower their ionic barrier properties, facilitating ion penetration within the IC^{39,53}.~~

For this purpose, we used negative mode ToF-SIMS depth profiling to evaluate the chemical stability of the SiN_x and SiO₂ passivation layers. ~~The [SiN⁻] and [SiO₂⁻] cluster ions were used to evaluate the layers⁵⁵. The moisture barrier properties of the passivation layers were also analyzed by monitoring the intensity of the [OH⁻] cluster ion within the depth profiles⁵⁷⁻⁵⁹. In all cases, results on aged ICs were compared to depth profile results from reference ICs (as is from foundry).~~

Profiling was done at three different depths: SiN_x-surface (analyzed from 0 - 15 nm), SiN_x-bulk (analyzed from 50 - 100 nm within the SiN_x layer), and SiO₂-bulk (analyzed from 100 to 200 nm within the SiO₂ passivation layer). Surface analysis (0 - 15 nm) was conducted with sub-nanometre (0.139 nm) step sizes (See Experimental Section). All bulk profiles were collected using ~1 nm step sizes.

The schematic in **Figure 6** summarizes the findings from the depth profile analysis. In the exposed regions, a thickness increase in the surface oxidation of the SiN_x passivation is observed. Within the oxidized layer, a slight increase in ionic impurities is also seen. In the PDMS-coated region, on the other hand, only an increase in oxidation thickness was found with no increase in the ionic impurities.

Figures 6a-b presents the negative mode depth profiles of a representative Chip-A sample after 7 months of implantation in rats, comparing the first 15 nm of the SiN_x passivation in the PDMS-coated and exposed regions. In both regions, the first 5 nm demonstrated low intensities of [SiN⁻] with high intensities of [OH⁻] and [SiO₂⁻]. This pattern indicates a surface oxidation of the SiN_x, revealing an additional degradation mechanism affecting the SiN_x passivation on the ICs. In the exposed region, a slightly higher chlorine [Cl⁻] and sulphur [S⁻] intensity was also detected in the oxidised layer. These ions could be a result of biofluids and amino acids touching the oxidized layer and penetrating within the layer⁶⁰. In the PDMS-coated region, on the other hand, the oxidized SiN_x layer showed lower [Cl⁻] and [S⁻] intensities, similar to reference level ICs. ~~To compare the thickness of the oxidized layers between the two regions, the depth in which the [SiO₂⁻] intensity is higher than the [SiN⁻] intensity is used to identify the oxidized layer on the SiN_x passivation. Results show a slightly thicker oxidized layer in the exposed region of the IC.~~ Similar surface ToF-SIMS depth profile results for a 7-month explanted Chip-B can be found in **Figure S23**, Supporting Information.

Figure 6c compares the thicknesses of the oxidized SiN_x layer in the exposed and PDMS-coated regions for Chip-A and B after 7 months *in vivo*. The oxidized layers on Chip-B, for both regions, shows a thinner layer compared to Chip-A, suggesting the higher density and stability of the nitride passivation on Chip-B. **Figure 6d** shows the thickness of the oxidized SiN_x in the PDMS-coated regions over time. After 12 months of implantation in rats, ~ 7.8 nm (0.78% of total SiN_x thickness) and ~ 4.9 nm (0.75% of total SiN_x thickness) of the nitride passivation has been oxidized for the Chip-A and B ICs, respectively. Note that for reference

ICs (as is from foundry, 0-month), a ~3.5 - 4 nm oxide layer is already present on the surface
of the SiN_x passivation. Assuming a linear oxidation rate, a ~3.9 nm/year and ~1.4 nm/year
oxidation rate can be anticipated for PDMS protected Chip-A and B ICs when implanted to
the body.

**Figure 6e** compares the effect of the two aging environments (PBS solution at 67 °C and rat
at 37 °C) on the oxidation of the nitride passivation in the PDMS-coated region. After 12
781 months, ICs soaked in 67 °C PBS solution showed a thicker oxidized layer on the SiN_x.
These results indicate that the 67 °C accelerates the moisture diffusion and oxidation
process within the SiN_x passivation layer.

Within the SiN_x-bulk, the [OH⁻] and [SiN⁻] signals were evaluated by averaging the intensities
within the 50 - 100 nm depth of the SiN_x layer. **Figure 6f** presents the [OH⁻] intensities within
the SiN_x-bulk for exposed and PDMS-coated regions after 7 months of implantation in rat
showing slightly higher [OH⁻] in the exposed regions. In the PDMS-coated regions, the [OH⁻]
intensity was also evaluated over time (**Figure 6g**), indicating a gradual increase in the
intensity with time. When comparing the two aging environments at 12-month, a marked
increase in the [OH⁻] intensity is observed for the ICs soaked in PBS solution at 67 °C,
(**Figure 6h**), possibly resulting from the higher diffusion rate of moisture with temperature.

~~Figures 6i-k give the [SiN⁻] intensity in the bulk region, showing stable intensities for all
samples except those soaked in PBS solution at 67 °C. The decreased [SiN⁻] intensity
observed in samples soaked in PBS solution suggests that elevated temperatures
accelerate moisture diffusion and promote the breakage of Si-N bonds within the SiN_x
passivation layer.~~

~~Next, the SiO₂-bulk was examined by evaluating the [OH⁻] and [SiO₂⁻] intensities in the layer
(averaging the intensities between 100–200 nm). **Figures 6l-m** show the average
intensities for the exposed and PDMS-coated regions after 12 months of exposure to both
aging environments. On both Chip-A and B samples, a stable [OH⁻] intensity is seen in the
SiO₂-layer for the PDMS-coated regions. A slight increase in [OH⁻] intensity is recorded for
the exposed areas for Chip-B. Note that the SiO₂ passivation can be directly exposed to the
aging environments (either PBS or body) for the exposed (uncoated) regions after the
complete dissolution of the nitride passivation. In the PDMS-coated regions, however, the
SiO₂ is always protected by both PDMS and the top SiN_x passivation layer.~~

~~The [SiO₂⁻] intensity within the bulk SiO₂-layer also revealed stable intensities for both
exposed and PDMS-coated regions after 12 months of aging, suggesting the chemical
stability of the bulk SiO₂ passivation layer on both IC foundries.~~

~~The [SiN⁻] intensity in the bulk region showed stable intensities for all samples except those
soaked in PBS solution at 67 °C, where a slight decrease was observed (Supplementary
**Note 7**). This decrease suggests that elevated temperatures accelerate moisture diffusion
and promote the breakage of Si-N bonds within the SiN_x passivation layer. The SiO₂
passivation was similarly investigated, showing the film's high stability and moisture barrier
properties (Supplementary **Note 7**).~~

from 0 - 15 nm of the PDMS-coated and exposed regions of a 7-month explanted Chip-A sample (acquired with 0.1 nm step-size). Surface depth profiles show oxidation of the SiN_x passivation with higher chlorine (Cl) and sulfur (S) impurity ions in the oxidized layer for the exposed region. **c - e**) Comparing the thicknesses of the oxidized SiN_x at different time points and aging media (in 7m or 12m, 'm' denotes month). **c**) Comparing the thicknesses in the PDMS-coated and exposed regions after 7 months *in vivo*, **d**) Comparing the thicknesses in the PDMS-coated regions after 7 and 12 months *in vivo* with that of a dry reference sample, and **e**) Comparing the thicknesses in the PDMS-coated regions after 12 months exposure to accelerated *in vitro* and *in vivo* environment. **f-n)** average [SiN] and [OH] intensities within the SiN_x passivation bulk (averaged ToF-SIMS depth profile data from 50 - 100 nm). ~~**n-o)** average [OH] and [SiO₂] intensities within the SiO₂ passivation bulk (averaged ToF-SIMS depth profile data from 100–200 nm). Note that in the uncoated region, at 12 months, the SiO₂ passivation is directly exposed to the body environment. Data is presented as average of 4 measurement results, performed on n=2 samples with 2 measurements per sample.~~

3.4.2. Effect of electrical field: DC electrical bias in PBS solution at 67 °C

Given that implantable ICs will be subjected to electrical bias voltages during their lifetime operation, as our final investigation, we evaluated the long-term effect of electric biasing on the stability of the IC materials and the PDMS-interfaces. ~~Note that a group of biased IDC structures experienced short term (1–2 months) failures (See Table S6, Supporting Information). For all these samples, the top metallization of the IC (M4 for Chip-A or M6 or Chip-B) was used which resulted in poor conformality or stress-induced cracking in the passivation. For investigating the longer term effects of electrical biasing, M3-IDC (Chip-A) and M5-IDC (Chip-B) test structures were used.~~

Figure 7a gives a schematic representation of a partially PDMS-coated IC applied to electrical bias (between the IDC metallization and solution) in PBS solution at 67 °C. During biasing, both combs of the IDC were connected to the negative (-) potential and +5 V or +15 V DC was connected to a stainless-steel rod in the PBS solution. The polarity was chosen to drive the alkaline ions (Na⁺ and K⁺) towards the IC passivation. The right schematic in **Figure 7a** gives an overview of the findings demonstrating the effect of the continuous biasing on the exposed and PDMS-coated regions of the chip.

~~Microscopically, structures continuously biased with 15 V DC voltage (n=6 from Chip-A and n=10 from Chip-B) showed signs of discoloration across their surface in the exposed region, as demonstrated in a representative M3-IDC structure from Chip-A (arrows in **Figure 7b**). In contrast, areas protected by PDMS showed no discoloration. Note that despite the discoloration in the exposed regions, stable capacitive behavior was recorded over the 12-month aging duration when measured between the comb structures and PBS (**Figure S8**, Supporting Information).~~

~~**Figure 7c** presents a cross-sectional SEM image taken at the PDMS-edge. In both the PDMS-coated and exposed regions, intact aluminum metallization was observed with no noticeable difference in the buried IC's multilayer stack. However, the exposed region showed the presence of a thinned SiN_x passivation layer (~0.33 μm). This contrasts with the unbiased Chip-A sample where complete dissolution of the SiN_x was observed after ~10 months of direct exposure to PBS solution at 67 °C (**Figure 4c**). **Figure 7c** also reveals a different contrast between the thinned SiN_x layer in the exposed region and the PDMS-protected region. The PDMS protected region showed the expected thickness for the SiN_x passivation (1 μm).~~

Negative and positive mode ToF-SIMS depth profiling was used to evaluate the chemical stability and barrier properties of the exposed and PDMS-coated regions of the ICs. For analysis, M3-IDC (n=2, Chip-A) and M5-IDC (n=2, Chip-B) test structures were used which were continuously biased for 12 months with 15 V DC in PBS solution at 67 °C. **Figure 7d** **7b** presents the depth profiling results for the exposed region on a representative M3-IDC

structure (Chip-A). Negative mode depth profile results revealed a ~120 nm oxidized layer
on top of a 100 nm SiN_x layer. Within the oxide layer, high intensities of [SiO₂⁻], [OH⁻], and
carbon [C⁻] were found. Positive mode depth profiles also revealed high intensities of [Na⁺]
and [K⁺] ions in the top ~120 nm oxidized layer. Beyond the 120 nm depth, once the profile
enters the remaining ~100 nm SiN_x layer, a marked drop in the impurity ions was observed,
demonstrating that the remaining SiN_x still acts as an ion barrier.

XPS depth profiling was used to quantify the chemical composition and ionic impurity levels
within the oxidized layers (See **Figure S24**, Supporting Information). The chemical
composition of the oxidized SiN_x layer indicated a slightly off-stoichiometric oxide film for
Chip-A (Si: 28%, O: 58%) and Chip-B (Si: 28%, O: 62%) samples. For the ionic impurities,
despite the high intensities detected by ToF-SIMS, XPS found a 2 - 3% sodium (Na)
incorporation in the oxidized layer for both Chip-A and B samples. Note that the sensitivity
levels of the ToF-SIMS measurement to Na⁺ and K⁺ ions can be 3 to 5 orders of magnitude
higher than XPS (See Experimental Section).

In comparison to the exposed region, depth profiling on the PDMS-coated region of the
biased IDC structures showed a stable [SiN⁻] profile within the entire ~ 1 μm SiN_x passivation
layer (**Figure 7e 7c**). A thin (~12 nm) oxide layer was observed on top of the SiN_x layer with
no ionic penetration. The absence of ionic penetration in the oxidized layer is due to the ionic
barrier properties of the PDMS coating⁶¹, in this case, even in the presence of an electrical
field.

~~As explained the previous section, due to PDMS's inherent moisture permeability, a gradual~~
~~oxidation of the SiN_x passivation can be expected in the PDMS-coated regions of the IC.~~ To
examine if the applied electrical fields affected the oxidation rate of the SiN_x, the thicknesses
of the oxidized layer in the PDMS-coated regions were measured for a group of samples
which were biased with a 5 V (n=2) or 15 V (n=2) DC voltage (between IDC and PBS) during
the 12-month aging study in PBS solution at 67 °C (**Figure 7f 7e**). At 5 V DC (n=2 from each
chip), no significant change in oxide thickness was measured compared to unbiased IDC
structures. However, at 15 V DC (n=2 from each chip), both Chip-A and Chip-B samples
showed a slightly thicker oxidized layer, suggesting that higher electrical fields may enhance
the oxidation rate of SiN_x in the PDMS-coated region.

**Figures 7g-h 7e-f** gives the average [OH⁻] intensity in the bulk (averaging the intensities
between 100 - 200 nm) of the SiN_x and SiO₂ passivation layers. In the SiN_x layer, higher
[OH⁻] intensities were observed on samples subjected to continuous 15 V DC voltages as
compared to 5 V and unbiased chips (**Figure 7g 7e**). The [OH⁻] incorporation may result
from the higher applied voltage, as higher electrical fields may introduce strain in the
molecular structure of the dielectric⁶², making the structure more susceptible to moisture
attack. It should be noted that the increased [OH⁻] incorporation in the SiN_x layer was
exclusively detected using ToF-SIMS due to its enhanced sensitivity. XPS depth profile
results in the SiN_x passivation revealed no oxygen and a N/Si ratio of ~ 1 (similar to
reference, as is from foundry ICs) for both Chip-A and Chip-B samples (**Figure S24**,
Supporting Information). The SiO₂ passivation was similarly investigated (**Figure 7i 7f**),
where a slightly higher [OH⁻] intensity was observed for devices under 15 V continuous DC
biasing.

Figure 7. Chemical stability and moisture and ion barrier properties of PDMS-coated and exposed passivation layers after continuous DC biasing in PBS solution at 67 °C. a) Left schematic: biasing configuration with DC voltage applied between the IDC metallization (combs) and an electrode in PBS solution. Right schematic: the effect of the continuous biasing on passivation. **b)** ~~Top view optical micrographs of a representative M3-IDC structure (Chip-A) sample after 12 months of~~

continuous 15 V biasing between M3-IDC and PBS showing the color change in the exposed region. Right image after PDMS decapsulation. **c)** Cross-sectional SEM image of the Chip-A sample at the PDMS edge. **d-e)** Positive and negative mode ToF-SIMS depth profiles of the exposed and PDMS-coated regions. **f)** Thickness of the oxidized SiN_x surface for Chip-A and Chip-B ICs in the PDMS-coated region after continuous biasing at five and 15V between IDC and PBS. For Chip-B samples, electrical biasing was applied between the M5-IDC structure (negative polarity) and PBS (positive polarity). **g-h)** Average [OH⁻] intensities in the bulk of the SiN_x and SiO₂ passivation layers (averaged from 100–200 nm within the layer) in the PDMS-coated region after continuous electrical biasing with 5 or 15 V-DC.

b-c) Positive and negative mode ToF-SIMS depth profiles of the exposed and PDMS-coated regions showing thinning and oxidation of the SiN_x passivation. Ion ingress in the top ~120 nm oxidized layer is detected. PDMS coated region shows a 12 nm oxide layer with no ion penetration. **e)** Thickness of the oxidized SiN_x surface for Chip-A and Chip-B ICs in the PDMS-coated region after continuous biasing at five and 15V between IDC and PBS. **f-g)** Average [OH⁻] intensities in the bulk of the SiN_x and SiO₂ passivation layers (averaged from 100 - 200 nm within the layer) in the PDMS-coated region after continuous electrical biasing with 5 or 15 V DC.

3. Discussion and guidelines for longevity

In this work, we intended to determine the functional lifetime of bare and PDMS-coated ICs in physiological media by testing devices until failure. However, through the 12-month accelerated *in vitro* study, very few IC failures were recorded on the n=62 tested chips. Therefore, complementary material evaluation techniques were performed on the *in vitro* and *in vivo* aged ICs to detect alternative degradation pathways that could help estimate the longevity of ICs in the body.

In the exposed bare die regions, we demonstrated the long-term moisture and ionic barrier properties of the SiN_x and SiO₂ passivation layers from both CMOS IC foundries. Nevertheless, as bare die, the dissolution of the SiN_x passivation was observed to be the primary degradation pathway *in vivo* (dissolution rate of ~125 and ~54 nm/month for Chip-A and B, respectively). Despite the nitride dissolution, the SiO₂ passivation underneath also exhibited excellent ion and moisture barrier properties, suggesting that uncoated/unprotected ICs can maintain their functionality in the body for at least 12 months. Note that this is only when the top metallization is not used. When the top metal is used, due to the poor conformality of the PECVD passivation layers, the top metal could be exposed to tissue within 6-7 months of implantation, causing metal corrosion (see cross-sectional SEM images in **Figure S25**, Supporting Information). Besides device malfunction, this would pose toxicity and safety risk as it would expose the body to electrical DC voltages. Therefore, for applications requiring the use of the top metal layer of the IC (nominal or thick top metal), applying a PDMS coating or a thin conformal coating is advised^{6,18}.

In the PDMS-coated regions, gradual surface oxidation of the SiN_x passivation was observed after 12 months *in vivo* (oxidation rate of ~3.9 and 1.4 nm/year for Chip-A and B, respectively). With this rate, it would take decades for the entire SiN_x passivation to oxidize. Nevertheless, in the case of total oxidation of the entire nitride layer, the SiO₂ passivation underneath would still serve as the second barrier protecting the underlying circuits. Taking a conservative estimate and considering the oxidation of the SiN_x to be the main degradation path for PDMS-coated ICs, we believe PDMS-coated ICs have the potential of reaching decades of functional lifetime in the body.

At first, it may seem unlikely that PDMS, a polymer with low moisture barrier properties, would be suitable for the long-term protection of ICs. However, in this paper it was demonstrated that the IC's passivation layers can have excellent moisture barrier properties. Therefore, other material characteristics of PDMS are key in achieving a long lifetime: 1) Low stress/strain ratio. Moisture-induced expansion of the polymer packaging may introduce stress on the IC's ceramic layers^{39,63-65}. Shear stress cracking of the IC's passivation has

950 been reported to be one of the main failure mechanisms for epoxy packaged chips^{37,54}.
PDMS's soft and compliant properties make it an ideal polymer for packaging implantable
ICs. 2) Long-term biostability. In addition to its optimal biocompatibility, PDMS is one of the
most biostable, medically used polymers⁶⁶. *In vivo*, degradation of other medical grade
polymers, e.g., parylene-C, has been reported through processes such as cracking^{67,68}. 3)
Long-term underwater adhesion to the IC's surfaces, especially to the nitride passivation.
Other polymers like epoxy and polyimide, typically used for IC packaging, have been
reported to have weak bonds to IC passivation when exposed to moist environments³⁷.

Miniaturizing implantable devices necessitates novel packaging materials and processes.
PDMS, despite its many favorable properties, has rarely been used as a long-term
standalone microelectronic packaging material due to its high moisture permeability. To
address the moisture challenge, in recent years various advanced thin-film barrier materials
and coatings have been proposed^{18-21,58}. However, packaging that depends on highly
specific materials and processes available only to a few academic labs will not easily find its
path to translation. For that, standardized materials and processes accessible to all must be
used. To this end, we have consciously chosen to rely on the barrier materials of
commercially fabricated ICs and use PDMS as an easily accessible standalone packaging
material. For a successful PDMS packaging, all components and interfaces underneath must
be moisture-resistant with good adhesion to the polymer⁴³. Our long-term electrical and
material evaluations suggest that ICs can be inherently hermetic, likely due to the advanced
processes and high-quality materials used by semiconductor foundries. Additionally, the
large sample size tested here (n=62 electrically tested) and the reproducibility of the results
underscore the consistency and reliability of IC manufacturing which has always been the
hallmark of the semiconductor industry.

By comparing the exposed bare-die and PDMS-coated regions of the IC structures, we
demonstrated how PDMS can effectively eliminate or retard the observed degradations seen
on the exposed regions of the die. Taking a conservative estimate and considering the
oxidation of the SiN_x to be the main degradation path, we believe PDMS-coated ICs have
the potential to reach decades of functional lifetime in the body (based on the observed
oxidation rates of ~3.9 and 1.4 nm/year for Chip-A and B, respectively).

For applications using bare IC such as neural probes, we showed the dissolution of the SiN_x
passivation to be one of the primary degradation pathways *in vivo* (dissolution rate of ~125
and ~54 nm/month for Chip-A and B, respectively). Despite the nitride dissolution, the SiO₂
passivation underneath also exhibited excellent ion and moisture barrier properties,
suggesting that uncoated/unprotected ICs can maintain their functionality in the body for at
least 12 months. Note that this is only when the top metallization is not used. When the top
metal is used, due to the poor conformality of the PECVD passivation layers, the top metal
could be exposed to tissue within 6-7 months of implantation, causing metal corrosion (see
cross-sectional SEM images in Supplementary **Figure S25**). Besides device malfunction,
this would pose toxicity and safety risks, as it would expose the body to electrical DC
voltages. Therefore, for applications requiring the use of the top metal layer of the IC
(nominal or thick top metal), applying a PDMS coating or a thin conformal coating is
advised^{6,18}.

In the accelerated *in vitro* study, corrosion of interconnect wire bonds was found to cause
failure for some of the devices. Corrosion is believed to be driven by moisture and galvanic
reactions between the gold wire bond and aluminum pads, causing the corrosion of the pad.
Non-bonded aluminum pads showed no signs of corrosion (**Figure S9**, Supporting
Information), mainly due to a native oxide layer on the aluminum, which has been reported to
create strong interface adhesion bonds with PDMS⁶³. As mentioned earlier given PDMS's
moisture permeability, ~~dissimilar metals should be avoided in the interconnect regions~~ all
components and interfaces underneath should be moisture resistant. For this purpose, we

propose using aluminum wire bonds instead of gold as it can prevent galvanic reactions and
result in a stronger adhesion between the wire bonds and the PDMS. In the case of wire-
bond interconnects, dissimilar metals should be avoided. A potential mitigation strategy
involves using aluminum wire bonds instead of gold, which could prevent galvanic reactions
and concurrently result in stronger adhesion with PDMS.

For emerging implantable applications incorporating embedded with an IC, selecting a
CMOS process suitable for chronic implantation requires extensive and long-term testing,
often not within the scope of academic research groups. Therefore, we encourage
researchers to use the technology nodes evaluated in this work when designing their ICs. It
should be highlighted though that not all foundry-fabricated ICs will have the same
hermeticity. In smaller CMOS technology nodes (90 nm and smaller), significant changes
have been made in the IC's materials. Most importantly, copper has replaced aluminum
metallization to reduce resistance and crosstalk. For the IMD layers, new porous dielectric
materials such as silicon oxycarbide (SiOC) have been adopted^{35,36}. Various investigations
have reported the instability of these porous dielectrics in the presence of moisture^{50,69,70}.
Therefore, if smaller technology nodes are to be used for implantable applications,
accelerated testing in combination with the methodology used in this work can be a valuable
tool in evaluating the stability of the IC material. As an example for the IMD layers, new
porous dielectric materials such as silicon oxycarbide (SiOC) are replacing the conventional
SiO_x and have been reported to have stability issues in the presence of moisture^{50,60,61}.
Therefore, if smaller technology nodes are to be used, the top shield metal and extra wall-of-
vias (WoV) around the IC may increase the chip's hermeticity and longevity by creating a
metal cage around the entire IC structure. In addition, a simplified version of the dielectric
sensor proposed in this study could be employed to detect early moisture ingress.

Accelerated *in vitro* testing in PBS solution has been extensively investigated as a model to
estimate the longevity of novel implantable devices^{18,34,67,71}. In section 2.3.1.2, we show that
in the PDMS-coated regions, elevated temperatures could be used as a model to accelerate
moisture penetration and degradation. However, for the exposed bare die regions of the
tested ICs, it was found that PBS solution is not an ideal model to mimic the complex *in vivo*
environment as a ~20-times higher nitride dissolution rate was observed in the body
environment compared to the estimated dissolution rate in PBS solution at 37 °C⁴⁵
(assuming an Arrhenius acceleration factor from 37 °C to 67 °C). Higher *in vivo* dissolution
rates for silicon-based dielectrics have been previously reported^{41,42,72}. In this work, using
ToF-SIMS analysis, we compared the chemistry of the first few nanometers (0–5 nm) of the
dissolving SiN_x layers, revealing slight differences in their chemistry after exposure to
different aging media (PBS at 67 °C and rat). All aged ICs created an off-stoichiometric
oxidized (SiO_x) layer with high [OH⁻] on top of the nitride passivation. ICs exposed to PBS
solution created a slightly thicker oxidized layer with higher [SiO₂⁻] and lower [OH⁻] intensity
compared to the ICs exposed to the body. Si-O bonds are thermodynamically more stable
than Si-N bonds. Therefore, the created oxidized layer on top of the nitride passivation can
act as a protective layer preventing or delaying attack by water molecules^{53,54}. The higher *in*
*in vivo* nitride dissolution rate could be due to the thinner top oxide layer making the Si-N
bonds more readily available for attack and dissolution. The theory of nitride dissolution rate
relying on the off-stoichiometric oxidation reactions has been proposed before⁷³ and has
been experimentally suggested in this investigation. The thinner oxidized layer on the SiN_x
passivation *in vivo* could be due to a similar mechanism observed for the corrosion of
titanium metal implants⁷⁴, where the body's proteins and enzymes covering the implant
inhibit the surface oxidation of the metal, allowing more bulk metal to be available for
corrosion.

In this work, the 12-month accelerated aging study found no change in the electrical
performance of both metal-shielded and unshielded test structures. For more extended

applications, however, we believe using a top shield and extra wall of vias (WoV) around the
IC may increase the chip's longevity by creating a metal cage around the entire IC structure.
These metal barriers could be explored to extend the chip's longevity, especially for
implantable ICs fabricated using smaller CMOS technology nodes incorporating porous
dielectric layers. In addition, a simplified version of the dielectric sensor used in this study
could be employed for implantable ICs fabricated in smaller CMOS technology nodes to
detect early moisture ingress.

As demonstrated in this work, the longevity of implantable ICs significantly relies on the
passivation layers. Any stress-induced cracking during packaging or post-processing could
undermine the protection offered by these layers. Die thinning, for example, has been
explored as a technique to enable die-embedded flexible bioelectronics^{9,75,76}. This technique,
however, may introduce stress and cracks in the passivation and dielectrics layers.
Moreover, as the silicon substrate becomes thinner, moisture penetration through this route
may need further research. Therefore, as a future investigation, one can investigate how thin
the ICs can be made without compromising the barrier properties of the IC's passivation
layers and silicon substrate.

Stress-induced cracking during packaging or post-processing could undermine the
hermeticity of the IC die. Die thinning, for example, has been explored as a technique to
enable die-embedded flexible bioelectronics^{9,62,63}. This technique, however, may introduce
stress and cracks in the passivation and dielectric layers. Moreover, as the silicon substrate
becomes thinner, moisture penetration through this route may need further research.
Therefore, as a future investigation, and by incorporating the dielectric sensor proposed
here, one can investigate how thin the ICs can be made without compromising the
hermeticity of the die.

The effect of continuous electric fields on the stability of IC materials was investigated by
analyzing a group of samples applied to DC bias voltages between the IDC metals and the
PBS solution. Using ToF-SIMS analysis, we identified enhanced [OH] incorporation in the
SiN_x and SiO₂ passivation layers for the ICs exposed to 15 V DC in comparison to unbiased
ICs. For lower voltages typically used for implantable ICs (5 V), the applied bias was found
to have no long-term effect on the stability of the passivation layers. Based on these insights,
for implantable ICs operating at higher voltages (>15 V DC), further investigations may be
required to determine the long-term stability of the materials and interfaces.
In our complementary investigation⁷⁷, the usefulness of these guidelines is validated by
demonstrating ultra-long lifetimes for PDMS-coated ICs.

**4.—Conclusion**

Ensuring the longevity of the IC in the body will be a key challenge when developing next
generation miniaturized active neural implants. The work presented here demonstrated that
the advanced materials applied for IC fabrication may be used as long-term protective
barriers when implanting ICs in the body. Evaluation results on ICs fabricated by two CMOS
foundries demonstrated the excellent moisture and ion barrier properties of the SiN_x/SiO₂
passivation layers, maintaining the IC's electrical functionality for at least a year, even when
exposed to PBS solution as bare die. Exposure of the IC to the physiological environments,
however, resulted in the degradation of the IC material, with the most prominent degradation
being the dissolution of the nitride passivation. PDMS coating of the ICs, on the other hand,
was shown to prevent the observed degradation, making it a promising soft biocompatible
encapsulant for multi-year implantations. The authors believe the insights collected in this
work are critical to the next step in engineering long-lasting active neural implants.

5. Experimental Section

[revised manuscript text omitted]

1309 is from foundry) samples.

Reviewer response key:

- Reviewer comments in blue
- Response to reviewer comments in black
- Changes to the manuscript in red

Reviewer 1 (comments to the Authors):

Comment 1.1:

I would like to thank the authors for their effort in revising the manuscript, which has significantly improved its clarity and readability. However, I still have some major concerns and comments that I would like the authors to address.

One significant caveat of the study is that in all the in vitro and in vivo tests, the ICs were largely free from mechanical stress. This does not accurately reflect the continuous mechanical stress the implants would experience during real-world applications, particularly noting that the ICs were not connected in the in vivo tests.

Response 1.1:

We thank the reviewer for their positive feedback on the improved clarity and readability. We also appreciate the comments in the second round of review, which we believe has greatly helped improve the quality of the manuscript. We have carefully read and addressed the remaining comments in the detailed response below.

The reviewer is correct in that no mechanical loading was applied to the samples in our in vitro tests. However, it should be noted that besides mechanical loading, the PDMS-device interfaces will also be impacted by hydrothermal stresses. In fact, hydrothermal stresses (moisture in combination with temperature) have been shown to degrade PDMS-device interface bonds with degradation rates which have been shown to follow the Arrhenius law [1]. Therefore, given that all samples in the in vitro study were soaked at an elevated temperature (67 °C), the interface bonds were chemically stressed. Moreover, the PDMS, due to its higher coefficient of thermal expansion (CTE) compared to the silicon die (>300 ppm for PDMS and ~3 ppm for silicon [2]), will experience more physical expansion at the tested elevated temperature causing more stress at the PDMS-device interface compared to the body temperature.

Therefore, we argue that although the in vitro samples were not subjected to cyclic mechanical stresses, they were subjected to hydrothermal stresses that would stress and degrade the bonds.

Regarding the samples in the 1-year in vivo animal study, as all devices were subcutaneously implanted, therefore, they were subjected to some form of mechanical stress while the animals were in motion, although this was not quantified.

Also, it should be clarified that in this study, ICs were partially coated with a ~200 - 700 µm PDMS layer, partially leaving the IC surface exposed. In real-world applications, the ICs, together with the metal interconnects, will be fully encapsulated with a thicker (a few millimeters) PDMS layer, which will act as a mechanical buffer, damping the externally applied mechanical stresses.

Nevertheless, we do acknowledge that the mechanical stresses applied to each device will vary based on their shape, size and location of implantation in the body. Modeling the mechanical loading conditions which the device would experience at different implantation sites was beyond the scope of this work. To do a more accurate predication on the longevity of the implant, the mechanical stress in the real-world conditions should be quantified and included in the accelerated in vitro.

As mechanical stresses and bonding strength are critical components to the reliability of PDMS-packaged implants, we have included a paragraph in the discussion section of the manuscript.

References:

[1] Lonys, Laurent, Anne Vanhoestenbergh, Nicolas Julémont, Stéphane Godet, Marie-Paule Delplancke, Pierre Mathys, and Antoine Nonclercq. "Silicone rubber encapsulation for an endoscopically implantable gastrostimulator." *Medical & biological engineering & computing* 53 (2015): 319-329.

[2] Ardebili, Haleh, Jiawei Zhang, and Michael G. Pecht. *Encapsulation technologies for electronic applications*. William Andrew, 2018.

Our modifications to the manuscript: Based on the reviewer's feedback and recommendations, we have included the paragraphs below in the Discussion and Supplementary Information section.

Manuscript: **added paragraph below in Discussion section**

Besides die hermeticity, as explained in section 2.2 and Supplementary Note 1, a successful PDMS packaging also requires strong and long-lasting PDMS adhesion to the underlying materials. During implantation, various environmental factors, such as body fluids and mechanical stress, can degrade and weaken these interfacial bonds [61]. In this study, qualitative evaluations of PDMS adhesion on both accelerated in vitro and in vivo samples showed no separation between the PDMS and IC surface after aging (see Supplementary Note 8). However, it should be noted that the mechanical stresses applied to each implant may vary depending on factors such as shape, size, and location of implantation [62, 63]. Therefore, for a more accurate prediction of implant longevity, the mechanical stresses expected in real-use conditions should be modeled and replicated in the accelerated in vitro study.

Added references:

[61] Niemiec, M. & Kim, K. Lifetime engineering of bioelectronic implants with mechanically reliable thin film encapsulations. *Progress in Biomedical Engineering* vol. 1 Preprint at <https://doi.org/10.1088/2516-1091/ad0b19> (2024).

[62] Sharma, U. et al. The development of bioresorbable composite polymeric implants with high mechanical strength. *Nat Mater* 17, (2018).

[63] Choi, Y. S. et al. Stretchable, dynamic covalent polymers for soft, long-lived bioresorbable electronic stimulators designed to facilitate neuromuscular regeneration. Nat Commun 11, (2020).

Supplementary Information: added **Supplementary Note 8**

Supplementary Note 8: As explained in Supplementary Note 1, adhesion between the PDMS and IC surface is crucial for preventing lateral fluid ingress and moisture condensation that could lead to shunt leakage paths. To qualitatively assess adhesion strength and detect any potential debonding after in vitro and in vivo aging, a shear force was applied to the PDMS edge on the 12-month aged IC structures (see figure below). This was performed by using tweezers to apply pressure on the PDMS while microscopically inspecting for any detachment from the IC surface. No separation of PDMS from the IC was observed after aging, even under significant applied pressure. Instead, the shear force primarily led to cohesive breakage within the PDMS.

It should be noted that in the body, PDMS interfaces are subjected to various stress factors, including moisture (which permeates through the PDMS) and mechanical loading. In the accelerated in vitro study, no mechanical loading was applied to the samples. However, PDMS-device interfaces are also affected by hydrothermal stresses, where the combination of moisture and temperature can accelerate bond degradation. Previous studies have shown that hydrothermal stress degrades PDMS-device interface bonds at rates that follow the Arrhenius law. Therefore, in the in vitro study, samples soaked at an elevated temperature (67 °C) experienced chemical stress on their interface bonds for approximately eight times longer than they would at body temperature.

Comment 1.2:

While the authors emphasize the critical nature of interfacial adhesion between the PDMS and the passivation layer, they do not provide bonding strength tests conducted during aging. This is a critical piece of data that should be included in the manuscript. Fundamentally, if the bonding of PDMS to the IC surface has significantly weakened, and the PDMS merely sits on top of the passivation layer after some time in the aging test or in vivo implantation, it may still act as a barrier to slow down the etching of SiNx under the mechanical stress-free conditions presented in this manuscript. However, it is unlikely to withstand real-world testing where mechanical stress between the device and PDMS is present. Providing convincing evidence that the PDMS-device bonding remains intact during long-term aging or implantation is essential for establishing PDMS as an effective barrier for implanted devices and for preventing any potential misinterpretation in the field.

Response 1.2:

We thank the reviewer for this detailed and valid comment. Due to the mm size of the IC structures used in this study, measuring and quantifying the bonding strength using standardized methods as described in [2] was not feasible.

For this reason, we used methods to qualitatively evaluate the bonding strength:

1. DC biasing and electrical measurements: as shown in Supplementary Note 1, PDMS debonding would allow the capillary forces of water to create a path for ionic liquids to reach the DC-biased wire-bonds (see added schematics in Supplementary Note 1). Thus, the stable electrical performance rules out potential debonding.

2. Shear force to mechanically remove PDMS from the IC surface: As shown in the example figure in Response 1, after both in vitro and in vivo aging, a shear force was applied to the PDMS-edge on all IC structures to evaluate the possibility of any debonding. This was done by using a tweezer and by mechanically pressing on the PDMS to remove the material from the IC surface. No separation of PDMS from the IC was observed after both accelerated in vitro and in vivo ageing, even when pressing hard.

The shear force mainly resulted in damage to the PDMS. Due to the strong adhesion, to decapsulate the IC structures post-ageing, a scalpel and PDMS solvent had to be used to cut and chemically remove the PDMS from the ICs (see Experimental Section).

The results of the second test were not reported in the earlier versions of the manuscript, as the data was merely quantitative. However, as the reviewer has also noted the importance of PDMS-device adhesion, we have included a paragraph in the Discussion section of the manuscript and included the explanatory figure in the Supplementary Information.

Our modifications to the manuscript: Based on the reviewer's feedback and recommendations, **Supplementary Note 8** has been added in the Supplementary information.

Comment 1.3:

I appreciate Supplementary Note 1, where the author highlights the interface between the PDMS and the IC surface. However, since no mechanical stress was applied and no evaluation of the bonding strength was conducted, this significant concern of degradation of bonding strength cannot be dismissed.

Response 1.3:

We have addressed this comment of the reviewer in Responses 1.1 and 1.2.

Comment 1.4:

A particularly misleading term used in the manuscript is 'Hermeticity/Hermetic' in the title and throughout the text. Hermetic packaging typically refers to air- and water-tight barriers, while the authors acknowledge that PDMS is a moisture-permeable material, which, in my opinion, renders it non-hermetic in nature. I recommend removing 'hermetic' from the title and abstract, and clearly distinguishing PDMS packaging from hermetic packaging throughout the manuscript.

Response 1.4:

We thank the reviewer for raising this point. As the reviewer has mentioned, PDMS packaging is indeed non-hermetic in nature when compared to conventional hermetic enclosures using metals or ceramics. However, in this manuscript, when using the term hermetic, we are not referring to the PDMS packaging but to the silicon-IC structure itself. In fact, as it has been highlighted in the Discussion section of the revised manuscript, as PDMS packaging is non-hermetic in nature (moisture permeable), for it to be a successful packaging solution, the structures underneath have to be inherently hermetic and moisture resistant themselves, i.e., unaffected by functioning in an environment saturated with moisture. Evaluating the inherent hermicity of IC structures has been the primary focus of this work. Nevertheless, based on the reviewer's feedback, and for further clarification, we have made the modifications below to the manuscript.

Our modifications to the manuscript: for clarification the following changes have been made to the manuscript.

Title:

- On the Longevity and **Inherent Hermeticity** of Silicon-ICs: Evaluation of Bare-Die and PDMS-Coated ICs After Accelerated Aging and Implantation Studies

Abstract:

- ~~Here, we evaluate if the advanced materials utilized by the IC foundries can function as effective long-term barriers, protecting the IC in the body.~~ Here, we evaluate the inherent hermeticity of bare die ICs, and examine the potential of PDMS, a moisture-permeable elastomer, as a standalone encapsulation material.
- ~~Results demonstrated stable electrical performance, suggesting the inherent hermeticity of bare IC structures.~~ Results revealed stable electrical performance, indicating the unaffected operation of ICs even when directly exposed to physiological fluids.

Manuscript:

- **Introduction:** Limited data, however, exists on the long-term stability and inherent hermeticity (i.e. the ability of the structure itself to prevent moisture ingress) of actual foundry-fabricated ICs upon their direct exposure to aggressive physiological environments.
- **Introduction:** This property makes PDMS packaging non-hermetic in comparison to conventional techniques using metal or ceramic enclosures.
- **Discussion:** Our long-term electrical and material evaluations suggest that IC structures can be inherently hermetic themselves, preventing the outer moisture and ions reaching the sensitive structures within.

Comment 1.5:

Another limitation of the PDMS packaging approach is its reliance on defect-free SiNx as the passivation layer, which not only provides an excellent moisture and ionic barrier for the active component but also facilitates strong interfacial bonding to PDMS when properly surface treated. However, this strategy may not be widely applicable, as many biological implants feature a metal top layer or vias without the protection of a passivation layer. ICs with these exposed metal components were not tested, and the strong bonding between metal and PDMS can be challenging, raising concerns about the adaptability of this approach in diverse applications. This limitation should be articulated in the discussion session.

Response 1.5:

We thank the reviewer for raising this point but would like to clarify a possible misunderstanding. All the IC structures investigated in this work included exposed aluminum IC metal pads. In the accelerated in vitro test, the IC pads were gold wire-bonded and electrically biased. In fact, we have highlighted the different PDMS-metal interfaces in Supplementary **Note 1**. In **Figures S9** and **S16** (Supplementary Information), we have qualitatively examined the adhesion of PDMS to aluminum pads by optical microscopy. No corrosion of the non-wire-bonded pads was observed in both the in vitro and in vivo aged samples, suggesting stable adhesion between PDMS and aluminum. It should be noted that PDMS-aluminum adhesion has already been investigated and confirmed in previous studies [3].

In the Discussion section, we have also explained the wire-bond failures which were because of the galvanic corrosion between the gold and aluminum interfaces. As a mitigation, we have suggested the use of aluminum wire-bonds instead of gold as it could both prevent the galvanic reactions, and maintain a strong interfacial adhesion with PDMS.

In the in vitro study, some pads (connected to adjacent non-tested structures) were intentionally left exposed to PBS solution, which showed heavy corrosion signs (**Figure S10**, Supplementary Information). Examining the pads using SEM cross-sectional analysis, however, revealed that except for the observed surface corrosion, the underlying layers of the IC material remained intact. This was mainly due to the anti-corrosive

properties of the thin TiN layer under the aluminum pad, which is used as a metal diffusion barrier in the semiconductor industry [4].

Reference:

[3] Iannuzzi, M. Bias Humidity Performance and Failure Mechanisms of Nonhermetic Aluminum SiC's in an Environment Contaminated with Cl₂. IEEE Transactions on Components, Hybrids, and Manufacturing Technology 6, (1983).

[4] Birkholz, M., K-E. Ehwald, D. Wolansky, I. Costina, C. Baristiran-Kaynak, M. Fröhlich, H. Beyer, A. Kapp, and F. Lisdat. "Corrosion-resistant metal layers from a CMOS process for bioelectronic applications." Surface and Coatings Technology 204, no. 12-13 (2010): 2055-2059.

Comment 1.6:

The most valuable result from the manuscript, in my opinion, is that the ICs remained functional and structurally intact under continuous DC bias for 12 months in vitro aging test at 67 degree. While it is well known that SiNx constructs a stable passivation layer, having these quantitative measures of longevity is valuable. The authors reported failures, too. It is important to clearly document the success rate in these tests.

Response 1.6:

We thank the reviewer for highlighting this point. Based on the reviewer's feedback, we have further clarified the success rate and observed failures in the manuscript.

Our modifications to the manuscript: Based on the reviewer's feedback and recommendations, the following modifications have been made.

Manuscript:

- **Results:** In total, n=34 ICs were tested for up to 12 months with n=17 being continuously biased with 5 V and n=17 with 15 V DC.
- **Results:** Some of our samples (14 out of 54) showed EIS irregularities (See Supplementary Table S6), which were attributed to either, 1) wire-bond corrosion (n=5) (See Supplementary Figures S9 and S10, Supplementary Information), 2) stress-induced cracking of the passivation layers, or 3) poor conformality of the passivation layers (n=9) (Supplementary Note 4).

Supplementary Information: **Modified tables S3 and S6 where the number of tested samples, duration in test and the time to failure has been specified.**

Comment 1.7:

Along this vein, the manuscript provides comprehensive characterizations of the longevity of IC implants under various in vitro and in vivo conditions, which will be valuable information for the field. However, I would like to request that the authors provide more specific details about the test conditions and results than what is currently included in

Table S6. Currently, the test durations are grouped together. Please include the sample number, time to failure, and failure mode for each test duration and condition.

Response 1.7:

We thank the reviewer for this comment which has been addressed below.

Our modifications to the manuscript: Based on the reviewer's feedback the following modifications have been made:

Supplementary Information: **Modified tables S3 and S6 as given in Response 1.6.**

Comment 1.8:

Furthermore, it would be helpful to label the samples used in each figure panel and to clearly indicate whether the same samples were used across multiple experiments. If the same samples were indeed reused, please justify this approach, as multiple samples should ideally be tested under identical conditions.

Response 1.8:

We thank the reviewer for this comment. In Table S3 (Supplementary Information) an explanation has been given on the material analysis procedure used for examining the samples.

For samples analyzed using both ToF-SIMS/XPS and SEM, the SEM analysis was performed after the ToF-SIMS/XPS analysis, as the sample surface must be coated with a thin conductive Pt layer for SEM. Therefore, all surface analysis techniques, such as AFM, ToF-SIMS, and XPS, were conducted prior to SEM.

Comment 1.9:

Some claims made should be more specific and not overstated. A few examples of potential overstated or misleading claims include:

The use of 'Hermeticity/Hermetic' in the title and throughout the manuscript, which I articulated above.

Response 1.9:

We have addressed this comment in Response 1.4.

Comment 1.10:

"Secondly, they show the electrical stability of the IC passivation and IMD layers surrounding the IDC metals when fully submerged in PBS solution for at least a year while being continuously subjected to a maximum electric field of 0.25 MV/cm (at 15V DC)" This section appears overly optimistic. Can the authors provide a more precise summary here, detailing the numbers of tested at various bias voltages and how many successfully lasted for 12 months? Additionally, please refer to Table S6 and modify it according to my earlier suggestions.

Response 1.10:

We thank the reviewer for their comment. For clarity on the test conditions and results, we have modified the manuscript as described in Response 1.6.

Our modifications to the manuscript: provided in Response 1.6.

Comment 1.11:

“Thirdly, regarding the PDMS coating, these results demonstrate that PDMS effectively prevents water leakage paths by forming long-lasting interfacial adhesion to the IC’s nitride passivation. In addition, they attest to the stable electrical insulation properties of PDMS while electrically biased and submerged in PBS solution.”

I don’t believe these results adequately address the role of PDMS. The authors have demonstrated that the IC passivation layer is stable without the presence of PDMS, but they did not test the interfacial adhesion between PDMS and the IC passivation layer. Even if the PDMS does not function as a barrier, the IC may still operate effectively. I will not enumerate all the instances along this line, but I suggest that the authors thoroughly review the manuscript to correct any overstated claims.

Response 1.11:

We thank the reviewer for their comment and would like to clarify a potential misunderstanding. As illustrated in Figure S1 (Supplementary Information), the PDMS also serves as an electrical insulator for the exposed gold wire-bonds and IC pads. This insulation is necessary to electrically isolate the metal wires from each other and from the surrounding salt solution. Any changes in the insulation or dielectric properties of PDMS would be expected to impact the impedance (EIS) measurement results. The stable EIS measurements over the one-year accelerated testing period suggest that PDMS’s electrical insulation properties remained unchanged, at least to a level which was undetectable when measuring impedances as high as ~100 GΩ (at 0.1 Hz).

Our modifications to the manuscript: Based on the reviewer’s feedback and for clarification the following modification have been made.

Supplementary Information: **included schematics** in Supplementary Note 1 showing how PDMS electrically insulates the metal wire bond both from each other and from the PBS solution.

Figure S1. A tilted optical micrograph of a representative wire-bonded M3-IDC test structure (Chip-A) used for the accelerated *in vitro* aging study demonstrating the 6 critical PDMS interface bonds on the test structure.

Other corrections and modifications to the manuscript:

- Changed the subscript of SiO_2 passivation to SiO_x for a clearer identification of its possible non-stoichiometric nature.
- In Figure 3c, correct a typo:
 - 0-month: Average: ~~470 (TΩ)~~ 474 (TΩ), ~~Std: 54 (TΩ)~~ to Std: 35 (TΩ)
 - 5-month: Average: ~~457 (TΩ)~~ 475 (TΩ), ~~Std: 65 (TΩ)~~ Std: 36 (TΩ)
- Added **Statistic and reproducibility:** For SEM cross-sectional analysis and AFM, each sample was measured at two separate locations with measured values being < 5% different from each other. For ToF-SIMS and XPS analysis, each sample was measured on two different locations for both the PDMS-coated and exposed regions of the IC surface. Data reproducibility was < 10%. For EIS electrical characterization, at each time point, the IDC structures were measured twice with reproducibility of the results being < 1%. Measurements were taken from different samples at different time points (see Table S3, Supplementary Information). For the dielectric sensor, at each time point, measurements across the entire sensor array were taken three consecutive times with measurements being < 5% different from each other.